# Outlier Synthesis via Hamiltonian Monte Carlo for Out-of-Distribution Detection

**Hengzhuang Li, Teng Zhang**[*]
School of Computer Science and Technology,
Huazhong University of Science and Technology, Wuhan, China
{hengzhuangli, tengzhang}@hust.edu.cn

## Abstract

Out-of-distribution (OOD) detection is crucial for developing trustworthy and reliable machine learning systems. Recent advances in training with auxiliary OOD data demonstrate efficacy in enhancing detection capabilities. Nonetheless, these methods heavily rely on acquiring a large pool of high-quality natural outliers. Some prior methods try to alleviate this problem by synthesizing virtual outliers but suffer from either poor quality or high cost due to the monotonous sampling strategy and the heavy-parameterized generative models. In this paper, we overcome all these problems by proposing the ***Ham**iltonian Monte Carlo **O**utlier **S**ynthesis* (HamOS) framework, which views the synthesis process as sampling from Markov chains. Based solely on the in-distribution data, the Markov chains can extensively traverse the feature space and generate diverse and representative outliers, hence exposing the model to miscellaneous potential OOD scenarios. The Hamiltonian Monte Carlo with sampling acceptance rate almost close to 1 also makes our framework enjoy great efficiency. By empirically competing with SOTA baselines on both standard and large-scale benchmarks, we verify the efficacy and efficiency of our proposed HamOS. Our code is available at:
https://github.com/Fir-lat/HamOS_OOD.

## 1 Introduction

Despite the impressive achievements of deep neural networks across various practical applications, they are unable to make dependable predictions when faced with *out-of-distribution* (OOD) samples (Nguyen et al., 2015; Bendale & Boult, 2016; Hendrycks & Gimpel, 2017), as the input data may not be drawn from *in-distribution* (ID) (Bommasani et al., 2021). To guarantee the reliability of machine learning systems, models should recognize data from unknown classes instead of making overconfident predictions, known as OOD detection. It has garnered significant attention within the Safety AI community, as erroneous predictions of OOD samples might result in perilous situations, especially vital for safety-critical applications like autonomous driving (Heidecker et al., 2021a;b) and medical image analysis (Ulmer et al., 2020; Linmans et al., 2020).

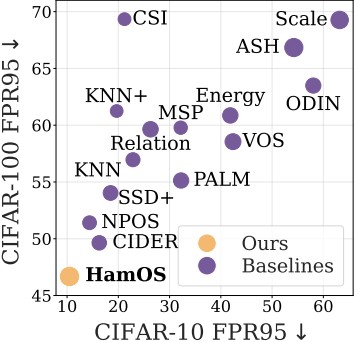

Figure 1: OOD detection performance on CIFAR and ImageNet. The size of the dots indicates AUROC w.r.t. ImageNet-1K.

Prior research can be categorized into two paradigms: post-hoc methods and regularization-based methods. The former focuses on designing a robust scoring function to differentiate between ID and OOD samples (Hendrycks & Gimpel, 2017; Liang et al., 2018; Liu et al., 2020; Mahalanobis, 1936; Sun et al., 2022), while the latter incorporates supplementary regularization during the training phase, with or without auxiliary OOD data, to enhance the model's discriminative ability (Hendrycks et al., 2019; Wei et al., 2022; Zhang et al., 2023a; Zhu et al., 2023b; Ming et al., 2023a), typically achieving better performance than post-hoc methods.

---

[*]Corresponding author.

Outlier exposure (OE) approaches (Hendrycks et al., 2019; Zhang et al., 2023a; Park et al., 2023), which employ meticulously gathered supplemental OOD data for training, can markedly enhance detection efficacy. Among them, certain studies (Chen et al., 2021; Ming et al., 2022; Jiang et al., 2024a) investigate sophisticated sampling strategies utilizing an extensive pool of auxiliary OOD data, while other studies (Zhang et al., 2023a; Wang et al., 2023; Zhu et al., 2023b; Zheng et al., 2023) suggest augmenting auxiliary outliers using a subset of natural outliers as anchors. However, these methods may be constrained as they require rigorously collected natural outlier data, which is infeasible for many domain-specific applications where high-quality outliers are expensive.

To address the issues encountered by OE methods, some studies (Du et al., 2022b; Tao et al., 2023; Du et al., 2023; Pourreza et al., 2021) propose to synthesize virtual outliers without assuming the availability of natural OOD data, resulting in a more flexible OOD-aware training scheme. Due to advancements in generative models (Goodfellow et al., 2020; Esser et al., 2021; Ho et al., 2020), certain studies (Lee et al., 2018a; Grcic et al., 2021; Kong & Ramanan, 2021; Du et al., 2023) generate outliers in pixel space, while others (Du et al., 2022b; Tao et al., 2023) primarily employ Gaussian sampling to generate outliers in feature space. The latter approach provides a more straightforward method for incorporating OOD supervision signals into training, avoiding the extensive computation of generative models. Despite attaining considerable performance, current studies either impose stringent assumptions on the IDs or exclusively sample from sub-regions near decision boundaries, resulting in synthesized outliers that lack diversity and representativeness, which is demonstrated to be crucial for learning with OOD outliers (Kong & Ramanan, 2021; Chen et al., 2021; Ming et al., 2022; Zhu et al., 2023b; Jiang et al., 2024a). Ensuring ID accuracy in the original work is also essential for regularization-based approaches. From the above analysis, a pivotal question arises:

*How can we efficiently synthesize diverse and representative outliers based solely on the ID data?*

Faced with the challenges mentioned above, we propose an innovative ***Ham**iltonian Monte Carlo **O**utlier **S**ynthesis* (**HamOS**) framework for OOD detection, presenting a paradigm shift in outlier synthesis by explicit sampling through Markov chains with ID feature embeddings. HamOS samples diverse and representative virtual outliers in the latent feature space, which act as crucial surrogate outlier supervision signals for learning essential unknown semantic information absent in the ID data. Unlike sampling through Gaussian distribution, we formulate the synthesis process as Markov chains, wherein a new outlier is generated based on the preceding one (see Figure 3). We characterize the OOD likelihood in hyperspherical space by a novel OOD-ness estimation to assess the probability level of a sample from unknown classes. An iterative sequence of outliers can be generated using Hamiltonian Monte Carlo (HMC) (Duane et al., 1987), one efficient sampling algorithm of score-based Markov Chain Monte Carlo (MCMC) (Metropolis et al., 1953; Neal, 1993), integrated with the OOD-ness estimation. The sequence of outliers exhibits varying degrees of OOD characteristics, which can yield distinct supervision signals for training. Figure 2 shows the OOD scores of generated virtual outliers, where HamOS generates more diverse outliers, which occupy a broader range of OOD scores with more significant variance.

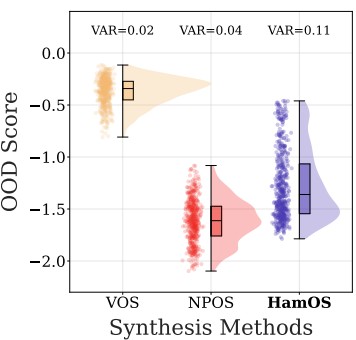

Figure 2: OOD scores of virtual outliers synthesized with different methods.

HamOS maintains and updates the ID feature embeddings during training as ID prior knowledge. Initialized with midpoints between ID clusters, HamOS chooses the data points with higher OOD-ness density values against the ID feature embeddings. Specifically, to sample a batch of outliers for training, we establish Markov chains between close ID clusters and gather the outliers in each Markov chain. To generate the next outlier feature in a Markov chain based on a pair of close ID clusters, we estimate the OOD-ness for candidate data using the averaged KNN distance (Gu et al., 2019; Zhao & Lai, 2022) from the pair of ID clusters. The candidate is chosen if it exhibits higher OOD-ness value. Additionally, to prevent erroneous outlier synthesis (i.e., points with higher OOD-ness values lie within ID clusters), we put a hard margin barrier that keeps the generated outliers away from ID clusters. The barrier is calculated via kernel density estimation with the von Mises-Fisher kernel. The generation pipeline ensures that we can get a mini-batch of outliers spanning a substantial area in the feature space while possessing diverse levels of OOD-ness. We then train the

model with OOD discernment loss and ID contrastive loss to learn a proper hyperspherical space. As for the inference-time OOD detection, we can adopt distance-based scoring functions to calculate detection scores. HamOS is a computationally tractable framework compatible with numerous HMC variants, ID contrastive losses, and scoring functions.

We conduct extensive empirical analysis to demonstrate the **state-of-the-art (SOTA)** performance of HamOS. On the standard OOD detection benchmark, HamOS significantly surpasses the competitive baselines that either synthesize virtual outliers (Du et al., 2022b; Tao et al., 2023) or regularize models (Tack et al., 2020; Sehwag et al., 2021; Sun et al., 2022; Ming et al., 2023b; Lu et al., 2024) by a considerable margin. Specifically, HamOS enhances the OOD detection performance indicated by FPR95 on CIFAR-10 and CIFAR-100 (Krizhevsky & Hinton, 2009) by $\mathbf{27.17}\%$ and $\mathbf{5.96}\%$ respectively, evaluated with five standard OOD test datasets. Experiments on ImageNet-1K (Deng et al., 2009) further demonstrate the flexibility for HamOS to scale to large scenarios. In Figure 1, HamOS demonstates superior performance on CIFAR-10/100 and ImageNet-1K. Comprehensive ablation studies elucidate the intrinsic mechanism of HamOS, showcasing its superiority against other baselines. Our primary contribution is delineated as follows:

- We are the first to investigate the new framework for outlier synthesis via Markov chains instead of sampling from Gaussian distribution. We hope this paper inspires more works to explore advanced algorithms to explicitly sample outliers from feature space.

- We propose the general Hamiltonian Monte Carlo Outlier Synthesis (HamOS) framework that generates diverse and representative virtual outliers to enhance ID-OOD separation in the latent space for better OOD detection performance.

- We conduct extensive experiments on standard and large-scale benchmarks to demonstrate that HamOS establishs **SOTA** performance compared with baselines. Comprehensive ablation is conducted to reveal the intrinsic mechanism and superiority of HamOS.

## 2 PRELIMINARIES

This paper studies OOD detection in multi-class classification setting, where $\mathcal{X} \subseteq \mathbb{R}^d$ is the input space and $\mathcal{Y} = \{1, 2, \ldots, C\}$ is the label set. Let $\mathcal{D}_{\text{train}}^{\text{ID}} = \{(\boldsymbol{x}_i, y_i)\}_{i=1}^n$ be the ID training dataset, drawn *i.i.d.* from the distribution $\mathcal{P}_{\mathcal{X}\mathcal{Y}}^{\text{ID}}$ on $\mathcal{X} \times \mathcal{Y}$, and $\mathcal{P}_{\mathcal{X}}^{\text{ID}}$ denote the marginal distribution on $\mathcal{X}$.

### 2.1 OOD DETECTION

OOD detection can be formulated as a binary classification task, i.e., for any input $\boldsymbol{x} \in \mathcal{X}$, the model should determine whether it is from $\mathcal{P}_{\mathcal{X}}^{\text{ID}}$ or $\mathcal{P}_{\mathcal{X}}^{\text{OOD}}$, where $\mathcal{P}_{\mathcal{X}}^{\text{OOD}}$ denotes some unknown distribution whose label set has no overlap with $\mathcal{Y}$. The prediction for OOD detection is an indicator function $\mathbf{1}\{S(\boldsymbol{x}) \leq \beta\}$ where $S(\boldsymbol{x})$ is some scoring function and $\beta$ is a threshold to ensure a high true positive rate of ID data (e.g., 95%). Given an input $\boldsymbol{x}$, it is identified as OOD data if $S(\boldsymbol{x}) \leq \beta$.

### 2.2 HAMILTONIAN MONTE CARLO

Inspired by the Hamiltonian mechanics (Hamilton, 1833), Hamiltonian Monte Carlo (HMC) (Duane et al., 1987) is proposed as an efficient Markov Chain Monte Carlo (MCMC) method. Given the target distribution density $P(\boldsymbol{z})$, HMC views $\boldsymbol{z}$ as position and introduces an auxiliary momentum $\boldsymbol{q}$, then each state is described as a tuple $(\boldsymbol{z}, \boldsymbol{q})$ and a new proposal $(\boldsymbol{z}^*, \boldsymbol{q}^*)$ is generated via the evolution of the Hamilton's Equation. After the Markov chain has burned in, the position sequence $\{\boldsymbol{z}_i\}$ are samples from the target distribution. Compared to the vanilla MCMC method (e.g., Metropolis algorithm), HMC can generate a succession of samples with less dependence meanwhile keep the rejection probability arbitrarily small, by making the evolution traverse a long distance in the state space with fine-tuned small step size. These two properties make it suitable for generating diverse outliers for regularization-based OOD detection. Please refer to Appendix C for more details.

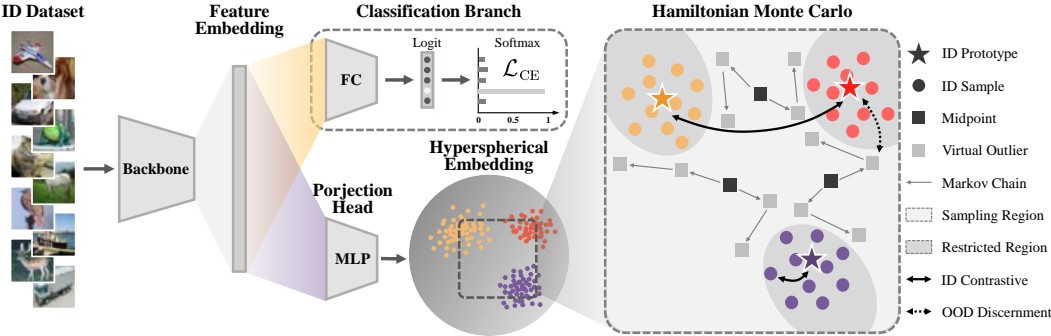

Figure 3: **Depiction of the HamOS training framework.** We design a dual-head framework utilizing a backbone for feature extraction. (1) The FC head preserves the initial ID classification efficacy; (2) The projection head transforms the feature embedding into a reduced-dimensional hyperspherical space, where we explicitly generate outliers through Hamiltonian Monte Carlo utilizing our innovative OOD-ness estimation. The spherical space is shaped by ID contrastive loss and OOD discernment loss to enhance differentiation between ID and OOD data.

## 3 METHOD: HAMILTONIAN MONTE CARLO OUTLIER SYNTHESIS

This section details how HamOS synthesizes outliers using solely the ID training data and how outliers are utilized to regularize the decision boundaries for better ID-OOD differentiation.

### 3.1 OVERVIEW

As illustrated in Figure 3, our framework HamOS consists of three components: 1) a feature embedding $f : \mathcal{X} \mapsto \mathcal{H}$ that maps input $\boldsymbol{x}$ into some higher dimensional feature space; 2) a fully connected (FC) layer: $f(\boldsymbol{x}) \mapsto \mathcal{Y}$ for the original ID classification task; 3) a latent embedding: $f(\boldsymbol{x}) \mapsto \mathrm{norm}(\mathrm{MLP}(f(\boldsymbol{x})))$ where $\mathrm{MLP}(\cdot)$ is a multiple layer perceptron and $\mathrm{norm}(\cdot)$ is the $\ell_2$ normalization so that the output embedding is on the unit hypersphere. HMC also generates outliers on this hypersphere. Different from previous works (Du et al., 2022b; Tao et al., 2023) which simply add Gaussian noise to boundary ID samples, our method utilizes Markov chains to explore a significantly larger area in the hyperspherical space. By setting the potential energy as the estimated OOD-ness density, the samples from the Markov chains of HMC are diverse and representative, ensuring the trained model with great power to distinguish between ID and OOD data.

### 3.2 SYNTHESIZING OUTLIERS VIA HAMILTONIAN MONTE CARLO

**Estimating OOD-ness via the distance to the $k$-th nearest neighbor.** Although we have only access to the ID data, we can measure the OOD-ness for any sample, i.e., a quantitative characterizing the likelihood that a sample is OOD rather than ID (Dang et al., 2015; Bergman et al., 2020; Sun et al., 2022), by leveraging its distance to the ID data. Specifically, for any embedding $\boldsymbol{z}$ lying on the unit hypersphere and the ID data embedding sets $\mathcal{Z}_1, \ldots, \mathcal{Z}_C$, one for each class, the OOD-ness for $\boldsymbol{z}$ w.r.t. $\mathcal{Z}_c$ is defined as the Euclidean distance:

$$P^{\mathrm{OOD}}(\boldsymbol{z}; \mathcal{Z}_c) = \|\boldsymbol{z} - \boldsymbol{z}_{c(k)}\|_2, \tag{1}$$

where $\boldsymbol{z}_{c(k)}$ is the $k$-th nearest neighbor in $\mathcal{Z}_c$ for $\boldsymbol{z}$. As the feature embeddings are projected to the unit hypersphere, our primary goal is to synthesize outliers lying between ID clusters, which are critical for shaping the ID boundaries and exposing the model to critical potential OOD scenarios. Consequently, for each pair of ID classes $\mathcal{Z}_u$ and $\mathcal{Z}_v$, we measure the OOD-ness for $\boldsymbol{z}$ by

$$P^{\mathrm{OOD}}(\boldsymbol{z}; \mathcal{Z}_u, \mathcal{Z}_v) = \frac{P^{\mathrm{OOD}}(\boldsymbol{z}; \mathcal{Z}_u) + P^{\mathrm{OOD}}(\boldsymbol{z}; \mathcal{Z}_v)}{2},$$

and run HMC with the potential energy as

$$U^{\mathrm{OOD}}(\boldsymbol{z}; \mathcal{Z}_u, \mathcal{Z}_v) = -\log P^{\mathrm{OOD}}(\boldsymbol{z}; \mathcal{Z}_u, \mathcal{Z}_v) = -\log \sum_{i=u,v} P^{\mathrm{OOD}}(\boldsymbol{z}; \mathcal{Z}_i) + \mathrm{constant}. \tag{2}$$

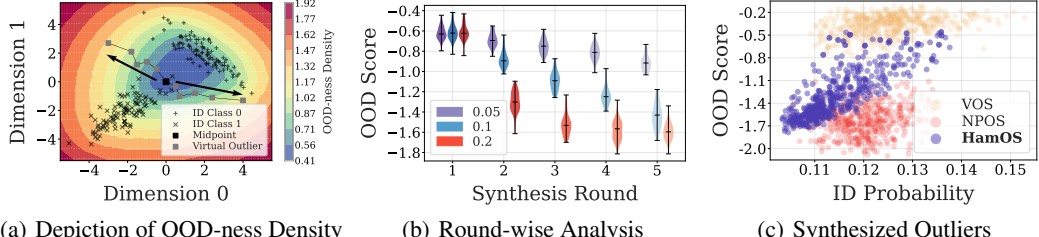

(a) Depiction of OOD-ness Density     (b) Round-wise Analysis     (c) Synthesized Outliers

Figure 4: **HamOS synthesizes outliers of varying levels of OOD-ness:** (a) OOD-ness density with illustrated synthesis process; (b) the OOD distribution of the virtual outliers at each synthesis round with different step sizes; (c) the ID probabilities and OOD scores of the synthesized outliers.

Figure 4(a) depicts the geometry of the OOD-ness estimation, showing that the OOD-ness density is low between ID clusters and high in the peripheral region. More analysis on the design of the OOD-ness density estimation is provided in Appendix G.

**Synthesizing outliers by OOD density estimation via HMC.** HMC incorporates an auxiliary momentum $q$ sampled from Gaussian distribution for describing the state as $(z, q)$. The *Hamiltonian* (Duane et al., 1987) is formulated as $H(z, q) = U^{\text{OOD}}(z) + \|q\|_2^2/2$ where the potential energy $U^{\text{OOD}}(z)$ is defined in Eqn. (2), thus the virtual outliers collected along the Markov chains obey the marginal distribution $\propto P^{\text{OOD}}$.

Given the potential energy, the Leapfrog discretization is typically adopted as the numerical approximation of the evolution of Hamilton's Equation (Hamilton, 1833). For HamOS, the target distribution lies on the unit hypersphere, thus the Spherical HMC (Lan et al., 2013) is applied and the Leapfrog update on the unit sphere turns to:

$$
\begin{aligned}
q^{(\ell+1/2)} &= q^{(\ell)} - \frac{\epsilon}{2}(\mathbf{I}_d - z^{(\ell)}(z^{(\ell)})^\top)\nabla_{z^{(\ell)}}U^{\text{OOD}}(z^{(\ell)}), \\
z^{(\ell+1)} &= z^{(\ell)}\cos\left(\|q^{(\ell+1/2)}\|_2\epsilon\right) + \frac{q^{(\ell+1/2)}}{\|q^{(\ell+1/2)}\|_2}\sin\left(\|q^{(\ell+1/2)}\|_2\epsilon\right), \\
q^{(\ell+1/2)} &\leftarrow -z^{(\ell)}\|q^{(\ell+1/2)}\|_2\sin\left(\|q^{(\ell+1/2)}\|_2\epsilon\right) + q^{(\ell+1/2)}\cos\left(\|q^{(\ell+1/2)}\|_2\epsilon\right), \\
q^{(\ell+1)} &= q^{(\ell+1/2)} - \frac{\epsilon}{2}(\mathbf{I}_d - z^{(\ell+1)}(z^{(\ell+1)})^\top)\nabla_{z^{(\ell+1)}}U^{\text{OOD}}(z^{(\ell+1)}),
\end{aligned}
\tag{3}
$$

where $\ell$ is the Leapfrog timestep, $\epsilon$ is the step size. Eqn. (3) ensures new candidates lie on the unit sphere. According to the Hamilton's Equation (Hamilton, 1833), the energy function $U^{\text{OOD}}(z)$ must be differentiable to ensure the solvability, with the gradient of the energy function calculated as:

$$
\nabla_z\left(U^{\text{OOD}}(z; \mathcal{Z}_u, \mathcal{Z}_v)\right) = -\exp\left(-U^{\text{OOD}}(z; \mathcal{Z}_u, \mathcal{Z}_v)\right)\cdot\left(\frac{z - z_{u(k)}}{\|z - z_{u(k)}\|_2} + \frac{z - z_{v(k)}}{\|z - z_{v(k)}\|_2}\right). \tag{4}
$$

The primary goal of synthesizing outliers via HMC is to generate outliers that represent diverse levels of OOD-ness, as well as spanning an extensive region in hyperspherical space, without necessitating the convergence to the target distribution. Given a pair of ID clusters $\mathcal{Z}_u$ and $\mathcal{Z}_v$, the midpoint of the two clusters $\mathbf{b}_{u,v} = \frac{\mu_u + \mu_v}{\|\mu_u + \mu_v\|_2}$ is employed as the initial state of a Markov chain due to its relatively low OOD-ness density, where $\mu_u$ and $\mu_v$ are the prototypes of the two ID clusters. Figure 4(a) shows the midpoint has a meager OOD-ness density value. We provide round-wise analysis in Figure 4(b) with different step size $\epsilon$ where outliers with varied levels of OOD-ness are generated in each round, exhibiting a broad range of OOD characteristics, also illustrated in Figure 2.

**Rejecting erroneous outliers located within ID clusters.** HMC performs the Metropolis-Hastings (Metropolis et al., 1953; Hastings, 1970) acceptance step to ensure that the newly generated sample possesses a reduced energy value. However, the design of potential energy of OOD-ness neglects the ID likelihood, potentially resulting in false outliers conflated with ID embeddings. Consequently, we introduce a hard margin in the acceptance step in HMC. We utilize the kernel density estimation (KDE) (Rosenblatt, 1956) with von Mises-Fisher kernel (Mardia & Jupp, 2009; Fisher, 1953) to approximate the ID probability, detailed in Appendix C.2. The hard margin $\delta$ is applied to

---

**Algorithm 1** Hamiltonian Monte Carlo Outlier Synthesis (HamOS)

---

**Input:** ID training embeddings: $\mathcal{Z}$, number of ID classes: $C$, synthesis rounds: $R$, Leapfrog steps: $L$, step size: $\epsilon$, number of adjacent ID clusters: $N_{\text{adj}}$, $k$-th nearest neighbor: $k$, hard margin: $\delta$ ;
**Output:** a batch of virtual outliers $\mathcal{Z}^{\text{OOD}}$;

1: **for** class $c = 1, \cdots, C$ **do**
2:     Calculate the initial points $\mathbf{b}_{cj} = \frac{\boldsymbol{\mu}_c + \boldsymbol{\mu}_j}{\|\boldsymbol{\mu}_c + \boldsymbol{\mu}_j\|_2}$, where $\boldsymbol{\mu}_i$ is the centroid of $\mathcal{Z}_i$, and
3:     $j = 1, \cdots, N_{\text{adj}}$ indexes the closest ID clusters from $\mathcal{Z}_c$ measured by cosine similarity.
4:     Set the current position $\boldsymbol{z} = \mathbf{b}_{cj}$.
5:     Calculate the hard margin threshold $t_-$ as Eqn. (5).
6:     **for** synthesis round $r = 1, \cdots, R$ **do**
7:         Initialize $\boldsymbol{z}^{(1)}$ at current position $\boldsymbol{z}$.
8:         Sample a new momemtum $\boldsymbol{q}^{(1)} \sim \mathcal{N}(0, \mathbf{I}_d)$.
9:         Set $\boldsymbol{q} \leftarrow \boldsymbol{q} - \boldsymbol{z}^{(1)}(\boldsymbol{z}^{(1)})^\top \boldsymbol{q}^{(1)}$.
10:        Calculate $H(\boldsymbol{z}^{(1)}, \boldsymbol{q}^{(1)}) = U^{\text{OOD}}(\boldsymbol{z}^{(1)}) + K(\boldsymbol{q}^{(1)})$.
11:        **for** $\ell = 1, \cdots, L$ **do**
12:           $\boldsymbol{q}^{(\ell+1/2)} = \boldsymbol{q}^{(\ell)} - \frac{\epsilon}{2}(\mathbf{I}_d - \boldsymbol{z}^{(\ell)}(\boldsymbol{z}^{(\ell)})^\top)\nabla_{\boldsymbol{z}^{(\ell)}} U^{\text{OOD}}(\boldsymbol{z}^{(\ell)})$.
13:           $\boldsymbol{z}^{(\ell+1)} = \boldsymbol{z}^{(\ell)} \cos(\|\boldsymbol{q}^{(\ell+1/2)}\|_2\epsilon) + \frac{\boldsymbol{q}^{(\ell+1/2)}}{\|\boldsymbol{q}^{(\ell+1/2)}\|_2} \sin(\|\boldsymbol{q}^{(\ell+1/2)}\|_2\epsilon)$.
14:           $\boldsymbol{q}^{(\ell+1/2)} \leftarrow -\boldsymbol{z}^{(\ell)}\|\boldsymbol{q}^{(\ell+1/2)}\|_2 \sin(\|\boldsymbol{q}^{(\ell+1/2)}\|_2\epsilon) + \boldsymbol{q}^{(\ell+1/2)} \cos(\|\boldsymbol{q}^{(\ell+1/2)}\|_2\epsilon)$.
15:           $\boldsymbol{q}^{(\ell+1)} = \boldsymbol{q}^{(\ell+1/2)} - \frac{\epsilon}{2}(\mathbf{I}_d - \boldsymbol{z}^{(\ell+1)}(\boldsymbol{z}^{(\ell+1)})^\top)\nabla_{\boldsymbol{z}^{(\ell+1)}} U^{\text{OOD}}(\boldsymbol{z}^{(\ell+1)})$.
16:        **end for**
17:        Calculate $H(\boldsymbol{z}^{(L+1)}, \boldsymbol{q}^{(L+1)}) = U^{\text{OOD}}(\boldsymbol{z}^{(L+1)}) + K(\boldsymbol{q}^{(L+1)})$ for the proposed state.
18:        Calculate the acceptance probability $\alpha = \exp\{-H(\boldsymbol{z}^{(L+1)}, \boldsymbol{q}^{(L+1)}) + H(\boldsymbol{z}^{(1)}, \boldsymbol{q}^{(1)})\}$.
19:        Accept $\boldsymbol{z}^{(L+1)}$ as current position $\boldsymbol{z}$ according to $\alpha$ and $-\log\max_c P_c^{\text{ID}}(\boldsymbol{z}^{(L+1)}) > t_-$.
20:     **end for**
21:     Gather the sequential positions in the $R$ rounds as $\mathcal{Z}_c^{\text{OOD}}$.
22: **end for**

---

the log-likelihood of the initial point $\mathbf{b}_{u,v}$ to form a lower threshold, defined as follows:

$$t_- = -\log\max_c P_c^{\text{ID}}(\mathbf{b}_{u,v}) - \delta. \tag{5}$$

where the estimated probability $P_{\text{ID}}^c$ is defined in Eqn. (11). The hard margin will block the new proposed point if it has a much higher ID probability than the initial point. Figure 4(c) shows that HamOS generates outliers with diverse OOD scores and low ID-likelihood.

We sample multiple Markov chains to form a mini-batch of outliers, parallelly generating outliers starting from midpoints w.r.t. the nearest $N_{\text{adj}}$ ID clusters for each ID class. The outlier batch encompasses a broad range of potential OOD scenarios, covering a massive area in hyperspherical space. We adopt the Spherical HMC (Lan et al., 2013) for sampling on the unit sphere. The pseudo-code is provided in Algorithm 1, with the whole training pipeline displayed in Algorithm 2 in Appendix D.

### 3.3 TRAINING WITH SYNTHESIZED OUTLIERS

The model is regularized jointly with ID training data and synthesized virtual outliers to produce embeddings with enhanced ID-OOD discernibility. The virtual outliers reside between ID clusters where the model is not optimized. Analogous to maximum likelihood estimation (MLE), our goal is to minimize the likelihood of being assigned to any ID classes for the synthesized outlier data :

$$\arg\min_\theta \prod_{i=1}^{M} p(y_j | \boldsymbol{z}_i; \{\kappa_j, \boldsymbol{\mu}_j\}_{j=1}^{C}), \tag{6}$$

where $\boldsymbol{\mu}_j$ is the ID prototype of class $j$ updated in exponential-moving-average (EMA) manner, $\boldsymbol{z}_i$ is

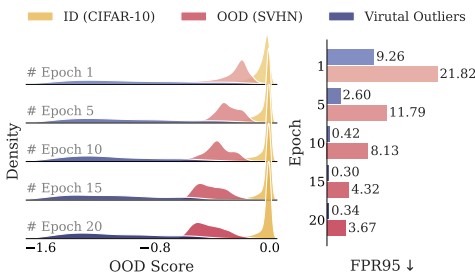

Figure 5: OOD performance is improved continuously with synthesized outliers.

Table 1: Main results with CIFAR-10/100 as ID dataset (%). Comparison with competitive OOD detection baselines. Results are averaged across multiple OOD test datasets with multiple runs.

| Methods | CIFAR-10 | | | | CIFAR-100 | | | |
|---|---|---|---|---|---|---|---|---|
| | FPR95↓ | AUROC↑ | AUPR↑ | ID-ACC↑ | FPR95↓ | AUROC↑ | AUPR↑ | ID-ACC↑ |
| *Post-hoc Methods* | | | | | | | | |
| MSP | $32.17_{\pm6.38}$ | $91.10_{\pm0.71}$ | $81.70_{\pm5.82}$ | $\mathbf{95.17_{\pm0.16}}$ | $59.78_{\pm2.16}$ | $77.25_{\pm1.28}$ | $66.86_{\pm1.58}$ | $\mathbf{76.69_{\pm0.24}}$ |
| ODIN | $58.04_{\pm18.46}$ | $85.70_{\pm4.17}$ | $70.08_{\pm11.84}$ | $\mathbf{95.17_{\pm0.16}}$ | $63.49_{\pm2.51}$ | $78.01_{\pm1.62}$ | $65.20_{\pm2.19}$ | $\mathbf{76.69_{\pm0.25}}$ |
| EBO | $41.85_{\pm13.78}$ | $91.79_{\pm1.54}$ | $79.70_{\pm8.10}$ | $\mathbf{95.17_{\pm0.16}}$ | $60.86_{\pm1.87}$ | $78.32_{\pm1.31}$ | $66.73_{\pm1.35}$ | $\mathbf{76.69_{\pm0.24}}$ |
| KNN | $22.86_{\pm1.12}$ | $92.98_{\pm0.42}$ | $88.74_{\pm0.79}$ | $\mathbf{95.17_{\pm0.16}}$ | $56.96_{\pm2.96}$ | $81.01_{\pm1.19}$ | $70.60_{\pm2.29}$ | $\mathbf{76.69_{\pm0.24}}$ |
| ASH | $54.22_{\pm26.06}$ | $87.37_{\pm6.60}$ | $72.33_{\pm16.40}$ | $95.10_{\pm0.14}$ | $66.84_{\pm0.87}$ | $77.14_{\pm1.12}$ | $62.24_{\pm0.73}$ | $76.20_{\pm0.23}$ |
| Scale | $63.18_{\pm23.64}$ | $77.74_{\pm16.24}$ | $63.03_{\pm20.52}$ | $95.15_{\pm0.16}$ | $69.27_{\pm2.31}$ | $77.25_{\pm1.01}$ | $61.42_{\pm1.42}$ | $\mathbf{76.69_{\pm0.24}}$ |
| Relation | $26.28_{\pm1.63}$ | $92.31_{\pm0.43}$ | $86.75_{\pm0.98}$ | $\mathbf{95.17_{\pm0.16}}$ | $59.64_{\pm2.48}$ | $79.69_{\pm1.08}$ | $68.76_{\pm1.78}$ | $\mathbf{76.69_{\pm0.24}}$ |
| *Regularization-based Methods* | | | | | | | | |
| CSI | $21.21_{\pm1.68}$ | $93.73_{\pm0.33}$ | $89.74_{\pm0.68}$ | $92.03_{\pm0.72}$ | $69.34_{\pm0.86}$ | $73.46_{\pm0.37}$ | $61.57_{\pm0.75}$ | $61.75_{\pm0.15}$ |
| SSD+ | $18.49_{\pm1.20}$ | $94.85_{\pm0.57}$ | $90.88_{\pm0.83}$ | $93.95_{\pm0.57}$ | $54.03_{\pm1.92}$ | $80.64_{\pm0.60}$ | $69.73_{\pm1.09}$ | $75.63_{\pm0.39}$ |
| KNN+ | $19.68_{\pm1.86}$ | $94.41_{\pm0.66}$ | $90.46_{\pm0.66}$ | $93.79_{\pm0.63}$ | $61.25_{\pm0.81}$ | $78.24_{\pm0.93}$ | $66.64_{\pm0.88}$ | $72.18_{\pm0.58}$ |
| VOS | $42.37_{\pm21.13}$ | $91.42_{\pm3.38}$ | $79.16_{\pm11.62}$ | $95.05_{\pm0.05}$ | $58.55_{\pm1.53}$ | $81.40_{\pm0.62}$ | $68.33_{\pm1.61}$ | $74.71_{\pm0.07}$ |
| CIDER | $16.28_{\pm0.68}$ | $95.76_{\pm0.37}$ | $92.36_{\pm0.06}$ | $93.98_{\pm0.16}$ | $49.64_{\pm1.80}$ | $81.77_{\pm0.95}$ | $73.22_{\pm1.12}$ | $75.09_{\pm0.49}$ |
| NPOS | $14.39_{\pm0.87}$ | $96.61_{\pm0.26}$ | $93.35_{\pm0.74}$ | $93.95_{\pm0.13}$ | $51.41_{\pm1.88}$ | $81.02_{\pm0.98}$ | $72.49_{\pm1.54}$ | $74.53_{\pm0.62}$ |
| PALM | $32.25_{\pm4.14}$ | $90.54_{\pm1.46}$ | $84.44_{\pm2.14}$ | $93.93_{\pm0.98}$ | $55.13_{\pm0.97}$ | $79.95_{\pm1.26}$ | $70.21_{\pm1.38}$ | $74.67_{\pm0.36}$ |
| **HamOS(ours)** | $\mathbf{10.48_{\pm0.76}}$ | $\mathbf{97.11_{\pm0.26}}$ | $\mathbf{94.94_{\pm0.86}}$ | $94.67_{\pm0.15}$ | $\mathbf{46.68_{\pm1.44}}$ | $\mathbf{83.64_{\pm0.64}}$ | $\mathbf{75.52_{\pm1.30}}$ | $76.12_{\pm0.14}$ |

the synthesized embedding of the outlier batch of size $M$. Specifically, the objective function is equivalent to minimizing the following OOD discernment loss in log-likelihood form:

$$\mathcal{L}_{\text{OOD-disc}} = \frac{1}{M} \sum_{i=1}^{M} \frac{1}{C} \sum_{j=1}^{C} \log \frac{\exp\left(\boldsymbol{z}_i^\top \boldsymbol{\mu}_j / \tau\right)}{\sum_{l=1}^{C} \exp\left(\boldsymbol{z}_i^\top \boldsymbol{\mu}_l / \tau\right)}, \tag{7}$$

where $\tau = 1/\kappa$ is the temperature. Since highly distinguishable ID representations are crucial for OOD detection (Tack et al., 2020; Sehwag et al., 2021; Ming et al., 2023b; Tao et al., 2023), we leverage the contrastive learning objective for shaping compact ID boundaries. Compatible with multiple contrastive losses (Chen et al., 2020; Khosla et al., 2020; Ming et al., 2023b) (shown in Table 12), HamOS adopts CIDER loss (Ming et al., 2023b) defaultly. We train with Cross-entropy loss on the classification head simultaneously. Our training objective is formalized as follows:

$$\mathcal{L}_{\text{HamOS}} = \mathcal{L}_{\text{CE}} + \mathcal{L}_{\text{ID-con}} + \lambda_d \mathcal{L}_{\text{OOD-disc}}, \tag{8}$$

where $\lambda_d$ is the trade-off co-efficient to balance $\mathcal{L}_{\text{OOD-disc}}$ and the other two terms. In Figure 5, we display the score distributions along the training phase, showing that the synthesized outliers span a wide range of OOD scores which help broaden the gap between ID and OOD data.

## 3.4 INFERENCE-TIME OOD DETECTION

We adopt the KNN distance scoring function as (Sun et al., 2022) for OOD detection during inference-time: $D_\beta(\boldsymbol{z}^*; k) = \mathbf{1}\{-\|\boldsymbol{z}^* - \mathcal{Z}_{(k)}\|_2 \geq \beta\}$, where $\mathcal{Z} = \{\boldsymbol{z}_1, \boldsymbol{z}_2, \cdots, \boldsymbol{z}_N\}$ is the ID embedding set, $\mathcal{Z}_{(k)}$ denotes the $k$-th nearest neighbor in $\mathcal{Z}$ w.r.t. $\boldsymbol{z}^*$, and $\mathbf{1}\{\cdot\}$ is the indicator function. The threshold $\beta$ is chosen to correctly classify a large part of ID data as ID (i.e., TPR=95%).

## 4 EXPERIMENTS AND ANALYSIS

This section consists of an extensive empirical analysis of the proposed HamOS. First, we provide the experimental settings in detail (in Section 4.1); second, we report the performance comparison of HamOS with a series of state-of-the-art baselines (in Section 4.2); third, we present further discussions and ablation studies to elucidate the intrinsic mechanism of HamOS (in Section 4.3).

## 4.1 SETUP

We display the critical parts of the experimental settings for the main performance comparison. For more details of experiment setups, please refer to Appendix D.2.

**Datasets.** Following the common practice for benchmarking the OOD detection (Zhang et al., 2023b), we use CIFAR-10, CIFAR-100 (Krizhevsky & Hinton, 2009) and ImageNet-1K (Deng et al., 2009) as ID datasets, and adopt a series of datasets as OOD testing data. For CIFAR ID datasets, we use MNIST (Deng, 2012), SVHN (Netzer et al., 2011), Textures (Cimpoi et al., 2014), Places365 (Zhou et al., 2017), and LSUN (Yu et al., 2015) as OOD testing data; for ImageNet-1K, we use iNatualist (Van Horn et al., 2018), Textures (Cimpoi et al., 2014), SUN (Xiao et al., 2010), and Places365 (López-Cifuentes et al., 2020) as OOD testing data.

Table 2: Results with ImageNet-1K as ID dataset (%). Comparison with regularization-based OOD detection baselines. Results are averaged across multiple runs.

| Methods | SUN | | | iNaturalist | | | Places-365 | | | Textures | | | Averaged | | | ID-ACC↑ |
|---|---|---|---|---|---|---|---|---|---|---|---|---|---|---|---|---|
| | FPR95↓ | AUROC↑ | AUPR↑ | FPR95↓ | AUROC↑ | AUPR↑ | FPR95↓ | AUROC↑ | AUPR↑ | FPR95↓ | AUROC↑ | AUPR↑ | FPR95↓ | AUROC↑ | AUPR↑ | |
| CSI | 54.77 | 77.35 | 94.50 | 45.34 | 85.18 | 97.30 | 60.01 | 75.50 | 80.81 | 45.43 | 88.44 | 96.79 | 51.38 | 81.62 | 92.35 | 67.76 |
| SSD+ | 56.86 | 80.86 | 94.64 | 27.69 | 91.80 | 98.01 | 61.54 | 79.02 | 84.12 | 38.62 | 89.97 | 98.61 | 46.18 | 85.41 | 93.85 | **76.52** |
| KNN+ | 59.48 | 76.56 | 93.60 | 42.95 | 81.77 | 95.63 | 61.80 | 76.59 | 83.15 | **36.57** | **90.86** | 98.75 | 50.20 | 81.44 | 92.78 | 76.34 |
| VOS | **48.11** | **87.89** | **96.72** | 30.66 | 91.35 | 97.96 | 58.59 | **83.23** | **86.75** | 46.49 | 89.22 | 98.39 | 45.96 | 87.92 | 94.95 | 76.25 |
| CIDER | 53.73 | 81.37 | 94.73 | 30.95 | 89.62 | 97.59 | **57.92** | 80.44 | 85.35 | 42.64 | 88.06 | 98.35 | 46.31 | 84.88 | 94.01 | 73.12 |
| NPOS | 61.84 | 79.03 | 93.93 | 34.59 | 87.75 | 97.00 | 63.80 | 78.21 | 83.39 | 41.40 | 88.83 | 98.48 | 50.41 | 83.45 | 93.20 | 69.92 |
| PALM | 53.04 | 83.83 | 95.35 | **25.33** | 93.99 | 98.55 | 61.03 | 79.96 | 84.52 | 43.09 | 88.20 | 98.39 | 45.62 | 86.50 | 94.20 | 74.10 |
| **HamOS(ours)** | 51.08 | 85.81 | 96.35 | 25.65 | **95.91** | **99.02** | 59.80 | 82.11 | 86.65 | 41.84 | 90.55 | **98.94** | **44.59** | **88.59** | **95.24** | 76.10 |

**Training scheme.** We assume the existence of pretrained models and constrict the training phase to the fine-tuning scenario, which is more feasible and practical in real-world cases. We adopt pretrained ResNet-34 (He et al., 2016) for CIFAR datasets and ResNet-50 (He et al., 2016) for ImageNet-1K dataset and use stochastic gradient descent (SGD) (Bottou, 2010) to fine-tune for 20 epochs using the ID training datasets solely, with drop out rate, momentum, and weight decay set to 0, 0.9, and $1.0 \times 10^{-4}$ respectively. The learning rate is initialized as 0.01 and decayed to 0 using cosine annealing (Loshchilov & Hutter, 2017). We synthesize a batch of outliers at each iteration.

**OOD detection baselines.** We evaluate the performance of the proposed HamOS against other competing baselines across two research lines: (1) for post-hoc methods, we adopt MSP (Hendrycks & Gimpel, 2017), ODIN (Liang et al., 2018), EBO (Liu et al., 2020), KNN (Sun et al., 2022), ASH (Djurisic et al., 2023), Scale (Xu et al., 2024), and Relation (Kim et al., 2023); (2) for regularization-based methods, we adopt CSI (Tack et al., 2020), SSD+ (Sehwag et al., 2021), KNN+ (Sun et al., 2022), VOS (Du et al., 2022b), CIDER (Ming et al., 2023b), NPOS (Tao et al., 2023), and PALM (Lu et al., 2024). The post-hoc methods are evaluated on pretrained models, whereas the regularization-based methods are evaluated after fine-tuning the pretrained models.

**Evaluation metrics.** We employ the following three prevalent metrics to assess the performance of OOD detection: (1) the Area Under the Receiver Operating Characteristic curve (AUROC) (Davis & Goadrich, 2006; Fawcett, 2006); (2) the Area Under the Precision-Recall curve (AUPR) (Manning & Schutze, 1999); and (3) the False Positive Rate (FPR) at 95% True Positive Rate (TPR), abbreviated as FPR95 (Liang et al., 2018). The AUROC quantifies the diagnostic efficacy of level-set estimation; the AUPR is especially suited for imbalanced scenarios; and FPR95 is a sensitive metric at high recall levels. We also report the classification accuracy on the ID datasets (ID-ACC) to evaluate the model's performance on the original classification task. The mean results across multiple runs with standard deviation are reported to demonstrate the stability of HamOS.

## 4.2 MAIN RESULTS

**HamOS outperforms competitive baselines.** In Table 1, we report the averaged results using five OOD testing datasets for the baselines and the proposed HamOS. The results show that the proposed HamOS can remarkably improve the OOD detection capability of the pretrained model. First, for the contrastive training-based methods (i.e., CSI, SSD+, KNN+, CIDER, and PALM), HamOS can additionally simulate representative supervision signals of OOD data that lies between ID classes, hence facilitating learning more proper representation on the unit sphere. Specifically, HamOS surpasses the CIDER by **35.63**% and **5.96**% in FPR95, respectively, on CIFAR-10 and CIFAR-100. Second, as for the synthesis-based methods (i.e., VOS and NPOS), the proposed HamOS generates outliers according to OOD-ness estimation rather than applying Gaussian noise to ID embeddings in low-likelihood regions, resulting in more diverse and representative outliers. Specifically, HamOS surpasses NPOS by **27.17**% and **9.20**% in FPR95 on CIFAR-10 and CIFAR-100 respectively. In general, the regularization-based methods can outperform the post-hoc methods by introducing more specific training, as the OOD metrics show, but at the expense of performance on the original classification task (i.e., lower ID-ACC). Nonetheless, HamOS achieves high OOD detection performance while preserving competitive ID classification performance. The detailed results for each OOD test dataset are in Table 5 in Appendix F. We also present the results on hard OOD benchmarks (Zhang et al., 2023b) in Table 6, 7 in Appendix F to validate the efficacy of HamOS under hard scenarios.

**HamOS is superior under the large-scale scene.** We also assess the efficacy of HamOS on the large-scale ImageNet-1K dataset. Table 2 presents the results compared with regularization-based baselines. As the results suggest, our proposed HamOS outstrips all the regularization-based baselines on average, improving the FPR95 by **2.98**% compared with VOS. We also provide evaluation

Table 3: Ablation results on compactness and separation (%). Comparison with competitive OOD detection baselines. Results are averaged across multiple runs.

| Method | ID-OOD Separation↑ | | | | | | Inter-class Dispersion↑ | Intra-class Compactness↓ |
|---|---|---|---|---|---|---|---|---|
| | MNIST | SVHN | Textures | Places365 | LSUN | **Averaged** | | |
| SSD+ | $18.50_{\pm 0.79}$ | $22.67_{\pm 0.68}$ | $19.75_{\pm 0.43}$ | $19.44_{\pm 0.10}$ | $22.75_{\pm 0.50}$ | $20.62_{\pm 0.23}$ | $53.70_{\pm 1.02}$ | $15.93_{\pm 0.09}$ |
| CIDER | $56.07_{\pm 1.64}$ | $68.33_{\pm 0.35}$ | $58.03_{\pm 0.47}$ | $54.77_{\pm 0.33}$ | $57.98_{\pm 3.87}$ | $59.04_{\pm 1.08}$ | $95.51_{\pm 0.14}$ | $34.66_{\pm 0.33}$ |
| NPOS | $66.87_{\pm 4.79}$ | $85.70_{\pm 1.49}$ | $65.42_{\pm 1.36}$ | $61.70_{\pm 0.83}$ | $72.88_{\pm 0.95}$ | $70.51_{\pm 0.94}$ | $90.57_{\pm 1.82}$ | $31.92_{\pm 0.19}$ |
| PALM | $50.09_{\pm 1.27}$ | $56.17_{\pm 1.32}$ | $49.92_{\pm 1.69}$ | $55.06_{\pm 0.40}$ | $54.37_{\pm 0.77}$ | $53.12_{\pm 0.16}$ | $93.72_{\pm 0.22}$ | $40.88_{\pm 0.32}$ |
| **HamOS(ours)** | $\mathbf{72.78}_{\pm 4.98}$ | $\mathbf{92.22}_{\pm 6.42}$ | $\mathbf{71.50}_{\pm 4.84}$ | $\mathbf{73.13}_{\pm 2.99}$ | $\mathbf{83.87}_{\pm 4.01}$ | $\mathbf{78.70}_{\pm 0.56}$ | $86.49_{\pm 4.65}$ | $34.68_{\pm 3.91}$ |

(a) Different sampling algorithms    (b) Different distance metrics    (c) Different contrastive losses

Figure 6: **HamOS is a general framework compatible with different sampling algorithms, scoring functions, and ID contrastive losses:** (a) HamOS can consistently achieve superior performance with variants of HMC with Random Walk as the baseline;(b) HamOS outstrips the baselines with both Mahalanobis and KNN distance as scoring functions; (c) HamOS displays consistent enhancement equipped with different ID contrastive losses.

on post-hoc methods in Appendix F. HamOS demonstrates enhanced performance when integrated with diverse post-hoc approaches (e.g., attaining **25.55(%)** in FPR95 with Scale scoring function), shown in Table 11 in Appendix F; the results show that the enhancement is consistent with various post-hoc methods, suggesting that HamOS can improve the intrinsic capability of the pretrained model for OOD detection. HamOS maintains the ID classification efficacy as well at the same time.

**HamOS helps to learn distinguishable representations.** Next, we quantitatively analyze the hyperspherical embedding quality for ID-OOD separability and the differentiation of ID clusters. We leverage the cosine similarity to reflect the *ID-OOD separation*, *ID inter-class dispersion*, and *ID intra-class compactness*, defined in Eqn. (14) in Appendix C.4. The ID-OOD Separation and the inter-class Dispersion are better with higher values, while the intra-class compactness is better with smaller values. Table 3 presents the results in *angular degree* form for better readability. The results show that HamOS leads to higher ID-OOD separation, reflecting superior OOD detection performance compared to the baselines. HamOS displays an averaged **11.62%** improvement compared with NPOS. As for the inter-class dispersion and intra-class compactness, though HamOS doesn't achieve distinguishable values, it can achieve competitive ID classification accuracy by maintaining the original classification head, as reported in Table 1, indicating that the inter-class dispersion and intra-class compactness are not straightforward metrics for evaluating the ID-OOD differentiation. For example, CIDER outstrips SSD+ by a large margin on the inter-class dispersion (in Table 3) while attaining approximately equivalent ID accuracy as SSD+ (in Table 1).

## 4.3 FURTHER DISCUSSIONS AND ABLATION

**HamOS demonstrates versatility with various sampling algorithms.** We substitute the original HMC algorithm in HamOS with other variants to validate the proposed framework's versatility. Figure 6(a) shows that the proposed framework HamOS is consistently effective with various sampling algorithms compared with the Random Walk algorithm as a baseline, highlighting the OOD-ness density estimation as the pivotal component. The full averaged results are in Table 9, where we also report the time required to synthesize a mini-batch of outliers in milliseconds, reflecting the time efficiency of different HMC variants. Please refer to Appendix C.3 for details of variants of HMC.

**HamOS is robust to different scoring functions.** To further demonstrate the robustness of the proposed HamOS, we ablate with different scoring functions. First, we compare HamOS with regularization-based methods using KNN distance (Sun et al., 2022) and Mahalanobis distance (Lee

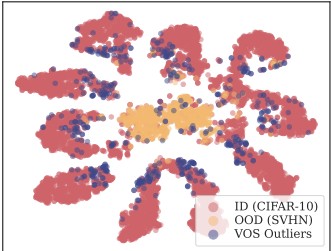 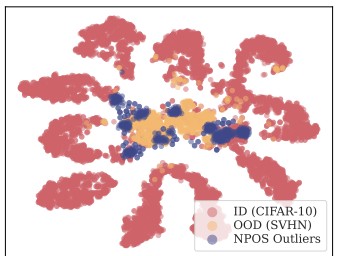 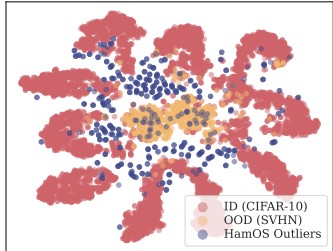

Figure 7: **t-SNE (van der Maaten & Hinton, 2008) Visualization:** (a)VOS; (b)NPOS; (c)HamOS.

et al., 2018b). The results on CIFAR-100 in Figure 6(b) show that our proposed HamOS achieves competitive performance with the two distance metrics. The full averaged results are in Table 10. Second, we evaluate HamOS with multiple scoring functions. Since HamOS retains the classification head, it is compatible with other post-hoc detection methods. We report the results when equipped with other scoring functions in Table 11 in Appendix F. The results demonstrate that HamOS enhances the intrinsic detection capability regardless of the scoring function employed.

**HamOS demonstrates consistent improvements using different ID contrastive losses.** Recall that our proposed HamOS synthesizes virtual outliers utilizing solely the ID feature embeddings, which are regularized with ID contrastive losses. We validate the efficacy of HamOS additionally using two advanced ID contrastive training losses, i.e., SimCLR (Chen et al., 2020; Winkens et al., 2020) and SupCon (Khosla et al., 2020), with Cross-entropy loss (Tao et al., 2023) as the baseline. In Figure 6(c), we report the FPR95 on CIFAR-10. It shows that our proposed HamOS can consistently enhance the detection capability compared to training singly with the ID contrastive loss. The full averaged results are in Table 12 in Appendix F with additional experiment details.

**HamOS displays exceptional efficiency.** Though the HMC typically incurs more computational cost than simpler Gaussian sampling, HamOS still keeps the time cost comparable to VOS and NPOS and even surpasses NPOS on ImageNet-1K, resulting from the high quality of sampling and highly efficient implementation. Refer to Appendix F.1 for detailed analysis.

**Ablation studies on loss weight $\lambda_d$, $k$ value for calculating OOD-ness density, hard margin $\delta$, Leapfrog steps $L$, step size $\epsilon$, number of nearest ID clusters $N_{\text{adj}}$, and synthesis rounds $R$.** We provide thorough ablation studies to elucidate the effect of various factors in our HamOS framework in Appendix F.2. The ablation demonstrates the robustness of HamOS for some hyperparameters and the tuning preference for others. Detailed implementation of HamOS is provided in Appendix D.

### 4.4 ADDITIONAL VISUALIZATIONS

We utilize t-SNE (van der Maaten & Hinton, 2008) to intuitively compare the synthesized outliers by VOS (Du et al., 2022b), NPOS (Tao et al., 2023), and our HamOS in Figure 7. The ID (i.e., CIFAR-10) and OOD (i.e., SVHN) features are obtained by the projection head with $\ell_2$ normalization. HamOS can synthesize outliers spanning a broad range of areas in the feature space between ID clusters, hence exposing the model to miscellaneous potential OOD scenarios. On the other hand, VOS might generate erroneous outliers that reside in ID clusters; NPOS might generate gathered outliers that lack diversity. We provide additional discussion with visualizations in Appendix G.

## 5 CONCLUSION

In this paper, we propose a novel framework, HamOS, for synthesizing virtual outliers via Hamiltonian Monte Carlo. Diverse and representative outliers are sampled from Markov chains which traverse a broad area in the hyperspherical space, facilitating representation learning by various OOD supervision signals between ID classes. HamOS adopts a Y-head architecture that optimizes CE loss for ID classification , and ID contrastive loss and OOD discernment loss for better ID-OOD separability on the unit sphere. Extensive empirical analysis demonstrates the superiority of HamOS against competitive baselines on both standard and large-scale benchmarks. We conduct sufficient ablation studies to reveal the mechanism of HamOS under various circumstances. We hope this paper provides an inspiring perspective for future research on synthesizing outliers for OOD detection.

ACKNOWLEDGMENTS

This work was supported by the National Science and Technology Major Project (2022ZD0114803), the Natural Science Foundation of Wuhan (2023010201020229), and the Major Program (JD) of Hubei Province (2023BAA024).

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

# Appendix

## Contents

## A    REPRODUCIBILITY STATEMENT

Some important aspects are summarized as follows to facilitate reproducing the results in this paper:

- **Datasets.** All datatsets adopted by this paper are publicly available, as is introduced in Section 4.1. For OOD detection evaluation, we ensure there is no overlap between OOD test data and ID training data. The data processing follows the common practice.

- **Assumption.** The main experiments are set under the fine-tuning scenario where a well-trained model on the original classification task is available, and the ID training dataset is also available for subsequent fine-tuning. We don't make any assumption of the existence of natural auxiliary outliers for fine-tuning, which is a more general and practical scheme.

- **Experiment environment.** All experiments in this paper are conducted for multiple runs on a single NVIDIA Tesla V100 Tensor Core with 32GB memory using Python version 3.10.9. The deep learning environment is established using PyTorch version 1.13.1 and Trochvision version 0.14.1 with CUDA 12.2 in the Ubuntu 18.04.6 system.

## B    RELATED WORK

### B.1    OOD DETECTION

**Post-hoc OOD Detection**    OOD detection is essential for constructing trustworthy machine learning systems in open-world scenarios (Nguyen et al., 2015). One straightforward line is to devise powerful scoring functions in a post-hoc fashion. As baseline methods, OpenMax (Bendale & Boult, 2016) introduces a new model layer for detecting unseen classes, and MSP (Hendrycks & Gimpel, 2017) leverages maximum softmax probability. The following works either focus on designing distinguishing scoring functions or trying to exploit feature-embeddings (Sun et al., 2021; Djurisic et al., 2023; Xu et al., 2024; Zhao et al., 2024; Liu & Qin, 2024), logits (Hendrycks et al., 2022; Wang et al., 2022), or probability outputs (Liang et al., 2018; Liu et al., 2020; Sun & Li, 2022). Additionally, Mahalanobis distance-based confidence score (Lee et al., 2018b) utilizes class conditional Gaussian distributions to differentiate OODs from IDs; GradNorm (Huang et al., 2021) utilizes information from gradient space; KNN-based score (Sun et al., 2022) calculates euclidean distance from ID priors; ASH (Djurisic et al., 2023) and Scale (Xu et al., 2024) explore the rectification of activation for better detection; Relation (Kim et al., 2023) inspects the structure of feature graph to detect OOD sample; NAC (Liu et al., 2024) models the neuron activation coverage for discrimination. Nonetheless, post-hoc methods only achieve suboptimal performance with fixed overconfident models compared with regularization-based methods.

**Regularized OOD Detection**    Another viable approach for OOD detection is to incorporate regularization techniques into the training process, with or without auxiliary OOD supervision signals. Contrastive training schemes (Winkens et al., 2020; Tack et al., 2020; Sehwag et al., 2021; Sun et al., 2022; Ming et al., 2023b; Lu et al., 2024) create distinguishable ID feature embeddings. LogitNorm (Wei et al., 2022) learns with a constant normalization; UM(Zhu et al., 2023a) inspects the middle training stage for better detection ability; ISH (Xu et al., 2024) integrates scaling technique into the training phase; Split-Ensemble (Chen et al., 2024a) splits the classification task into sub-ones within a single model. Natural OOD supervision forces the model to produce uniform distributions (Hendrycks et al., 2019; Zhang et al., 2023a) or higher EBO scores (Liu et al., 2020) for outliers. Recent works propose advanced sampling strategies for auxiliary OOD data, among which ATOM (Chen et al., 2021) greedily mines auxiliary OOD data, POEM (Ming et al., 2022) adopts Thompson sampling to explore and exploit the auxiliary OOD data, while DOS (Jiang et al., 2024a) selects informative outliers from clusters.

**Generative OOD Detection**    Without available natural OOD data, some studies propose to synthesize virtual outliers (Lee et al., 2018a; Moller et al., 2021; Pourreza et al., 2021; Du et al., 2023; Zheng et al., 2023) with generative networks (e.g., GAN(Goodfellow et al., 2020) or diffusion model(Ho et al., 2020)) or from feature space (Du et al., 2022a; Tao et al., 2023). There are methods that focus on OOD detection on energy-based model, such as Hat EBM (Hill et al., 2022) which incorporates the generator into its forward pass, thus enabling explicit modeling of the generator's outputs. DOE (Wang et al., 2023) transforms the original OOD samples into worse ones;

DivOE (Zhu et al., 2023b) leverages information extrapolation for synthesizing more informative outliers; TOE (Park et al., 2023) generates textual outliers for CLIP-based OOD detection; Tag-Fog (Chen et al., 2024b) utilizes Jigsaw technique to generate fake outliers; RODEO (Mirzaei et al., 2024) adversarially synthesizes hard outliers; NF-DT (Kamkari et al., 2024) generates outliers considering both low-likelihood and local intrinsic dimension. However, these works suffer from limitations for requiring real outliers as anchors or high computational costs with heavy-parameterized generative models. Synthesizing outliers based solely on ID data has been underexplored hitherto, with existing works primarily focusing on applying Gaussian noise to get new outliers. This study investigates sampling outliers from Markov chains via Hamiltonian Monte Carlo for diversified, high-quality virtual OOD samples without the assumptions of the availability of natural outliers. Extensive empirical analysis demonstrates the efficiency and efficacy of the proposed framework.

### B.2 Score-based Markov Chain Monte Carlo Sampling

Markov Chain Monte Carlo (Metropolis et al., 1953) employs a transition kernel to produce a sample that has the target distribution, including score-based Markov Chain Monte Carlo, which leverages the gradient to generate new points explicitly. Hamiltonian Monte Carlo (HMC) (Duane et al., 1987) is an efficient algorithm that samples from high dimensional space with a high acceptance ratio. The Metropolis-adjusted Langevin algorithm (MALA) (Besag, 1994) is a variant of HMC with the Leapfrog step as 1. HMC can be extended to be Riemann Manifold HMC (RMHMC) (Girolami & Calderhead, 2011) using a dynamic covariance matrix for sampling the momentum from Gaussian distribution. This paper utilizes the Spherical HMC (Lan et al., 2013) to synthesize outliers on the unit sphere and ablate on variants of HMC to demonstrate the generality of the proposed HamOS.

## C Additional Preliminaries

This section provides more preliminaries for our outlier synthesis framework.

### C.1 Training with Auxiliary OOD Data

To regularize the model for better detection capability, a research line (Hendrycks et al., 2019; Chen et al., 2021; Ming et al., 2022; Zhang et al., 2023a) devise a paradigm that utilizes an auxiliary OOD dataset during training, among which outlier exposure (OE) (Hendrycks et al., 2019) is a representative method. The learning objective can be generally formalized as follows:

$$\min_{\theta} \mathbb{E}_{\mathcal{P}^{\text{ID}}_{\mathcal{X}\mathcal{Y}}}[\mathcal{L}_{\text{CE}}(\boldsymbol{x}, y; \theta)] + \lambda \mathbb{E}_{\mathcal{P}^{\text{OOD}}_{\mathcal{X}}}[\mathcal{L}_{\text{OE}}(\hat{\boldsymbol{x}}; \theta)], \tag{9}$$

where $\lambda$ is the trade-off hyperparameter, $\mathcal{L}_{\text{CE}}$ is the Cross-entropy loss, and $\mathcal{L}_{\text{OE}}$ is the regularization with auxiliary outliers $\hat{\boldsymbol{x}}$. $\mathcal{L}_{\text{OE}}$ can be defined as Kullback-Leibler divergence between softmax outputs and the uniform distribution. Though OE is a powerful research line in OOD detection, heavy reliance on natural OOD data limits its application. Thus, generating virtual outliers is proposed to mitigate this challenge (Du et al., 2022b; Tao et al., 2023; Du et al., 2023). However, how to sample diverse and representative outliers distinctive from ID remains a key challenge.

### C.2 Modeling the ID Hyperspherical Distribution

Benefiting from the advanced representation learning (Ming et al., 2023b; Lu et al., 2024), we formulate the IDs in sampling space with the hyperspherical model which projects an ID input $\boldsymbol{x}$ as $\boldsymbol{z}$ onto the unit sphere ($\|\hat{\boldsymbol{z}}\|_2 = 1$) in a $d$-dimensional space. The projected embedding $\boldsymbol{z}$ can then be effectively modeled by the von Mises-Fisher (vMF) distribution (Mardia & Jupp, 2009; Fisher, 1953). The probability density kernel for a normalized vector $\boldsymbol{z}$ is formulated as:

$$p_c(\boldsymbol{z}; \boldsymbol{\mu}, \kappa) = Z_d(\kappa) \exp\left(\kappa \boldsymbol{\mu}^\top \boldsymbol{z}\right) = \frac{\kappa^{d/2-1}}{(2\pi)^{d/2} \boldsymbol{I}_{d/2-1}(\kappa)} \exp\left(\kappa \boldsymbol{\mu}^\top \boldsymbol{z}\right), \tag{10}$$

where $\boldsymbol{\mu}$ denotes the mean vector of the vMF kernel, $\kappa \geq 0$ represents the concentration parameter, $\boldsymbol{I}_\nu(\kappa)$ is the modified Bessel function of the first kind at order $\nu$, and $Z_d(\kappa)$ is the normalization factor. We approximate the class-conditional probability density function with kernel density estimation (KDE) (Rosenblatt, 1956) to produce a more precise estimation of the ID probability density

function. Specifically, let $\mathcal{Z}_c = \{\boldsymbol{z}_1, \boldsymbol{z}_2, \cdots, \boldsymbol{z}_n\}$ denotes the normalized embedding set of training data from class $c$. The class-conditional probability density function of $\boldsymbol{z}$ is then approximated as:

$$\hat{p}_c(\boldsymbol{z}; \mathcal{Z}_c, \kappa) = \frac{1}{n} \sum_{i=1}^{n} p_d(\boldsymbol{z}; \boldsymbol{z}_i, \kappa), \tag{11}$$

where $p_d(\boldsymbol{z}; \boldsymbol{z}_i^c, \kappa)$ is delineated as Eqn. (10), and $n$ is the number of samples from class $c$. The final ID probability density function is given by softmax as follows:

$$P_c^{\text{ID}}(\boldsymbol{z}) = \frac{\hat{p}_c(\boldsymbol{z}; \mathcal{Z}_c, \kappa)}{\sum_{i=1}^{C} \hat{p}_i(\boldsymbol{z}; \mathcal{Z}_i, \kappa)}. \tag{12}$$

The estimated probability density function with the vMF kernel offers accurate modeling of the unit sphere in such a high-dimension space, which is employed to prevent erroneous outlier generation.

### C.3  HAMILTONIAN MONTE CARLO

Given the target distribution density $P(\boldsymbol{z}) \propto \exp(-U(\boldsymbol{z}))$ where $U(\boldsymbol{z})$ is interpreted as the potential energy in position $\boldsymbol{z}$ and the momentum $\boldsymbol{q}$ sampled from the standard normal distribution $\mathcal{N}(\boldsymbol{0}, \boldsymbol{I})$, the Hamiltonian function is the total energy

$$H(\boldsymbol{z}, \boldsymbol{q}) = U(\boldsymbol{z}) + \frac{1}{2}\|\boldsymbol{q}\|_2^2,$$

and the Hamilton's Equation is given by

$$\frac{\mathrm{d}z_i}{\mathrm{d}t} = \frac{\partial H}{\partial q_i} = q_i, \quad \frac{\mathrm{d}q_i}{\mathrm{d}t} = -\frac{\partial H}{\partial z_i} = -\frac{\partial U}{\partial z_i},$$

where $t$ is the fictional time. To simulate the evolution, in each round, HMC takes the current $(\boldsymbol{z}, \boldsymbol{q})$ as initial state and performs $L$ steps Leapfrog discretization with step size $\epsilon$,

$$\boldsymbol{z}(0 + \epsilon/2) = \boldsymbol{z}(0) + \frac{\epsilon}{2}\frac{\mathrm{d}\boldsymbol{z}}{\mathrm{d}t}(0) = \boldsymbol{z}(0) + \frac{\epsilon}{2}\boldsymbol{q}(0),$$

$$\boldsymbol{q}(n\epsilon) = \boldsymbol{q}((n-1)\epsilon) + \epsilon\frac{\mathrm{d}\boldsymbol{q}}{\mathrm{d}t}((n-\nicefrac{1}{2})\epsilon) = \boldsymbol{q}((n-1)\epsilon) - \epsilon\frac{\mathrm{d}U}{\mathrm{d}\boldsymbol{z}}(\boldsymbol{z}(n-\nicefrac{1}{2})\epsilon), \ n = 1, \ldots, L$$

$$\boldsymbol{z}((n+\nicefrac{1}{2})\epsilon) = \boldsymbol{z}((n-\nicefrac{1}{2})\epsilon) + \epsilon\frac{\mathrm{d}\boldsymbol{z}}{\mathrm{d}t}(n\epsilon) = \boldsymbol{z}((n-\nicefrac{1}{2})\epsilon) + \epsilon\boldsymbol{q}(n\epsilon), \ n = 1, \ldots, L-1$$

$$\boldsymbol{z}(L\epsilon) = \boldsymbol{z}((L-\nicefrac{1}{2})\epsilon) + \frac{\epsilon}{2}\boldsymbol{q}(L\epsilon).$$

The final state $(\boldsymbol{z}(L\epsilon), \boldsymbol{q}(L\epsilon))$ is the proposal, which is accepted and set as the initial state of the next round with probability $\min\{1, \exp(H(\boldsymbol{z}(0), \boldsymbol{q}(0)) - H(\boldsymbol{z}(L\epsilon), \boldsymbol{q}(L\epsilon)))\}$. Since Hamilton's Equation implies the conservation of energy, i.e., $H(\boldsymbol{z}, \boldsymbol{q})$ is constant as evolution, the acceptance rate of the Metropolis test for HMC should be exactly 1. However, due to the small error caused by Leapfrog discretization, occasionally the value of $H(\boldsymbol{z}, \boldsymbol{q})$ may decrease which results in some rejection of proposals.

Additionally, we conduct the ablation study on the score-based Markov Chain Monte Carlo algorithms to demonstrate the generality of our HamOS framework. We utilize the random walk algorithm with Metropolis-Hastings acceptance step (Metropolis et al., 1953; Hastings, 1970) as the baseline. We also employ the Riemann Manifold Hamiltonian Monte Carlo (RMHMC) (Girolami & Calderhead, 2011), which is effective if the target distribution is concentrated on a low-dimensional manifold. To accelerate computation, we adopt the simplified version of RMHMC where the dynamic covariance matrix for sampling momentum $\boldsymbol{q}_i$ is calculated as:

$$\Sigma^* = \Sigma(\boldsymbol{z}_{i-1-J}, \boldsymbol{z}_{i-J}, \cdots, \boldsymbol{z}_{i-1}), \tag{13}$$

where $J$ (set to 2) is the number of previous states considered. When setting the Leapfrog step to 1, HMC reduces to the Metropolis-Adjusted Langevin Algorithm (MALA) (Besag, 1994), also known as Langevin Monte Carlo (LMC). Manifold MALA (dubbed as mMALA) (Girolami & Calderhead, 2011) can also be derived from RMHMC. Therefore, we experiment with the following MCMC algorithms for sampling outliers: Random Walk, HMC (default), MALA, mMALA, and RMHMC.

### C.4 HYPERSPHERICAL EVALUATION METRIC

We define the *ID-OOD separation*, *ID inter-class dispersion*, and *ID intra-class compactness* to analyze the quality of hyperspherical embeddings for representation learning methods, following CIDER (Ming et al., 2023b). The ID-OOD separation $\cos_{\text{Seperation}}$ measures the angular distance between ID and OOD; a larger value indicates that it is easier to distinguish OOD from ID. The inter-class dispersion $\cos_{\text{Dispersion}}$ reflects how sparse the IDs distribute on the unit sphere. The intra-class compactness $\cos_{\text{Compactness}}$ demonstrates the degree to which ID samples are clustered to its prototype. Straightforwardly, a model with better detection capability should produce embeddings with large $\cos_{\text{Seperation}}$ value. The three metrics are defined as follows respectively,

$$
\cos_{\text{Seperation}}(\mathcal{D}_{\text{test}}^{\text{OOD}}, \boldsymbol{\mu}) = \frac{1}{|\mathcal{D}_{\text{test}}^{\text{OOD}}|} \sum_{i=1}^{|\mathcal{D}_{\text{test}}^{\text{OOD}}|} \max_c \boldsymbol{z}_i^\top \boldsymbol{\mu}_c,
$$

$$
\cos_{\text{Dispersion}}(\boldsymbol{\mu}) = \frac{1}{C} \sum_{i=1}^{C} \frac{1}{C-1} \sum_{j=1}^{C} \boldsymbol{\mu}_i^\top \boldsymbol{\mu}_j \mathbf{1}\{i \neq j\},
$$

$$
\cos_{\text{Compactness}}(\mathcal{D}_{\text{test}}^{\text{ID}}, \boldsymbol{\mu}) = \frac{1}{|\mathcal{D}_{\text{test}}^{\text{ID}}|} \sum_{i=1}^{|\mathcal{D}_{\text{test}}^{\text{ID}}|} \sum_{j=1}^{C} \boldsymbol{z}_i^\top \boldsymbol{\mu}_j \mathbf{1}\{y_i = j\},
$$

(14)

where $\boldsymbol{\mu}_c$ is one of prototypes that have the largest cosine similarity with the OOD test sample, $\mathcal{D}_{\text{test}}^{\text{OOD}}$ is the OOD test data, $\mathcal{D}_{\text{test}}^{\text{ID}}$ is the ID test data, and $\mathbf{1}\{\cdot\}$ is the indicator function.

## D DETAILED IMPLEMENTATION OF HAMOS

### D.1 REALIZATION OF HAMOS

In this section, we provide the detailed implementation of our proposed HamOS. The outlier synthesis process is described in Algorithm 1, and the training pipeline of HamOS is described in Algorithm 2. The model is assumed to be well-trained on the ID data.

Our primary goal is to synthesize diverse and representative outliers and integrate the synthesized virtual outliers into the training phase with ID training data to regularize the model for better ID-OOD differentiation. For outlier synthesis, we utilize the Hamiltonian Monte Carlo with our OOD-ness density estimation in Eqn. (2) to sample sequences of outliers that span a wide range of OOD-ness. For regularization training, we optimize Cross-entropy loss and ID contrastive loss with ID data and OOD discernment loss with synthesized OOD data.

For outlier synthesis, we maintain a class-conditional dynamic buffer of ID feature embeddings $\{\mathcal{Z}_i\}_{i=1}^{C}$ for calculating OOD-ness density. Specifically, we update the fixed-size array of ID classes when taking a new batch of ID data, eliminating the oldest ones. The $k$-th nearest neighbor embeddings in ID clusters are then retrieved to calculate OOD-ness density. The initial points of Markov chains are obtained by the centroids of ID cluster pairs, i.e., the normalized mean vector given two ID clusters. The initial points are treated as a mini-batch, each deriving a Markov chain parallelly through matrix operations. The number of synthesis rounds is fixed (e.g., default to be 5). For rejecting erroneous outliers, the ID embedding buffer is utilized to estimate the ID probability for a candidate, as in Eqn. (5). The accepted candidates are treated as positions for the next round of synthesis, while the rejected candidates are discarded with their preceding points as the positions of the next round of synthesis. The samples in each Markov chain are gathered to form a mini-batch of outliers for each training iteration.

For regularization training, we follow the common practice of contrastive training (Tack et al., 2020; Chen et al., 2020; Ming et al., 2023b; Lu et al., 2024) to apply dual augmentation to get two perspectives of ID data. The augmented ID data is then used to update the ID embedding buffer and calculate the CE loss. The ID embeddings are also used to update the ID prototypes in the EMA manner, which are used to calculate the ID contrastive with the ID embeddings as in Eqn. (22) or the OOD-discernment loss with the virtual outliers as in Eqn. (7). The loss is calculated as Eqn. (8).

---

**Algorithm 2** Training Pipeline of HamOS

---

**Input:** model: $\{f(\cdot), \mathrm{MLP}(\cdot); \theta\}$, fine-tuning epochs: $T$, ID training data: $\mathcal{D}_{\mathrm{train}}^{\mathrm{ID}}$, learning rate: $\eta$, OOD-discernment loss coefficient: $\lambda_d$;
**Output:** fine-tuned model $\theta^{(T)}$;

1: **for** epoch $= 1, \cdots, T$ **do**
2:     **for** ID data mini-batch $B_i^{\mathrm{ID}}$ from $\mathcal{D}_{\mathrm{train}}^{\mathrm{ID}}$ **do**
3:         Conduct dual augmentation for ID sample $(\boldsymbol{x}_j, y_j) \sim B_i^{\mathrm{ID}}$ to get $(\tilde{\boldsymbol{x}}_j \oplus \tilde{\boldsymbol{x}}_j', y \oplus y)$.
4:         Project the ID data to hyperspherical space $\boldsymbol{z}_j = \mathrm{norm}(\mathrm{MLP}(f(\tilde{\boldsymbol{x}}_j \oplus \tilde{\boldsymbol{x}}_j')))$.
5:         Update ID embedding buffer with $\boldsymbol{z}_j$.
6:         Calculate Cross-Entropy loss $\mathcal{L}_{\mathrm{CE}}$ with $f(\tilde{\boldsymbol{x}}_j \oplus \tilde{\boldsymbol{x}}_j')$ and $y \oplus y$.
7:         Calculate the ID contrastive loss $\mathcal{L}_{\mathrm{ID\text{-}con}}$ with $\boldsymbol{z}_j$ (and $y \oplus y$).
8:         Sample a mini-batch of virtual outliers $\mathcal{Z}^{\mathrm{OOD}}$ as in Algorithm 1.
9:         Calculate the OOD discernment loss $\mathcal{L}_{\mathrm{OOD\text{-}disc}}$.
10:        Calculate the final loss $\mathcal{L}_{\mathrm{HamOS}} = \mathcal{L}_{\mathrm{CE}} + \mathcal{L}_{\mathrm{ID\text{-}con}} + \lambda_d \mathcal{L}_{\mathrm{OOD\text{-}disc}}$.
11:        Update the model weights: $\theta \leftarrow \theta - \eta \nabla_\theta \mathcal{L}_{\mathrm{HamOS}}$.
12:     **end for**
13: **end for**

---

Table 4: Training Configurations of HamOS.

| Factor | CIFAR-10 | CIFAR-100 | ImageNet-1K |
|---|---|---|---|
| Training epochs | 20 | 20 | 20 |
| Trainable part | whole | whole | layer-4+MLP |
| Learning rate | 0.01 | 0.01 | 0.0001 |
| Momentum | 0.9 | 0.9 | 0.9 |
| Batch size | 128 | 128 | 256 |
| Feature dimension | 128 | 128 | 256 |
| Weight decay | $1.0 \times 10^{-4}$ | $1.0 \times 10^{-4}$ | $1.0 \times 10^{-4}$ |
| LR schedule | Cos. Anneal | Cos. Anneal | Cos. Anneal |
| Prototype update factor | 0.95 | 0.95 | 0.95 |
| Buffer size of ID data | 1000 | 1000 | 100 |
| Bandwidth $\kappa$ | 2.0 | 2.0 | 2.0 |
| OOD-discernment weight $\lambda_d$ | 0.1 | 0.1 | 0.1 |
| $k$ for KNN distance | 200 | 200 | 100 |
| Hard margin $\delta$ | 0.1 | 0.1 | 0.1 |
| Leapfrog step $L$ | 3 | 3 | 3 |
| Step size $\epsilon$ | 0.1 | 0.1 | 0.1 |
| Number of adjacent ID clusters $N_{\mathrm{adj}}$ | 4 | 4 | 4 |
| Synthesis rounds $R$ | 5 | 5 | 5 |

## D.2 DETAILED EXPERIMENT SETUPS

For fine-tuning a pretrained model, we set the training epochs to 20 following previous works (Ming et al., 2023b; Tao et al., 2023), with the size of mini-batch set to 128 for CIFAR-10/100 and 256 for ImageNet-1K. For ImageNet-1K, we freeze the first three layers of the pretrained model following Ming et al. (2023b) and use the cosine annealing as the learning rate schedule beginning at 0.0001. For memory efficiency, we set the size of the class-conditional ID buffer to 1000 for CIFAR-10/100 and 100 for ImageNet-1K. We summarize the default training configurations of HamOS in Table 4. For experiments that trains the model from scratch, the setups are detailed in Appendix F.1.

## E BASELINES AND HYPERPARAMETERS

In this section, we briefly introduce the baselines adopted compared with HamOS, as well as their specific hyperparameters adopted by this paper.

### E.1 POST-HOC BASELINES

**Maximum Softmax Probability (MSP)** (Hendrycks & Gimpel, 2017). MSP leverages the maximum softmax probability to distinguish OOD input from ID samples. The scoring function is

defined as follows,

$$S_{\text{MSP}}(\boldsymbol{x}; \theta) = \max_c P(y = c | \boldsymbol{x}; \theta) = \max \text{softmax}(F(\boldsymbol{x}; \theta)), \tag{15}$$

where $\theta$ is the weights of the well-trained model and $c$ is one of the classes. A larger MSP score indicates a sample is more likely to be ID data, reflecting the model's confidence in the sample.

**ODIN** (Liang et al., 2018). ODIN proposes to scale the softmax outputs with a temperature factor $\tau$ and conduct tiny perturbations to the input data. The ODIN scoring function is defined as follows,

$$S_{\text{ODIN}}(\tilde{\boldsymbol{x}}; \theta) = \max_c P(y = c | \tilde{\boldsymbol{x}}; \theta) = \max \text{softmax}(F(\tilde{\boldsymbol{x}}; \theta)/\tau), \tag{16}$$

where $\tau$ is the temperature factor, and $\tilde{\boldsymbol{x}}$ is the perturbed input samples controled by $\epsilon$. The $\tau$ and $\epsilon$ is set to $1.0 \times 10^4$ and $1.4 \times 10^{-1}$ defaultly as suggested in Liang et al. (2018).

**EBO** (Liu et al., 2020). The EBO scoring function measures the information contained by the logits of the model, which is adopted to discriminate between ID and OOD samples. The EBO scoring function is defined as follows,

$$S_{\text{EBO}}(\boldsymbol{x}; \theta) = -\tau \log \sum_{c=1}^{C} \exp\left(F(\boldsymbol{x}; \theta)_c / \tau\right), \tag{17}$$

where $\tau$ is the temperature factor which is set to 1.0 defaultly in this paper. OOD samples tend to have higher EBO scores, and ID samples typically have lower scores.

**K-Nearest Neighbor (KNN)** (Sun et al., 2022). This scoring function finds the $k$-th nearest neighbor for an input sample given the training dataset embeddings. The KNN scoring function is defined as follows,

$$S_{\text{KNN}}(\boldsymbol{x}; f) = \|\boldsymbol{x} - \boldsymbol{x}_{(k)}\|_2, \tag{18}$$

where $f$ is the well-trained feature extractor that maps an input to the feature space, $\|\cdot\|_2$ is L-2 norm, and $\boldsymbol{x}_{(k)}$ is the $k$-th nearest neighbor in the feature space. The $k$ is set to 50 default in this paper.

**ASH** (Djurisic et al., 2023). ASH is an efficient post-hoc method that prunes and scales the model's activation. The modified representation is then put forward to the rest of the modules for ID classification or OOD detection. In this paper, we use the ASH-S version and set the prune percentile to 65% as in Djurisic et al. (2023) and use the EBO scoring function for detection.

**Scale** (Xu et al., 2024). Scale is a subsequent work following ASH, which conducts solely the scaling of the activation based on the mechanism analysis of ASH. Scale can further increase the EBO score separation between ID and OOD data. We adopt the 85% prune rate as suggested in Xu et al. (2024) and use the EBO scoring function for Scale.

**Relation** (Kim et al., 2023). Relation utilizes the relational structure graph and identifies the OOD data via the similarity kernel values. Specifically, the Relation scoring function is defined as follows,

$$S_{\text{Relation}}(\boldsymbol{x}; \theta) = \frac{1}{\sum_{i \in S} k(\boldsymbol{x}, \boldsymbol{x}'_i)}, \tag{19}$$

where $k$ is the similarity kernel function defined as in Kim et al. (2023), and $\boldsymbol{x}'_i$ is the training data.

### E.2 REGULARIZATION-BASED BASELINES

**CSI** (Tack et al., 2020). CSI initiates a new contrastive training scheme for OOD detection, which contrasts the sample with a distributionally-shifted augmentation of itself and trains the model with the contrastive training objective. The CSI scoring function is defined as,

$$S_{\text{CSI}}(\boldsymbol{x}; f) = \max_m \cos\left(f(\boldsymbol{x}_m), f(\boldsymbol{x})\right) \cdot \|f(\boldsymbol{x})\|, \tag{20}$$

where $\cos$ is the cosine similarity, and $\boldsymbol{x}_m$ is a sample from the training dataset with the maximum cosine similarity with the input data.

**SSD+** (Sehwag et al., 2021). SSD+ utilizes the supervised contrastive learning objective (i.e., Sup-Con (Khosla et al., 2020)) to get proper representation and utilizes the Mahalanobis distance-based

scoring function (Lee et al., 2018b) for detecting OOD data in feature space. The Mahalanobis distance scoring function is defined as,

$$S_{\text{Mahalanobis}}(\boldsymbol{x}; f) = \max_{c} \left( -(f(\boldsymbol{x}) - \boldsymbol{\mu}_c)^\top \hat{\Sigma}^{-1} (f(\boldsymbol{x} - \boldsymbol{\mu}_c)) \right), \tag{21}$$

where $\boldsymbol{\mu}_c$ is the mean of multivariate Gaussian distribution of class $c$ and $\hat{\Sigma}$ is the tied covariance of the $C$ class-conditional Gaussian distributions.

**KNN+** (Sun et al., 2022). KNN+ typically adopts the same training pipeline as SSD+ and leverages the KNN scoring function for detecting the OOD samples, defined as Eqn. (18).

**VOS** (Du et al., 2022b). VOS opens a promising direction that trains the model for OOD detection with virtual outliers. It models the feature space by mixtures of Gaussian distribution and synthesizes virtual outliers using class-conditional information.

**CIDER** (Ming et al., 2023b). CIDER proposes to project the feature embeddings onto the hyperspherical space and optimize the dispersion loss (termed as $\mathcal{L}_{\text{disp}}$) and compactness loss (termed as $\mathcal{L}_{\text{comp}}$) for proper ID representation. The losses are defined respectively as follows,

$$
\begin{aligned}
\mathcal{L}_{\text{disp}} &= \frac{1}{C} \sum_{i=1}^{C} \log \frac{1}{C-1} \sum_{j=1}^{C} \mathbf{1}\{j \neq i\} \exp\left(\boldsymbol{\mu}_i^\top \boldsymbol{\mu}_j / \tau\right), \\
\mathcal{L}_{\text{comp}} &= -\frac{1}{N} \sum_{i=1}^{N} \log \frac{\exp\left(\boldsymbol{z}_i^\top \boldsymbol{\mu}_{c(i)} / \tau\right)}{\sum_{j=1}^{C} \exp\left(\hat{\boldsymbol{z}}_i^\top \boldsymbol{\mu}_j / \tau\right)},
\end{aligned}
\tag{22}
$$

where $\boldsymbol{\mu}_c$ is the prototype of class $c$ and $\tau$ is the temperature factor. The final loss objective of CIDER is formulated as $\mathcal{L}_{\text{CIDER}} = \mathcal{L}_{\text{disp}} + \lambda_c \mathcal{L}_{\text{comp}}$, where $\lambda_c$ is set to 0.5 defaultly.

**NPOS** (Tao et al., 2023). NPOS proposes a non-parametric framework for synthesizing virtual outliers by selecting the far ID samples as anchors for sampling outliers from Gaussian distribution. The synthesized outliers are then used to optimize the binary training objective. The hyperparameters are set as suggested in Tao et al. (2023).

**PALM** (Lu et al., 2024). PALM is a subsequent method following CIDER, which proposes to learn with a mixture of prototypes for each class instead of forcing the ID data to be close to only one prototype. The training configuration is set as in Lu et al. (2024).

## F  ADDITIONAL EXPERIMENT RESULTS

In this section, we offer additional experiment results further to validate the efficacy and efficiency of the proposed HamOS.

### F.1  EFFIACY ANALYSIS OF HAMOS

**Full comparison results on CIFAR-10 and CIFAR-100.** We provide the main OOD performance comparison results in Table 5. As the results show, our proposed HamOS can outstrip the baselines on most OOD test datasets and achieve significant averaged improvement, demonstrating the effectiveness of HamOS on multiple OOD scenarios.

**Additional results on hard OOD benchmarks.** To further validate the efficacy of HamOS, we provide additional performance evaluation on the hard OOD benchmark (Zhang et al., 2023b). In Table 6, we treat CIFAR-100 and Tiny-ImageNet (Le & Yang, 2015) as hard OOD test datasets for CIFAR-10. In Table 7, we treat CIFAR-10 and Tiny-ImageNet as hard OOD test datasets for CIFAR-100. The hard OOD benchmark provides a more OOD scenario where the ID and OOD datasets have similar semantics. The results display that our proposed HamOS can outstrip all the baselines with CIFAR-10 as the ID dataset, while none of the methods achieve absolute superiority with CIFAR-100 as the ID dataset, indicating the difficulty of this benchmark.

**ID-OOD separation with CIFAR-100 as ID dataset.** We provide the compactness and separation analysis with CIFAR-100 as the ID dataset to quantitatively analyze the quality of learned representations. Shown as in Table 8, HamOS can help the model to produce representations with larger

Table 5: Full results of the main comparison (%). Comparison with competitive OOD detection baselines. Results are averaged across multiple runs.

| $\mathcal{D}_{ID}$ | Methods | FPR95↓ | AUROC↑ | AUPR↑ | FPR95↓ | AUROC↑ | AUPR↑ | FPR95↓ | AUROC↑ | AUPR↑ |
|---|---|---|---|---|---|---|---|---|---|---|
| | | MNIST | | | SVHN | | | LSUN | | |
| | | *Post-hoc Methods* | | | | | | | | |
| | MSP | 25.20±10.28 | 92.58±1.82 | 74.84±14.48 | 35.52±6.07 | 89.23±0.37 | 77.56±5.49 | 16.48±2.11 | 94.43±0.68 | 94.67±1.61 |
| | ODIN | 32.34±31.16 | 93.96±5.38 | 73.49±25.36 | 80.76±12.32 | 73.99±5.68 | 47.86±11.45 | 27.33±24.73 | 96.08±2.78 | 93.89±5.66 |
| | EBO | 27.17±20.01 | 94.72±2.81 | 75.90±18.58 | 54.76±15.16 | 88.18±1.42 | 70.61±8.04 | 12.11±3.87 | 97.10±1.12 | 96.24±2.34 |
| | KNN | 20.53±0.54 | 93.82±0.29 | 82.92±1.57 | 23.60±2.66 | 91.28±0.69 | 87.62±1.40 | 18.66±1.53 | 94.44±0.38 | 95.22±0.37 |
| | ASH | 38.21±38.06 | 90.80±8.35 | 68.47±31.77 | 59.57±25.61 | 84.51±6.93 | 65.12±18.47 | 30.17±31.01 | 94.85±4.21 | 92.41±7.97 |
| | Scale | 47.69±35.01 | 79.85±21.40 | 56.31±32.31 | 71.25±20.14 | 73.48±16.29 | 53.97±21.68 | 39.03±40.24 | 84.95±17.54 | 81.71±21.92 |
| | Relation | 23.09±0.45 | 93.27±0.49 | 79.93±1.75 | 26.30±4.89 | 90.86±1.06 | 86.44±2.34 | 21.64±1.91 | 94.25±0.35 | 94.66±0.43 |
| | | *Regularization-based Methods* | | | | | | | | |
| | CSI | 21.50±2.48 | 93.27±1.45 | 83.47±2.31 | 15.76±1.52 | 95.67±0.87 | 92.88±1.00 | 10.82±1.84 | 97.34±0.54 | 97.67±0.41 |
| | SSD+ | 23.65±4.21 | 92.48±1.86 | 81.31±3.19 | 12.83±1.95 | 96.98±0.64 | 94.44±0.90 | 10.01±1.60 | 97.46±0.58 | 97.77±0.45 |
| | KNN+ | 24.59±2.34 | 92.54±1.05 | 80.51±1.30 | 14.32±1.91 | 96.43±0.41 | 93.64±0.79 | 13.23±2.73 | 96.05±1.30 | 96.71±1.00 |
| | VOS | 26.47±17.30 | 94.64±2.56 | 76.24±17.33 | 46.11±33.13 | 89.27±6.34 | 73.44±19.05 | 18.86±13.38 | 96.25±2.30 | 94.36±5.07 |
| | CIDER | 18.91±0.23 | 95.44±0.53 | 85.22±0.47 | 8.47±0.13 | 98.37±0.05 | 96.27±0.02 | 10.81±3.17 | 97.08±1.09 | 97.47±0.83 |
| | NPOS | 16.61±4.60 | 96.32±1.50 | 87.71±3.49 | **3.12±1.03** | **99.17±0.19** | 97.80±0.59 | 7.04±1.97 | 98.41±0.27 | 98.53±0.28 |
| | PALM | 37.61±6.15 | 84.98±1.26 | 68.60±5.43 | 34.81±4.40 | 90.56±2.41 | 83.28±3.02 | 10.48±6.00 | 97.29±1.71 | 97.58±1.50 |
| | **HamOS(ours)** | **10.63±0.91** | **97.14±0.99** | **91.37±0.20** | 3.67±0.39 | 99.01±0.95 | **98.13±0.59** | **4.03±0.43** | **98.42±0.34** | **98.83±0.43** |
| | | Texture | | | Places | | | Averaged | | |
| | | *Post-hoc Methods* | | | | | | | | |
| | MSP | 39.08±6.35 | 90.04±0.41 | 92.29±1.61 | 44.56±9.16 | 89.21±0.84 | 69.13±6.49 | 32.17±6.38 | 91.10±0.71 | 81.70±5.82 |
| | ODIN | 78.70±12.77 | 81.64±4.93 | 82.64±6.75 | 71.08±15.46 | 82.85±5.60 | 52.50±13.77 | 58.04±18.46 | 85.70±4.17 | 70.08±11.84 |
| | EBO | 59.81±17.70 | 89.17±1.48 | 89.90±3.35 | 55.41±15.25 | 89.79±1.77 | 65.83±9.25 | 41.85±13.78 | 91.79±1.54 | 79.70±8.10 |
| | KNN | 22.83±2.49 | 93.29±0.85 | 96.13±0.60 | 28.70±2.34 | 92.05±0.45 | 81.81±1.96 | 22.86±1.12 | 92.98±0.42 | 88.74±0.79 |
| | ASH | 68.62±20.82 | 85.06±6.27 | 85.28±8.55 | 74.52±7.46 | 81.65±7.46 | 50.38±15.82 | 54.22±26.06 | 87.37±6.60 | 72.33±16.40 |
| | Scale | 79.80±13.73 | 74.37±16.16 | 78.24±13.51 | 78.13±11.76 | 76.04±10.15 | 44.89±14.08 | 63.18±23.64 | 77.74±16.24 | 63.03±20.52 |
| | Relation | 25.62±3.82 | 92.46±1.03 | 95.50±0.78 | 34.73±3.99 | 90.74±0.62 | 77.22±2.90 | 26.28±1.63 | 92.31±0.43 | 86.75±0.98 |
| | | *Regularization-based Methods* | | | | | | | | |
| | CSI | 23.96±2.43 | 92.64±0.80 | 95.89±0.45 | 34.02±3.66 | 89.72±1.16 | 78.80±2.49 | 21.21±1.68 | 93.73±0.33 | 89.74±0.68 |
| | SSD+ | 20.14±1.56 | 94.40±0.47 | 96.80±0.29 | 25.80±1.03 | 92.94±0.41 | 84.06±0.47 | 18.49±1.20 | 94.85±0.57 | 90.88±0.83 |
| | KNN+ | 21.14±1.80 | 93.78±0.51 | 96.43±0.48 | 25.12±1.42 | 93.27±0.32 | 84.98±0.51 | 19.68±1.86 | 94.41±0.66 | 90.46±0.66 |
| | VOS | 60.94±23.98 | 88.19±3.12 | 88.71±5.31 | 59.46±18.79 | 88.75±2.74 | 63.06±11.57 | 42.37±21.13 | 91.42±3.38 | 79.16±11.62 |
| | CIDER | 19.67±1.24 | 93.89±0.63 | 96.62±0.34 | 23.52±1.39 | 94.01±0.16 | 86.24±0.88 | 16.28±0.68 | 95.76±0.37 | 92.36±0.06 |
| | NPOS | 20.41±1.39 | 95.06±0.44 | 97.09±0.25 | 24.78±0.66 | 94.06±0.03 | 85.61±0.67 | 14.39±0.87 | 96.61±0.26 | 93.35±0.74 |
| | PALM | 45.04±9.50 | 88.19±3.17 | 92.31±2.36 | 33.31±3.89 | 91.67±1.19 | 80.42±2.67 | 32.25±4.14 | 90.54±1.46 | 84.44±2.14 |
| CIFAR-10 | **HamOS(ours)** | **15.89±0.52** | **95.82±0.22** | **97.64±0.82** | **18.19±0.76** | **95.18±0.08** | **88.73±1.04** | **10.48±0.76** | **97.11±0.26** | **94.94±0.86** |

| $\mathcal{D}_{ID}$ | Methods | FPR95↓ | AUROC↑ | AUPR↑ | FPR95↓ | AUROC↑ | AUPR↑ | FPR95↓ | AUROC↑ | AUPR↑ |
|---|---|---|---|---|---|---|---|---|---|---|
| | | MNIST | | | SVHN | | | LSUN | | |
| | | *Post-hoc Methods* | | | | | | | | |
| | MSP | 58.77±3.70 | 75.34±2.23 | 47.81±3.75 | 62.47±6.75 | 76.56±3.58 | 61.68±5.57 | 56.16±1.77 | 79.62±0.71 | 81.32±0.82 |
| | ODIN | 55.58±2.56 | **81.47±2.80** | 52.13±2.84 | 70.97±9.72 | 71.42±5.91 | 54.36±8.84 | 51.70±4.14 | 82.51±1.51 | 83.47±1.62 |
| | EBO | 61.62±2.58 | 75.80±2.95 | 44.78±2.19 | 59.02±9.46 | 79.83±3.91 | 65.72±6.93 | 54.51±2.94 | 80.63±1.45 | 82.26±1.41 |
| | KNN | 50.60±4.65 | 80.86±1.23 | **58.09±4.00** | 58.25±9.19 | 81.25±4.26 | 67.22±7.16 | 60.29±2.85 | 81.33±0.43 | 80.33±0.90 |
| | ASH | 76.04±3.15 | 71.74±2.37 | 30.98±1.11 | 59.79±10.15 | 80.61±4.54 | 65.68±8.15 | 56.07±3.52 | 79.71±0.57 | 80.70±0.85 |
| | Scale | 73.30±5.01 | 72.41±2.14 | 33.11±3.36 | 62.41±8.49 | 80.73±3.93 | 64.33±6.87 | 60.91±5.24 | 80.47±1.37 | 80.01±1.87 |
| | Relation | 51.47±3.90 | 79.13±1.36 | 57.79±3.22 | 61.11±8.70 | 79.82±4.10 | 65.30±6.90 | 66.49±3.15 | 79.88±0.65 | 77.30±1.03 |
| | | *Regularization-based Methods* | | | | | | | | |
| | CSI | 70.12±2.31 | 68.01±1.49 | 38.24±2.81 | 59.36±1.64 | 80.66±1.15 | 66.41±1.50 | 56.71±0.31 | 83.86±0.08 | 83.42±0.12 |
| | SSD+ | 77.28±2.60 | 66.39±1.27 | 31.03±3.18 | 36.77±0.88 | 89.28±0.31 | 81.60±0.51 | 50.35±4.02 | 82.22±0.68 | 83.47±1.30 |
| | KNN+ | 76.74±2.11 | 66.08±2.62 | 31.79±2.87 | 37.12±2.17 | 90.71±0.77 | 82.13±1.16 | 61.32±2.53 | 77.81±2.25 | 78.58±1.70 |
| | VOS | 54.94±1.06 | 80.91±0.99 | 52.08±1.87 | 67.54±7.49 | 78.59±1.80 | 60.97±5.67 | 32.52±3.66 | **92.00±1.47** | 92.14±1.42 |
| | CIDER | 44.31±2.70 | 68.05±2.69 | 40.41±2.81 | **30.11±0.82** | **92.83±0.43** | **86.28±0.59** | 45.23±3.19 | 83.55±1.00 | 85.67±1.06 |
| | NPOS | 62.27±5.75 | 70.10±2.91 | 47.28±5.46 | 31.67±1.38 | 91.88±0.71 | 85.07±1.04 | 54.29±3.52 | 80.10±1.08 | 81.71±1.51 |
| | PALM | 58.45±9.65 | 67.50±8.29 | 49.01±10.16 | 59.84±3.76 | 82.18±1.58 | 66.72±2.95 | **24.35±6.18** | 91.78±3.07 | **93.19±2.33** |
| | **HamOS(ours)** | 60.67±1.71 | 75.79±0.44 | 49.35±1.23 | 30.38±0.12 | 92.68±0.71 | 86.12±1.48 | 39.90±1.23 | 85.63±0.30 | 87.86±0.82 |
| | | Texture | | | Places | | | Averaged | | |
| | | *Post-hoc Methods* | | | | | | | | |
| | MSP | 62.93±3.95 | 76.25±1.48 | 84.69±1.21 | 58.54±0.80 | 78.47±0.36 | 58.82±0.74 | 59.78±2.16 | 77.25±1.28 | 66.86±1.58 |
| | ODIN | 70.51±2.36 | 77.68±1.08 | 84.08±1.03 | 68.67±2.05 | 76.99±0.83 | 51.98±2.11 | 63.49±2.51 | 78.01±1.62 | 65.20±2.19 |
| | EBO | 66.77±2.48 | 76.91±1.22 | 84.41±0.87 | 62.39±1.64 | 78.42±0.61 | 56.48±1.47 | 60.86±1.87 | 78.32±1.31 | 66.73±1.35 |
| | KNN | 55.25±1.40 | 82.46±0.32 | 88.95±0.27 | 60.41±1.03 | 79.13±0.18 | 58.43±0.74 | 56.96±2.96 | 81.01±1.19 | 70.60±2.29 |
| | ASH | 71.10±1.64 | 77.44±0.52 | 83.93±0.43 | 70.79±3.67 | 76.19±1.12 | 49.89±3.17 | 66.84±0.87 | 77.14±1.12 | 62.24±0.73 |
| | Scale | 75.20±1.42 | 76.56±0.62 | 82.62±0.56 | 74.53±3.77 | 76.05±1.51 | 47.05±3.38 | 67.29±2.31 | 77.25±1.01 | 61.42±1.42 |
| | Relation | 55.69±1.24 | 80.74±0.65 | 88.17±0.38 | 63.44±0.69 | 78.88±0.24 | 55.23±0.45 | 59.64±2.48 | 79.69±1.08 | 68.76±1.78 |
| | | *Regularization-based Methods* | | | | | | | | |
| | CSI | 88.34±3.90 | 65.07±1.85 | 72.76±2.74 | 72.14±0.70 | 69.71±0.32 | 47.05±0.77 | 69.34±0.86 | 73.46±0.30 | 61.57±0.75 |
| | SSD+ | 50.31±1.90 | 83.62±0.75 | 89.69±0.52 | 55.40±0.73 | **81.67±0.29** | 62.86±0.46 | 54.03±1.92 | 80.64±0.60 | 69.73±1.09 |
| | KNN+ | 65.34±0.55 | 79.80±0.36 | 85.84±0.11 | 65.71±0.74 | 76.79±0.10 | 54.84±0.39 | 61.25±0.81 | 78.24±0.93 | 66.64±0.88 |
| | VOS | 67.70±4.57 | 78.97±1.98 | 85.24±1.72 | 70.05±0.68 | 76.54±0.22 | 51.21±0.55 | 58.55±1.53 | 81.40±0.62 | 68.33±1.61 |
| | CIDER | **48.43±1.44** | 83.54±0.65 | 90.07±0.47 | 54.97±1.60 | 80.90±0.45 | 63.65±1.08 | 44.61±1.80 | 81.77±0.95 | 73.22±1.12 |
| | NPOS | 48.82±0.43 | **83.85±0.17** | **90.16±0.18** | 59.99±0.96 | 79.19±0.57 | 58.21±1.14 | 51.41±1.88 | 81.02±0.98 | 72.49±1.54 |
| | PALM | 70.34±2.33 | 80.44±2.02 | 85.31±1.30 | 62.66±0.93 | 77.85±0.37 | 56.80±0.55 | 55.13±0.97 | 79.95±1.26 | 70.21±1.38 |
| CIFAR-100 | **HamOS(ours)** | 49.21±1.10 | 82.70±0.33 | 89.62±0.13 | **53.23±1.83** | 81.41±0.27 | **64.66±0.10** | **46.68±1.44** | **83.64±0.64** | **75.52±1.30** |

ID-OOD separation values, significantly benefiting the ID-OOD differentiation. The inter-class dispersion and intra-class compactness don't directly correlate with the ID-ACC or the OOD performance. Although HamOS doesn't exhibit the highest inter-class dispersion or the lowest intra-class compactness, it indeed surpasses all the methods compared in Table 3, 8 by both OOD metrics and ID-ACC.

**Sampling with variants of HMC.** We provide the full averaged results of HamOS with variants of HMC algorithms in Table 9. We treat the Random Walk with Metropolis-Hastings as the baseline. The results show that our proposed outlier synthesis framework is compatible with various sampling algorithms. More advanced sampling algorithms can be adopted for specific scenarios.

**Using different distance metrics as scoring functions.** We offer the full averaged results of using Mahalanobis (i.e., defined as Eqn. (21)) and KNN (i.e., defined as Eqn. (18)) distance metrics in Ta-

Table 6: Results on hard OOD benchmarks with CIFAR-10 as ID data(%). Comparison with competitive OOD detection baselines. (averaged by multiple runs)

| Methods | CIFAR-100 | | | TinyImageNet | | | Averaged | | |
|---|---|---|---|---|---|---|---|---|---|
| | FPR95↓ | AUROC↑ | AUPR↑ | FPR95↓ | AUROC↑ | AUPR↑ | FPR95↓ | AUROC↑ | AUPR↑ |
| *Post-hoc Methods* | | | | | | | | | |
| MSP | $55.79_{\pm6.79}$ | $87.19_{\pm0.70}$ | $85.54_{\pm2.13}$ | $45.17_{\pm10.18}$ | $89.15_{\pm0.79}$ | $89.32_{\pm2.23}$ | $50.48_{\pm8.48}$ | $88.17_{\pm0.75}$ | $87.43_{\pm2.18}$ |
| ODIN | $81.41_{\pm8.35}$ | $77.49_{\pm4.32}$ | $72.54_{\pm6.77}$ | $77.14_{\pm11.25}$ | $80.14_{\pm5.15}$ | $77.78_{\pm7.65}$ | $79.27_{\pm9.77}$ | $78.81_{\pm4.73}$ | $75.16_{\pm7.20}$ |
| EBO | $68.87_{\pm9.60}$ | $86.17_{\pm1.53}$ | $82.46_{\pm3.70}$ | $58.51_{\pm14.41}$ | $89.03_{\pm1.61}$ | $87.47_{\pm3.68}$ | $63.69_{\pm11.99}$ | $87.60_{\pm1.57}$ | $84.97_{\pm3.69}$ |
| KNN | $37.22_{\pm1.22}$ | $89.86_{\pm0.25}$ | $90.50_{\pm0.35}$ | $29.75_{\pm1.56}$ | $91.78_{\pm0.29}$ | $93.52_{\pm0.34}$ | $33.49_{\pm1.36}$ | $90.82_{\pm0.27}$ | $92.01_{\pm0.35}$ |
| ASH | $78.55_{\pm13.02}$ | $79.25_{\pm7.49}$ | $73.97_{\pm11.11}$ | $72.30_{\pm17.28}$ | $81.94_{\pm7.55}$ | $79.09_{\pm11.10}$ | $75.42_{\pm15.11}$ | $80.60_{\pm7.51}$ | $76.53_{\pm11.10}$ |
| Scale | $85.48_{\pm8.59}$ | $70.17_{\pm13.00}$ | $66.55_{\pm12.66}$ | $82.13_{\pm11.10}$ | $72.94_{\pm13.93}$ | $72.03_{\pm13.39}$ | $83.80_{\pm9.82}$ | $71.56_{\pm13.46}$ | $69.29_{\pm13.02}$ |
| Relation | $40.40_{\pm2.16}$ | $88.99_{\pm0.28}$ | $89.13_{\pm0.43}$ | $33.28_{\pm1.94}$ | $90.89_{\pm0.32}$ | $92.63_{\pm0.39}$ | $36.84_{\pm2.05}$ | $89.94_{\pm0.30}$ | $90.88_{\pm0.41}$ |
| *Regularization-based Methods* | | | | | | | | | |
| CSI | $36.77_{\pm1.60}$ | $88.22_{\pm0.87}$ | $89.91_{\pm0.71}$ | $29.53_{\pm2.76}$ | $90.82_{\pm0.99}$ | $93.09_{\pm0.73}$ | $33.15_{\pm2.04}$ | $89.52_{\pm0.93}$ | $91.50_{\pm0.71}$ |
| SSD+ | $36.52_{\pm0.68}$ | $89.56_{\pm0.21}$ | $90.36_{\pm0.17}$ | $28.43_{\pm0.92}$ | $91.79_{\pm0.23}$ | $93.46_{\pm0.08}$ | $32.48_{\pm0.77}$ | $90.67_{\pm0.21}$ | $91.91_{\pm0.06}$ |
| KNN+ | $35.00_{\pm1.07}$ | $90.02_{\pm0.56}$ | $91.05_{\pm0.30}$ | $26.57_{\pm1.98}$ | $92.34_{\pm0.48}$ | $94.05_{\pm0.38}$ | $30.79_{\pm1.52}$ | $91.18_{\pm0.51}$ | $92.55_{\pm0.32}$ |
| VOS | $69.56_{\pm13.18}$ | $85.74_{\pm2.46}$ | $81.66_{\pm5.35}$ | $60.86_{\pm17.63}$ | $88.59_{\pm2.31}$ | $86.55_{\pm5.00}$ | $65.21_{\pm15.41}$ | $87.16_{\pm2.39}$ | $84.10_{\pm5.17}$ |
| CIDER | $32.63_{\pm0.61}$ | $90.47_{\pm0.13}$ | $90.95_{\pm0.27}$ | $23.62_{\pm0.62}$ | $93.17_{\pm0.05}$ | $94.91_{\pm0.07}$ | $28.12_{\pm0.39}$ | $91.82_{\pm0.08}$ | $92.93_{\pm0.12}$ |
| NPOS | $32.46_{\pm1.22}$ | $90.73_{\pm0.32}$ | $91.49_{\pm0.43}$ | $25.73_{\pm0.36}$ | $93.41_{\pm0.30}$ | $94.82_{\pm0.24}$ | $29.09_{\pm0.62}$ | $92.07_{\pm0.26}$ | $93.15_{\pm0.27}$ |
| PALM | $55.75_{\pm3.88}$ | $81.91_{\pm1.50}$ | $83.41_{\pm1.46}$ | $46.54_{\pm5.95}$ | $86.21_{\pm1.89}$ | $89.00_{\pm1.81}$ | $51.15_{\pm4.85}$ | $84.06_{\pm1.70}$ | $86.20_{\pm1.62}$ |
| **HamOS(ours)** | $\mathbf{29.57_{\pm0.95}}$ | $\mathbf{91.36_{\pm0.40}}$ | $\mathbf{92.11_{\pm0.91}}$ | $\mathbf{20.72_{\pm1.35}}$ | $\mathbf{94.37_{\pm0.31}}$ | $\mathbf{95.58_{\pm1.07}}$ | $\mathbf{25.14_{\pm0.72}}$ | $\mathbf{92.86_{\pm0.17}}$ | $\mathbf{93.85_{\pm0.80}}$ |

Table 7: Results on hard OOD benchmarks with CIFAR-100 as ID data(%). Comparison with competitive OOD detection baselines. (averaged by multiple runs)

| Methods | CIFAR-10 | | | TinyImageNet | | | Averaged | | |
|---|---|---|---|---|---|---|---|---|---|
| | FPR95↓ | AUROC↑ | AUPR↑ | FPR95↓ | AUROC↑ | AUPR↑ | FPR95↓ | AUROC↑ | AUPR↑ |
| *Post-hoc Methods* | | | | | | | | | |
| MSP | $\mathbf{62.91_{\pm0.33}}$ | $77.05_{\pm0.16}$ | $77.71_{\pm0.46}$ | $52.74_{\pm0.79}$ | $80.81_{\pm0.30}$ | $86.64_{\pm0.23}$ | $\mathbf{57.83_{\pm0.35}}$ | $78.93_{\pm0.10}$ | $82.18_{\pm0.15}$ |
| ODIN | $68.10_{\pm1.66}$ | $75.97_{\pm0.44}$ | $75.47_{\pm0.74}$ | $66.25_{\pm1.30}$ | $78.70_{\pm0.42}$ | $83.50_{\pm0.50}$ | $67.18_{\pm1.25}$ | $77.34_{\pm0.30}$ | $79.49_{\pm0.48}$ |
| EBO | $64.18_{\pm1.63}$ | $77.82_{\pm0.53}$ | $\mathbf{77.86_{\pm0.88}}$ | $57.42_{\pm1.10}$ | $81.02_{\pm0.16}$ | $86.12_{\pm0.20}$ | $60.80_{\pm1.13}$ | $79.42_{\pm0.29}$ | $81.99_{\pm0.48}$ |
| KNN | $72.31_{\pm2.09}$ | $76.86_{\pm0.46}$ | $74.27_{\pm0.81}$ | $\mathbf{49.62_{\pm0.92}}$ | $\mathbf{83.33_{\pm0.26}}$ | $\mathbf{88.38_{\pm0.18}}$ | $60.97_{\pm1.39}$ | $80.10_{\pm0.33}$ | $81.33_{\pm0.44}$ |
| ASH | $70.60_{\pm3.80}$ | $75.04_{\pm1.66}$ | $73.84_{\pm2.17}$ | $71.51_{\pm3.98}$ | $76.88_{\pm1.43}$ | $81.33_{\pm1.64}$ | $71.06_{\pm3.85}$ | $75.96_{\pm1.54}$ | $77.59_{\pm1.90}$ |
| Scale | $74.68_{\pm1.95}$ | $74.48_{\pm1.32}$ | $72.14_{\pm1.61}$ | $79.61_{\pm2.85}$ | $73.55_{\pm1.56}$ | $77.53_{\pm1.71}$ | $77.15_{\pm2.37}$ | $74.01_{\pm1.44}$ | $74.83_{\pm1.65}$ |
| Relation | $75.26_{\pm1.11}$ | $76.71_{\pm0.37}$ | $72.43_{\pm0.48}$ | $51.13_{\pm0.40}$ | $82.74_{\pm0.17}$ | $87.64_{\pm0.12}$ | $63.19_{\pm0.55}$ | $79.72_{\pm0.21}$ | $80.04_{\pm0.21}$ |
| *Regularization-based Methods* | | | | | | | | | |
| CSI | $74.13_{\pm1.92}$ | $69.09_{\pm0.27}$ | $69.95_{\pm0.83}$ | $67.64_{\pm0.57}$ | $73.51_{\pm0.30}$ | $80.96_{\pm0.35}$ | $70.89_{\pm1.25}$ | $71.30_{\pm0.28}$ | $75.46_{\pm0.59}$ |
| SSD+ | $66.41_{\pm0.89}$ | $\mathbf{77.99_{\pm0.25}}$ | $76.80_{\pm0.43}$ | $51.19_{\pm0.50}$ | $82.48_{\pm0.48}$ | $87.94_{\pm0.25}$ | $58.80_{\pm0.21}$ | $\mathbf{80.24_{\pm0.36}}$ | $\mathbf{82.37_{\pm0.13}}$ |
| KNN+ | $80.28_{\pm2.28}$ | $73.08_{\pm0.45}$ | $69.98_{\pm1.19}$ | $57.78_{\pm1.05}$ | $80.64_{\pm0.28}$ | $85.92_{\pm0.36}$ | $69.03_{\pm1.61}$ | $76.86_{\pm0.36}$ | $77.95_{\pm0.76}$ |
| VOS | $72.46_{\pm0.67}$ | $73.61_{\pm0.55}$ | $72.73_{\pm0.72}$ | $70.02_{\pm2.02}$ | $78.06_{\pm0.56}$ | $82.59_{\pm0.80}$ | $71.24_{\pm1.09}$ | $75.84_{\pm0.29}$ | $77.66_{\pm0.49}$ |
| CIDER | $69.43_{\pm0.75}$ | $76.59_{\pm0.05}$ | $75.31_{\pm0.57}$ | $51.82_{\pm0.62}$ | $81.95_{\pm0.17}$ | $87.63_{\pm0.07}$ | $60.62_{\pm0.47}$ | $79.27_{\pm0.07}$ | $81.47_{\pm0.25}$ |
| NPOS | $77.27_{\pm2.20}$ | $73.98_{\pm0.45}$ | $70.66_{\pm1.40}$ | $55.49_{\pm0.91}$ | $80.55_{\pm0.61}$ | $86.19_{\pm0.45}$ | $66.38_{\pm1.19}$ | $77.27_{\pm0.52}$ | $78.42_{\pm0.90}$ |
| PALM | $96.06_{\pm0.22}$ | $48.47_{\pm0.63}$ | $45.68_{\pm0.44}$ | $87.23_{\pm0.35}$ | $64.13_{\pm0.61}$ | $69.85_{\pm0.31}$ | $91.64_{\pm0.19}$ | $56.30_{\pm0.42}$ | $57.77_{\pm0.18}$ |
| **HamOS(ours)** | $73.57_{\pm0.18}$ | $76.82_{\pm1.16}$ | $73.51_{\pm0.96}$ | $51.07_{\pm0.08}$ | $82.69_{\pm0.29}$ | $88.04_{\pm0.20}$ | $62.32_{\pm1.61}$ | $79.75_{\pm0.20}$ | $80.78_{\pm1.50}$ |

Table 8: Ablation results on compactness and separation with CIFAR-100 as ID dataset(%). Comparison with competitive OOD detection baselines. Results are averaged across multiple runs.

| Method | ID-OOD Separation↑ | | | | | | Inter-class Separation↑ | Intra-class Compactness↓ |
|---|---|---|---|---|---|---|---|---|
| | MNIST | SVHN | Textures | Places365 | LSUN | **Averaged** | | |
| SSD+ | $28.99_{\pm0.76}$ | $38.82_{\pm0.28}$ | $36.32_{\pm0.29}$ | $35.27_{\pm0.42}$ | $34.76_{\pm0.54}$ | $34.83_{\pm0.43}$ | $69.05_{\pm0.29}$ | $56.69_{\pm0.33}$ |
| CIDER | $42.26_{\pm1.34}$ | $59.61_{\pm0.48}$ | $52.47_{\pm0.50}$ | $50.58_{\pm0.24}$ | $51.56_{\pm0.59}$ | $51.30_{\pm0.56}$ | $90.37_{\pm0.05}$ | $73.81_{\pm0.17}$ |
| NPOS | $43.69_{\pm1.00}$ | $60.47_{\pm0.59}$ | $52.91_{\pm0.21}$ | $50.95_{\pm0.20}$ | $51.55_{\pm0.29}$ | $51.91_{\pm0.32}$ | $90.41_{\pm0.05}$ | $74.00_{\pm0.07}$ |
| PALM | $34.32_{\pm0.96}$ | $43.83_{\pm2.90}$ | $39.24_{\pm1.47}$ | $36.71_{\pm1.20}$ | $37.60_{\pm0.86}$ | $38.34_{\pm0.93}$ | $84.44_{\pm0.45}$ | $70.23_{\pm0.25}$ |
| **HamOS(ours)** | $\mathbf{45.87_{\pm0.97}}$ | $\mathbf{60.61_{\pm0.08}}$ | $\mathbf{53.67_{\pm0.11}}$ | $\mathbf{52.30_{\pm0.03}}$ | $\mathbf{53.14_{\pm0.39}}$ | $\mathbf{53.12_{\pm0.11}}$ | $90.30_{\pm0.01}$ | $73.70_{\pm0.01}$ |

Table 9: Ablation results on different sampling algorithms (%, time reported in milliseconds). Comparison with competitive OOD detection baselines. Results are averaged across multiple runs.

| Sam. Algo. | CIFAR-10 | | | | | CIFAR-100 | | | | |
|---|---|---|---|---|---|---|---|---|---|---|
| | FPR95↓ | AUROC↑ | AUPR↑ | ID-ACC↑ | Time↓ | FPR95↓ | AUROC↑ | AUPR↑ | ID-ACC↑ | Time↓ |
| Random Walk | $18.90_{\pm0.23}$ | $95.08_{\pm0.50}$ | $91.35_{\pm1.05}$ | $92.68_{\pm0.60}$ | 0.96 | $50.05_{\pm2.36}$ | $81.42_{\pm1.25}$ | $72.69_{\pm1.35}$ | $75.33_{\pm0.25}$ | 3.24 |
| HMC | $\mathbf{10.48_{\pm0.76}}$ | $\mathbf{97.11_{\pm0.26}}$ | $\mathbf{94.94_{\pm0.86}}$ | $\mathbf{94.67_{\pm0.15}}$ | 34.71 | $\mathbf{46.68_{\pm1.44}}$ | $\mathbf{83.64_{\pm0.64}}$ | $\mathbf{75.52_{\pm1.30}}$ | $76.12_{\pm0.14}$ | 287.48 |
| MALA | $11.06_{\pm0.62}$ | $96.74_{\pm0.22}$ | $94.29_{\pm0.36}$ | $94.51_{\pm0.05}$ | 9.06 | $47.83_{\pm1.79}$ | $83.10_{\pm0.25}$ | $74.66_{\pm1.48}$ | $75.78_{\pm0.44}$ | 29.80 |
| mMALA | $10.89_{\pm1.27}$ | $97.06_{\pm0.80}$ | $94.53_{\pm0.10}$ | $94.00_{\pm0.55}$ | 15.32 | $47.27_{\pm1.73}$ | $83.36_{\pm0.32}$ | $74.97_{\pm0.35}$ | $75.90_{\pm0.06}$ | 44.96 |
| RMHMC | $11.21_{\pm0.52}$ | $97.07_{\pm0.15}$ | $94.77_{\pm0.33}$ | $94.02_{\pm0.08}$ | 47.03 | $46.80_{\pm0.68}$ | $83.62_{\pm0.66}$ | $74.86_{\pm1.02}$ | $\mathbf{76.16_{\pm0.08}}$ | 384.65 |

ble 10. The results show that our proposed HamOS is effective with different distance-based scoring functions. Since HamOS maintains the original classification head, which makes it compatible with other post-hoc methods, we further conduct ablation studies on the scoring functions in Table 11. HamOS consistently improves when equipped with different post-hoc methods, distinguishing its superiority in enhancing the model's intrinsic detection capability.

**Integrating with various ID contrastive objectives.** Recall that HamOS only requires differentiable ID representations, which makes it compatible with multiple ID contrastive losses. We provide

Table 10: Ablation results on different distance metric (%). Comparison with competitive OOD detection baselines. Results are averaged across multiple runs.

| Distance Metric | Methods | CIFAR-10 | | | CIFAR-100 | | |
|---|---|---|---|---|---|---|---|
| | | FPR95↓ | AUROC↑ | AUPR↑ | FPR95↓ | AUROC↑ | AUPR↑ |
| Mahalanobis | SSD+ | $18.49_{\pm1.20}$ | $94.85_{\pm0.57}$ | $90.88_{\pm0.83}$ | $54.03_{\pm1.92}$ | $80.64_{\pm0.60}$ | $69.73_{\pm1.09}$ |
| | CIDER | $16.47_{\pm0.90}$ | $95.38_{\pm0.49}$ | $92.22_{\pm0.17}$ | $48.34_{\pm2.24}$ | $81.99_{\pm1.14}$ | $73.67_{\pm1.54}$ |
| | NPOS | $\mathbf{13.10}_{\pm1.06}$ | $\mathbf{96.31}_{\pm0.52}$ | $\mathbf{93.77}_{\pm0.81}$ | $49.85_{\pm2.11}$ | $81.02_{\pm0.76}$ | $72.52_{\pm1.85}$ |
| | PALM | $32.25_{\pm4.14}$ | $90.54_{\pm1.46}$ | $84.44_{\pm2.14}$ | $55.13_{\pm0.97}$ | $79.95_{\pm1.26}$ | $70.21_{\pm1.38}$ |
| | **HamOS(ours)** | $13.29_{\pm0.79}$ | $95.56_{\pm0.48}$ | $93.37_{\pm0.39}$ | $\mathbf{46.29}_{\pm0.15}$ | $\mathbf{83.23}_{\pm0.35}$ | $\mathbf{75.22}_{\pm0.30}$ |
| KNN | KNN+ | $19.68_{\pm1.86}$ | $94.41_{\pm0.66}$ | $90.46_{\pm0.66}$ | $61.25_{\pm0.81}$ | $78.24_{\pm0.93}$ | $66.64_{\pm0.88}$ |
| | CIDER | $16.28_{\pm0.68}$ | $95.76_{\pm0.37}$ | $92.36_{\pm0.06}$ | $49.64_{\pm1.80}$ | $81.77_{\pm0.95}$ | $73.22_{\pm1.12}$ |
| | NPOS | $14.39_{\pm0.87}$ | $96.61_{\pm0.26}$ | $93.35_{\pm0.74}$ | $51.41_{\pm1.88}$ | $81.02_{\pm0.98}$ | $72.49_{\pm1.54}$ |
| | PALM | $40.03_{\pm2.66}$ | $84.95_{\pm1.72}$ | $78.28_{\pm1.74}$ | $72.51_{\pm2.48}$ | $74.08_{\pm2.12}$ | $60.04_{\pm1.83}$ |
| | **HamOS(ours)** | $\mathbf{10.48}_{\pm0.76}$ | $\mathbf{97.11}_{\pm0.26}$ | $\mathbf{94.94}_{\pm0.86}$ | $\mathbf{46.68}_{\pm1.44}$ | $\mathbf{83.64}_{\pm0.64}$ | $\mathbf{75.52}_{\pm1.30}$ |

the full averaged results of using different ID contrastive objectives in Table 12. Cross-entropy loss is adopted as the baseline. We calculate the ID probabilities as in Eqn. (12). The results show that the synthesized outliers in HamOS serve as valuable regularization for better ID-OOD differentiality.

**HamOS outstrips baselines when training from scratch.** We also apply the proposed HamOS to train from scratch scenarios. We train the initial model for 100 and 500 epochs on CIFAR-10 and CIFAR-100 datasets, with a learning rate initially set to 0.1 and 0.5, respectively. The warming-up epochs are 20 and 50, respectively, for training 100 epochs and training 500 epochs, during which the model is only trained on ID data with Cross-entropy loss. For synthesis-based methods (i.e., VOS, NPOS, and our HamOS), the synthesis process begins at the 50th and 250th epoch, respectively. The results of training from scratch for 100 epochs are in Table 13, showing that HamOS is still effective when training from scratch. Note that the results in Table 13 are generally worse than those in Table 1, which can be reasonable since the fine-tuning is based on the well-trained models. The results of training from scratch for 500 epochs are in Table 14, demonstrating that HamOS can still outperform the baselines in this scenario.

**Computational Efficiency of HamOS.** We report the synthesis time of a mini-batch for VOS (Du et al., 2022b), NPOS (Tao et al., 2023) and our HamOS in Table 15 on CIFAR-10, CIFAR-100, and ImageNet-1K respectively. The results show that HamOS only introduces polynomial extra time cost while synthesizing outliers of much higher quality than the baselines. The computational efficiency results from two factors:

- High quality sampling. Since HamOS can generate diverse and representative outliers, it necessitates much fewer outliers to achieve superior performance than VOS and NPOS. In default configurations, HamOS only generates 20 OOD samples per class, whereas VOS samples 200 outliers and NPOS samples 400 outliers per class from Gaussian distribution. The low requirement of virtual outliers enables HamOS to perform efficiently, particularly in large-scale scenarios.

- High efficient implementation. The Markov chains are generated parallelly across all classes using matrix operations. Additionally, the advanced similarity search technique also boosts the calculation of $k$-NN scores. This can be more straightforwardly verified by consulting the code in the supplementary material.

**Sensitivity of the initial midpoints.** As our primary goal is to synthesize diverse and representative outliers with the estimation of the target unknown distribution produced by the $k$-NN distance metric, we iteratively sample data points from low OOD-ness region to high OOD-ness region. To this end, we choose the midpoints of ID cluster pairs with relatively low $k$-NN distances from ID clusters, while remaining confusing and pivotal for the two ID clusters.

To examine the sensitivity of HamOS to the choice of midpoints, we apply Gaussian noise to the initial midpoints at different levels as small perturbations and display the results in Table 2 below. The scale of Gaussian noise is controlled by the standard deviation $\sigma$. The results in Table 16 show that the performance of HamOS continuously deteriorates as the amount of Gaussian noise to the midpoints increases, implying that the choice of midpoints between ID clusters is essential. Intuitively, perturbations on the midpoints may compel the initial states into areas that are of high

Table 11: Ablation on different scoring functions (%). Comparison with competitive OOD detection baselines. Results are averaged across multiple runs.

| ID datasets | Scoring Function | FPR95↓ | AUROC↑ | AUPR↑ |
|---|---|---|---|---|
| CIFAR-10 | MSP | $32.17_{\pm6.38}$ | $91.10_{\pm0.71}$ | $81.70_{\pm5.82}$ |
| | MSP+HamOS | $\mathbf{18.63_{\pm1.90}}$ | $\mathbf{94.41_{\pm0.75}}$ | $\mathbf{90.12_{\pm0.92}}$ |
| | ODIN | $58.04_{\pm18.46}$ | $85.70_{\pm4.17}$ | $70.08_{\pm11.84}$ |
| | ODIN+HamOS | $\mathbf{26.20_{\pm1.68}}$ | $\mathbf{94.62_{\pm1.11}}$ | $\mathbf{88.10_{\pm0.60}}$ |
| | EBO | $41.85_{\pm13.78}$ | $91.79_{\pm1.54}$ | $79.70_{\pm8.10}$ |
| | EBO+HamOS | $\mathbf{17.13_{\pm0.98}}$ | $\mathbf{96.35_{\pm0.22}}$ | $\mathbf{91.77_{\pm1.45}}$ |
| | KNN | $22.86_{\pm1.12}$ | $92.98_{\pm0.42}$ | $88.74_{\pm0.79}$ |
| | KNN+HamOS | $\mathbf{10.48_{\pm0.76}}$ | $\mathbf{97.11_{\pm0.26}}$ | $\mathbf{94.94_{\pm0.86}}$ |
| | ASH | $54.22_{\pm26.06}$ | $87.37_{\pm6.60}$ | $72.33_{\pm16.40}$ |
| | ASH+HamOS | $\mathbf{33.21_{\pm0.32}}$ | $\mathbf{92.83_{\pm0.04}}$ | $\mathbf{81.35_{\pm0.51}}$ |
| | Scale | $63.18_{\pm23.64}$ | $77.74_{\pm16.24}$ | $63.03_{\pm20.52}$ |
| | Scale+HamOS | $\mathbf{23.09_{\pm0.45}}$ | $\mathbf{95.59_{\pm0.59}}$ | $\mathbf{86.95_{\pm1.17}}$ |
| | Relation | $26.28_{\pm1.63}$ | $92.31_{\pm0.43}$ | $86.75_{\pm0.98}$ |
| | Relation+HamOS | $\mathbf{13.07_{\pm0.80}}$ | $\mathbf{96.85_{\pm1.17}}$ | $\mathbf{93.45_{\pm0.96}}$ |
| CIFAR-100 | MSP | $59.78_{\pm2.16}$ | $77.25_{\pm1.28}$ | $66.86_{\pm1.58}$ |
| | MSP+HamOS | $\mathbf{52.97_{\pm1.19}}$ | $\mathbf{80.46_{\pm0.87}}$ | $\mathbf{71.78_{\pm0.48}}$ |
| | ODIN | $63.49_{\pm2.51}$ | $78.01_{\pm1.62}$ | $65.20_{\pm2.19}$ |
| | ODIN+HamOS | $\mathbf{59.06_{\pm0.22}}$ | $\mathbf{79.99_{\pm0.13}}$ | $\mathbf{68.55_{\pm0.44}}$ |
| | EBO | $60.86_{\pm1.87}$ | $78.32_{\pm1.31}$ | $66.73_{\pm1.35}$ |
| | EBO+HamOS | $\mathbf{49.68_{\pm0.92}}$ | $\mathbf{83.55_{\pm0.52}}$ | $\mathbf{74.52_{\pm0.40}}$ |
| | KNN | $56.96_{\pm2.96}$ | $81.01_{\pm1.19}$ | $70.60_{\pm2.29}$ |
| | KNN+HamOS | $\mathbf{46.68_{\pm1.44}}$ | $\mathbf{83.64_{\pm0.64}}$ | $\mathbf{75.52_{\pm1.30}}$ |
| | ASH | $66.84_{\pm0.87}$ | $77.14_{\pm1.12}$ | $62.24_{\pm0.73}$ |
| | ASH+HamOS | $\mathbf{60.31_{\pm1.79}}$ | $\mathbf{77.19_{\pm0.15}}$ | $\mathbf{66.63_{\pm0.51}}$ |
| | Scale | $69.27_{\pm2.31}$ | $77.25_{\pm1.01}$ | $61.42_{\pm1.42}$ |
| | Scale+HamOS | $\mathbf{53.60_{\pm1.40}}$ | $\mathbf{84.16_{\pm0.12}}$ | $\mathbf{72.57_{\pm0.64}}$ |
| | Relation | $59.64_{\pm2.48}$ | $79.69_{\pm1.08}$ | $68.76_{\pm1.78}$ |
| | Relation+HamOS | $\mathbf{46.88_{\pm0.45}}$ | $\mathbf{84.69_{\pm1.03}}$ | $\mathbf{76.65_{\pm1.28}}$ |
| ImageNet-1K | MSP | $57.91_{\pm0.35}$ | $82.63_{\pm0.24}$ | $93.08_{\pm0.20}$ |
| | MSP+HamOS | $\mathbf{56.49_{\pm1.05}}$ | $\mathbf{83.09_{\pm0.40}}$ | $\mathbf{93.14_{\pm0.91}}$ |
| | ODIN | $52.12_{\pm1.39}$ | $86.34_{\pm0.48}$ | $94.45_{\pm0.74}$ |
| | ODIN+HamOS | $\mathbf{48.71_{\pm0.35}}$ | $\mathbf{87.86_{\pm0.10}}$ | $\mathbf{94.87_{\pm0.74}}$ |
| | EBO | $47.71_{\pm1.85}$ | $86.77_{\pm0.86}$ | $94.50_{\pm0.14}$ |
| | EBO+HamOS | $\mathbf{46.15_{\pm0.58}}$ | $\mathbf{87.21_{\pm1.07}}$ | $\mathbf{94.59_{\pm0.46}}$ |
| | KNN | $46.60_{\pm1.41}$ | $83.99_{\pm0.09}$ | $93.08_{\pm1.28}$ |
| | KNN+HamOS | $\mathbf{44.59_{\pm0.42}}$ | $\mathbf{88.59_{\pm0.08}}$ | $\mathbf{95.24_{\pm1.32}}$ |
| | ASH | $26.75_{\pm1.18}$ | $93.37_{\pm1.00}$ | $96.46_{\pm0.81}$ |
| | ASH+HamOS | $\mathbf{25.78_{\pm1.44}}$ | $\mathbf{94.07_{\pm0.94}}$ | $\mathbf{97.04_{\pm1.14}}$ |
| | Scale | $34.06_{\pm1.80}$ | $91.53_{\pm0.36}$ | $95.89_{\pm0.25}$ |
| | Scale+HamOS | $\mathbf{25.55_{\pm1.11}}$ | $\mathbf{94.01_{\pm0.30}}$ | $\mathbf{96.67_{\pm0.98}}$ |
| | Relation | $\mathbf{47.06_{\pm0.50}}$ | $\mathbf{86.98_{\pm0.67}}$ | $94.00_{\pm1.23}$ |
| | Relation+HamOS | $49.08_{\pm0.91}$ | $85.81_{\pm0.80}$ | $\mathbf{94.22_{\pm0.86}}$ |

OOD-ness, undermining the diversity of outliers. The perturbed midpoints may also reside in the ID clusters, bringing down the acceptance rate.

**Visual verification of the efficacy of HamOS.** We further verify the effectiveness of HamOS in Figure 8, showing that outliers synthesized by HamOS can help broaden the gap between ID and OOD distributions. The distribution of synthesized outliers occupies a wide range of OOD scores, indicating its diversity and representativeness. We additionally provide the score distributions of ID and OOD samples along the training on CIFAR-100 in Figure 9 as a supplement to Figure 5. Figure 9 shows that the outliers of various OOD scores are helpful in regularizing the model for better detection performance.

Table 12: Ablation results on different ID contrastive training objectives (%). Comparison with competitive OOD detection baselines. Results are averaged across multiple runs.

| Method | CIFAR-10 | | | | CIFAR-100 | | | |
|---|---|---|---|---|---|---|---|---|
| | FPR95↓ | AUROC↑ | AUPR↑ | ID-ACC↑ | FPR95↓ | AUROC↑ | AUPR↑ | ID-ACC↑ |
| CE | 28.46 | 91.09 | 86.08 | **94.26** | 66.42 | 75.34 | 61.22 | **76.64** |
| CE+HamOS | **21.21** | **93.20** | **88.28** | 94.04 | **57.28** | **78.64** | **64.68** | 74.46 |
| SimCLR | 22.12 | 92.74 | 88.19 | **93.66** | 58.37 | 76.14 | 62.05 | 72.01 |
| SimCLR+HamOS | **16.56** | **93.99** | **88.57** | 93.37 | **53.76** | **81.45** | **67.72** | **72.79** |
| SupCon | 19.68 | 94.41 | 90.46 | **93.79** | 61.25 | 78.24 | 66.64 | 72.18 |
| SupCon+HamOS | **15.49** | **95.74** | **93.46** | 93.72 | **55.94** | **81.76** | **67.61** | **72.69** |
| CIDER | 16.28 | 95.76 | 92.36 | 93.98 | 49.64 | 81.77 | 73.22 | 75.09 |
| CIDER+HamOS | **10.48** | **97.11** | **94.94** | **94.67** | **46.68** | **83.64** | **75.52** | **76.12** |

Table 13: Training from scratch for 100 epochs with CIFAR-10/100 as ID dataset (%). Comparison with competitive OOD detection baselines. Results are averaged across multiple OOD test datasets with multiple runs.

| Methods | CIFAR-10 | | | | CIFAR-100 | | | |
|---|---|---|---|---|---|---|---|---|
| | FPR95↓ | AUROC↑ | AUPR↑ | ID-ACC↑ | FPR95↓ | AUROC↑ | AUPR↑ | ID-ACC↑ |
| CSI | $19.64_{\pm1.44}$ | $94.99_{\pm0.47}$ | $90.59_{\pm0.68}$ | $91.61_{\pm0.24}$ | $81.42_{\pm0.69}$ | $64.80_{\pm0.66}$ | $50.93_{\pm0.41}$ | $50.30_{\pm1.91}$ |
| SSD+ | $20.19_{\pm0.74}$ | $93.96_{\pm0.13}$ | $89.55_{\pm0.77}$ | $93.84_{\pm0.26}$ | $56.86_{\pm2.08}$ | $79.69_{\pm1.64}$ | $68.53_{\pm2.49}$ | $73.84_{\pm0.85}$ |
| KNN+ | $22.74_{\pm0.10}$ | $93.17_{\pm0.11}$ | $88.48_{\pm0.35}$ | $93.76_{\pm0.29}$ | $60.88_{\pm2.00}$ | $76.07_{\pm1.30}$ | $65.15_{\pm0.96}$ | $73.39_{\pm1.25}$ |
| VOS | $27.06_{\pm1.78}$ | $93.89_{\pm0.32}$ | $85.95_{\pm1.64}$ | $\mathbf{95.32_{\pm0.08}}$ | $61.27_{\pm2.52}$ | $77.80_{\pm0.88}$ | $65.67_{\pm1.89}$ | $\mathbf{77.57_{\pm0.69}}$ |
| CIDER | $18.79_{\pm2.14}$ | $95.51_{\pm0.72}$ | $90.42_{\pm1.24}$ | $93.37_{\pm0.06}$ | $54.36_{\pm2.40}$ | $80.27_{\pm1.02}$ | $70.31_{\pm1.72}$ | $74.43_{\pm0.33}$ |
| NPOS | $26.42_{\pm4.04}$ | $92.70_{\pm0.74}$ | $85.48_{\pm1.71}$ | $93.33_{\pm0.13}$ | $77.56_{\pm1.20}$ | $73.69_{\pm0.35}$ | $54.95_{\pm1.03}$ | $74.71_{\pm0.34}$ |
| PALM | $16.79_{\pm1.29}$ | $96.02_{\pm0.61}$ | $92.16_{\pm0.49}$ | $94.43_{\pm0.57}$ | $57.54_{\pm0.32}$ | $75.32_{\pm1.76}$ | $67.40_{\pm0.54}$ | $74.00_{\pm0.15}$ |
| **HamOS(ours)** | $\mathbf{12.14_{\pm0.25}}$ | $\mathbf{96.67_{\pm0.07}}$ | $\mathbf{94.45_{\pm0.35}}$ | $94.99_{\pm0.10}$ | $\mathbf{50.12_{\pm1.31}}$ | $\mathbf{80.91_{\pm0.65}}$ | $\mathbf{72.61_{\pm1.47}}$ | $75.47_{\pm0.09}$ |

Table 14: Training from scratch for 500 epochs with CIFAR-10/100 as ID dataset (%). Comparison with competitive OOD detection baselines. Results are averaged across multiple OOD test datasets with multiple runs.

| Methods | CIFAR-10 | | | | CIFAR-100 | | | |
|---|---|---|---|---|---|---|---|---|
| | FPR95↓ | AUROC↑ | AUPR↑ | ID-ACC↑ | FPR95↓ | AUROC↑ | AUPR↑ | ID-ACC↑ |
| CSI | $12.35_{\pm0.07}$ | $96.83_{\pm0.05}$ | $\mathbf{94.53_{\pm0.07}}$ | $95.17_{\pm0.19}$ | $67.68_{\pm1.00}$ | $72.77_{\pm1.44}$ | $61.05_{\pm1.27}$ | $65.48_{\pm0.11}$ |
| SSD+ | $21.27_{\pm0.38}$ | $92.78_{\pm0.09}$ | $89.32_{\pm0.42}$ | $93.60_{\pm0.21}$ | $63.72_{\pm0.85}$ | $73.09_{\pm0.57}$ | $61.60_{\pm0.79}$ | $71.40_{\pm1.06}$ |
| KNN+ | $34.21_{\pm6.01}$ | $90.23_{\pm1.78}$ | $83.34_{\pm2.71}$ | $88.77_{\pm2.61}$ | $65.81_{\pm2.02}$ | $75.11_{\pm0.95}$ | $62.73_{\pm1.06}$ | $69.84_{\pm1.40}$ |
| VOS | $32.53_{\pm1.78}$ | $91.98_{\pm0.34}$ | $85.59_{\pm1.20}$ | $93.62_{\pm0.25}$ | $52.31_{\pm3.67}$ | $82.10_{\pm1.10}$ | $72.82_{\pm2.44}$ | $75.32_{\pm2.05}$ |
| CIDER | $11.59_{\pm0.10}$ | $97.24_{\pm0.04}$ | $94.44_{\pm0.31}$ | $\mathbf{95.43_{\pm0.20}}$ | $48.71_{\pm1.25}$ | $79.93_{\pm0.75}$ | $73.39_{\pm0.87}$ | $\mathbf{75.76_{\pm0.23}}$ |
| NPOS | $19.61_{\pm1.10}$ | $95.20_{\pm0.30}$ | $88.06_{\pm2.73}$ | $95.33_{\pm0.04}$ | $72.89_{\pm2.29}$ | $71.85_{\pm2.26}$ | $58.65_{\pm1.33}$ | $71.26_{\pm6.62}$ |
| PALM | $15.52_{\pm5.17}$ | $77.47_{\pm2.77}$ | $93.74_{\pm6.32}$ | $93.17_{\pm0.15}$ | $45.56_{\pm2.82}$ | $86.10_{\pm0.69}$ | $77.62_{\pm1.13}$ | $70.36_{\pm0.08}$ |
| **HamOS(ours)** | $\mathbf{10.57_{\pm2.17}}$ | $\mathbf{97.73_{\pm0.29}}$ | $94.41_{\pm1.14}$ | $93.83_{\pm0.16}$ | $\mathbf{40.81_{\pm1.07}}$ | $\mathbf{89.31_{\pm0.65}}$ | $\mathbf{79.69_{\pm0.30}}$ | $75.46_{\pm12.47}$ |

Table 15: Synthesis time for generating a mini-batch of outliers (seconds).

| Method | CIFAR-10 | CIFAR-100 | ImageNet-1K |
|---|---|---|---|
| VOS | 0.021 | 0.233 | 3.371 |
| NPOS | 0.027 | 0.254 | 4.695 |
| **HamOS(ours)** | 0.035 | 0.287 | 3.625 |

Table 16: Perturbation on the initial midpoints (%). Comparison with competitive OOD detection baselines. Results are averaged across multiple runs.

| $\sigma$ | CIFAR-10 | | | | CIFAR-100 | | | |
|---|---|---|---|---|---|---|---|---|
| | FPR95↓ | AUROC↑ | AUPR↑ | ID-ACC↑ | FPR95↓ | AUROC↑ | AUPR↑ | ID-ACC↑ |
| 0.0 (default) | **10.48** | **97.11** | **94.94** | 94.67 | **46.68** | **83.64** | **75.52** | **76.12** |
| 0.001 | 10.53 | 96.96 | 94.38 | 94.48 | 47.18 | 83.63 | 74.87 | 75.64 |
| 0.01 | 10.82 | 96.50 | 94.89 | 94.34 | 47.64 | 82.67 | 74.42 | 75.73 |
| 0.05 | 11.08 | 96.49 | 94.04 | 94.44 | 47.43 | 82.55 | 74.02 | 75.63 |
| 0.1 | 12.61 | 96.16 | 93.41 | **94.72** | 47.64 | 82.90 | 74.62 | 76.06 |
| 0.2 | 11.83 | 96.69 | 94.12 | 94.33 | 48.11 | 82.74 | 74.19 | 75.76 |

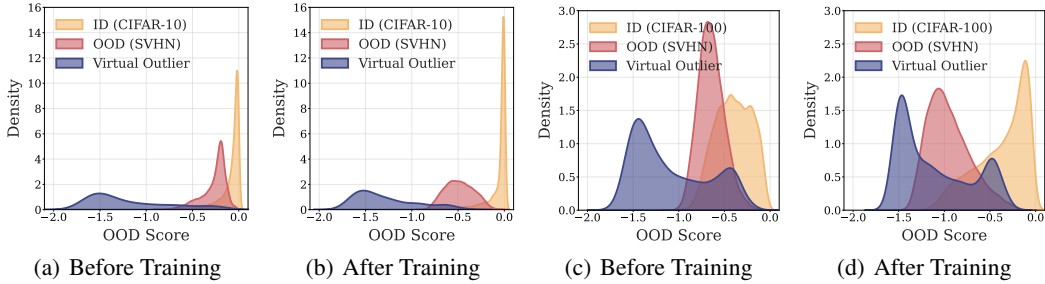

(a) Before Training     (b) After Training     (c) Before Training     (d) After Training

Figure 8: **Visual verification of the efficacy of HamOS on CIFAR-10 and CIFAR-100:** (a) before training with HamOS outliers;(b) after training with HamOS outliers;(c) before training with HamOS outliers;(d) after training with HamOS outliers.

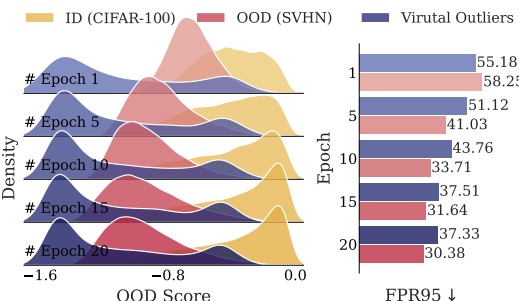

Figure 9: The trend of OOD score distributions of ID, OOD, and virtual samples along the training on CIFAR-100.

Table 17: Ablation results on loss weight $\lambda_d$ (%). Comparison with competitive OOD detection baselines. Results are averaged across multiple runs.

| $\lambda_d$ | CIFAR-10 | | | | CIFAR-100 | | | |
|---|---|---|---|---|---|---|---|---|
| | FPR95↓ | AUROC↑ | AUPR↑ | ID-ACC↑ | FPR95↓ | AUROC↑ | AUPR↑ | ID-ACC↑ |
| 0.01 | $12.92_{\pm1.53}$ | $96.61_{\pm0.34}$ | $93.52_{\pm0.81}$ | $\mathbf{94.67_{\pm0.21}}$ | $48.12_{\pm1.38}$ | $82.51_{\pm0.02}$ | $74.40_{\pm0.96}$ | $75.88_{\pm0.37}$ |
| 0.05 | $11.89_{\pm1.45}$ | $96.71_{\pm1.19}$ | $94.22_{\pm0.41}$ | $94.59_{\pm0.44}$ | $47.17_{\pm0.77}$ | $82.83_{\pm0.68}$ | $74.72_{\pm1.17}$ | $75.91_{\pm0.54}$ |
| **0.1(default)** | $\mathbf{10.48_{\pm0.76}}$ | $\mathbf{97.11_{\pm0.26}}$ | $\mathbf{94.94_{\pm0.86}}$ | $\mathbf{94.67_{\pm0.15}}$ | $\mathbf{46.68_{\pm1.44}}$ | $\mathbf{83.64_{\pm0.64}}$ | $\mathbf{75.52_{\pm1.30}}$ | $\mathbf{76.12_{\pm0.14}}$ |
| 0.3 | $12.71_{\pm0.39}$ | $96.51_{\pm0.51}$ | $94.21_{\pm0.06}$ | $94.38_{\pm0.06}$ | $48.37_{\pm0.03}$ | $82.86_{\pm0.29}$ | $74.69_{\pm1.11}$ | $75.71_{\pm0.49}$ |
| 0.5 | $17.38_{\pm0.20}$ | $95.31_{\pm0.39}$ | $92.28_{\pm1.46}$ | $94.32_{\pm0.29}$ | $48.56_{\pm0.14}$ | $82.94_{\pm0.16}$ | $73.81_{\pm0.62}$ | $75.42_{\pm0.03}$ |

## F.2 ADDITIONAL ABLATION STUDIES

**Ablation on loss weight for $\mathcal{L}_{\textbf{OOD-disc}}$.** In HamOS, the virtual outliers are used to derive the OOD discernment loss $\mathcal{L}_{\text{OOD-disc}}$, which is combined to the CE loss and ID contrastive loss with a weight $\lambda_d$. In Table 17, we study the efficacy of weight $\lambda_d = \{0.01, 0.05, 0.1, 0.3, 0.5\}$. On the one hand, too small a value (e.g., 0.01) of $\lambda_d$ introduces mild supervision on the outliers, which leads to degraded OOD performance. On the other hand, $\lambda_d$ of high value (e.g., 0.5) may distort the hyperspherical space and thus undermine detection capability.

**Ablation on the $k$ value in OOD-ness estimation.** Recall that we estimate the OOD-ness density by averaging the KNN distance within a pair of ID clusters in Eqn. (2). In Table 18, we analyze the efficacy of $k$ with different values. The results show that a $k$ value that is too small can lead to degraded OOD performance while the performance is stable when $k$ is bigger.

**Ablation on hard margin $\delta$.** Recall that we apply a hard margin $\delta$ based on the initial points to prevent erroneous outliers that have high ID probability in Eqn. (5). We ablate on the hard margin $\delta$ in Table 19 with verying values (e.g., $\delta = \{0.0, 0.05, 0.1, 0.15, 0.2\}$). The results show that with a hard margin that is too small, erroneous outliers will be generated, which can introduce spurious OOD supervision signals, while a large hard margin will increase the rejection ratio and result in less diverse and representative outliers.

Table 18: Ablation results on k value for OOD-ness estimation (%). Comparison with competitive OOD detection baselines. Results are averaged across multiple runs.

| $k$ | CIFAR-10 | | | | CIFAR-100 | | | |
|---|---|---|---|---|---|---|---|---|
| | FPR95↓ | AUROC↑ | AUPR↑ | ID-ACC↑ | FPR95↓ | AUROC↑ | AUPR↑ | ID-ACC↑ |
| 50 | $13.63_{\pm1.98}$ | $96.05_{\pm0.89}$ | $93.14_{\pm0.55}$ | $94.46_{\pm0.37}$ | $49.47_{\pm1.68}$ | $81.70_{\pm0.22}$ | $72.92_{\pm1.25}$ | $76.09_{\pm0.01}$ |
| 100 | $10.69_{\pm1.46}$ | $96.60_{\pm0.38}$ | $94.41_{\pm1.17}$ | $\mathbf{94.68_{\pm0.28}}$ | $46.98_{\pm1.70}$ | $82.97_{\pm0.99}$ | $74.86_{\pm0.81}$ | $\mathbf{76.18_{\pm0.42}}$ |
| **200(default)** | $\mathbf{10.48_{\pm0.76}}$ | $\mathbf{97.11_{\pm0.26}}$ | $\mathbf{94.94_{\pm0.86}}$ | $94.67_{\pm0.15}$ | $\mathbf{46.68_{\pm1.44}}$ | $\mathbf{83.64_{\pm0.64}}$ | $75.52_{\pm1.30}$ | $76.12_{\pm0.14}$ |
| 500 | $11.00_{\pm0.02}$ | $96.50_{\pm0.72}$ | $93.82_{\pm1.11}$ | $93.93_{\pm0.43}$ | $47.11_{\pm1.26}$ | $83.58_{\pm0.10}$ | $75.34_{\pm0.80}$ | $75.78_{\pm0.45}$ |

Table 19: Ablation results on hard margin $\delta$ (%). Comparison with competitive OOD detection baselines. Results are averaged across multiple runs.

| $\delta$ | CIFAR-10 | | | | CIFAR-100 | | | |
|---|---|---|---|---|---|---|---|---|
| | FPR95↓ | AUROC↑ | AUPR↑ | ID-ACC↑ | FPR95↓ | AUROC↑ | AUPR↑ | ID-ACC↑ |
| 0.0 | $12.48_{\pm1.15}$ | $96.60_{\pm0.82}$ | $93.65_{\pm0.43}$ | $94.16_{\pm0.31}$ | $47.99_{\pm1.36}$ | $82.43_{\pm0.83}$ | $74.08_{\pm1.36}$ | $75.82_{\pm0.19}$ |
| 0.05 | $11.90_{\pm1.71}$ | $96.55_{\pm0.88}$ | $93.89_{\pm1.49}$ | $94.61_{\pm0.13}$ | $49.14_{\pm0.82}$ | $81.99_{\pm0.74}$ | $73.86_{\pm0.76}$ | $76.06_{\pm0.03}$ |
| **0.1(default)** | $\mathbf{10.48_{\pm0.76}}$ | $\mathbf{97.11_{\pm0.26}}$ | $\mathbf{94.94_{\pm0.86}}$ | $\mathbf{94.67_{\pm0.15}}$ | $\mathbf{46.68_{\pm1.44}}$ | $\mathbf{83.64_{\pm0.64}}$ | $\mathbf{75.52_{\pm1.30}}$ | $\mathbf{76.12_{\pm0.14}}$ |
| 0.15 | $11.48_{\pm0.41}$ | $96.72_{\pm0.49}$ | $93.94_{\pm0.74}$ | $94.57_{\pm0.58}$ | $48.18_{\pm0.06}$ | $82.19_{\pm0.36}$ | $74.02_{\pm1.02}$ | $75.79_{\pm0.22}$ |
| 0.2 | $12.45_{\pm0.50}$ | $96.56_{\pm0.79}$ | $93.62_{\pm0.01}$ | $94.51_{\pm0.41}$ | $47.34_{\pm0.22}$ | $83.40_{\pm0.71}$ | $74.67_{\pm1.23}$ | $76.03_{\pm0.00}$ |

Table 20: Ablation results on Leapfrog steps $L$ (%). Comparison with competitive OOD detection baselines. Results are averaged across multiple runs.

| $L$ | CIFAR-10 | | | | CIFAR-100 | | | |
|---|---|---|---|---|---|---|---|---|
| | FPR95↓ | AUROC↑ | AUPR↑ | ID-ACC↑ | FPR95↓ | AUROC↑ | AUPR↑ | ID-ACC↑ |
| 1 | $11.01_{\pm1.72}$ | $96.91_{\pm0.69}$ | $94.69_{\pm1.11}$ | $94.50_{\pm0.45}$ | $47.83_{\pm1.79}$ | $83.10_{\pm0.25}$ | $74.66_{\pm1.48}$ | $75.78_{\pm0.44}$ |
| 2 | $11.76_{\pm1.65}$ | $96.74_{\pm0.97}$ | $94.19_{\pm0.95}$ | $94.51_{\pm0.59}$ | $48.87_{\pm0.20}$ | $82.20_{\pm0.58}$ | $73.63_{\pm0.70}$ | $75.97_{\pm0.56}$ |
| **3(default)** | $\mathbf{10.48_{\pm0.76}}$ | $\mathbf{97.11_{\pm0.26}}$ | $\mathbf{94.94_{\pm0.86}}$ | $\mathbf{94.67_{\pm0.15}}$ | $\mathbf{46.68_{\pm1.44}}$ | $\mathbf{83.64_{\pm0.64}}$ | $\mathbf{75.52_{\pm1.30}}$ | $76.12_{\pm0.14}$ |
| 4 | $11.14_{\pm0.84}$ | $96.53_{\pm0.20}$ | $94.14_{\pm0.79}$ | $94.50_{\pm0.58}$ | $48.40_{\pm1.20}$ | $82.47_{\pm0.36}$ | $73.97_{\pm1.01}$ | $75.64_{\pm0.12}$ |
| 5 | $11.55_{\pm0.16}$ | $96.46_{\pm0.86}$ | $94.36_{\pm0.75}$ | $94.33_{\pm0.05}$ | $48.31_{\pm0.16}$ | $82.68_{\pm0.69}$ | $74.38_{\pm1.12}$ | $\mathbf{76.23_{\pm0.46}}$ |

Table 21: Ablation results on step size $\epsilon$ (%). Comparison with competitive OOD detection baselines. Results are averaged across multiple runs.

| $\epsilon$ | CIFAR-10 | | | | CIFAR-100 | | | |
|---|---|---|---|---|---|---|---|---|
| | FPR95↓ | AUROC↑ | AUPR↑ | ID-ACC↑ | FPR95↓ | AUROC↑ | AUPR↑ | ID-ACC↑ |
| 0.01 | $12.29_{\pm1.55}$ | $96.42_{\pm0.10}$ | $93.87_{\pm0.22}$ | $94.31_{\pm0.26}$ | $48.92_{\pm0.17}$ | $82.29_{\pm1.01}$ | $73.89_{\pm0.97}$ | $76.03_{\pm0.57}$ |
| 0.05 | $12.45_{\pm0.42}$ | $96.56_{\pm0.41}$ | $94.02_{\pm0.85}$ | $94.33_{\pm0.26}$ | $48.27_{\pm0.45}$ | $82.36_{\pm0.58}$ | $74.07_{\pm0.21}$ | $76.06_{\pm0.04}$ |
| **0.1(default)** | $\mathbf{10.48_{\pm0.76}}$ | $\mathbf{97.11_{\pm0.26}}$ | $\mathbf{94.94_{\pm0.86}}$ | $\mathbf{94.67_{\pm0.15}}$ | $\mathbf{46.68_{\pm1.44}}$ | $\mathbf{83.64_{\pm0.64}}$ | $\mathbf{75.52_{\pm1.30}}$ | $\mathbf{76.12_{\pm0.14}}$ |
| 0.3 | $11.54_{\pm0.47}$ | $96.54_{\pm0.61}$ | $93.98_{\pm0.15}$ | $94.66_{\pm0.23}$ | $46.86_{\pm1.67}$ | $83.45_{\pm0.73}$ | $75.36_{\pm0.82}$ | $75.93_{\pm0.24}$ |
| 0.5 | $12.04_{\pm0.30}$ | $96.49_{\pm0.50}$ | $93.87_{\pm0.95}$ | $94.18_{\pm0.44}$ | $47.13_{\pm0.32}$ | $83.14_{\pm0.65}$ | $74.81_{\pm0.85}$ | $75.98_{\pm0.33}$ |

**Ablation on Leapfrog steps $L$.** Leapfrog Integrator is a technique that simulates the true continue solution to Hamilton's Equation with discrete approximation. Leapfrog conducts $L$ updates to the momentum $q$ and position $z$. We provide ablation on different Leapfrog steps in Table 20. The results show that HamOS is insensitive to the Leapfrog steps $L$, achieving superior OOD detection performance with varied Leapfrog step values.

**Ablation on step size $\epsilon$.** The step size of HMC controls the rate of convergence. Large step size encourages fast convergence to the high OOD-ness regions while small step size leads to low convergence, which results in less diverse outliers, with results in Table 21 proving this attribute. A moderate step size $\epsilon$ facilitates generating diverse outliers that stretch out a wide range of OOD-ness density.

We provide visualizations of the synthesized outliers with different Leapfrog steps $L \in \{1, 2, 3, 4, 5\}$ and step sizes $\epsilon \in \{0.01, 0.05, 0.1, 0.3, 0.5\}$ on CIFAR-10 and CIFAR-100 in Figure 12 and Figure 13. The visualizations intuitively show that the synthesized outliers in each round will be less diverse with smaller step size (i.e., $\epsilon = 0.01$) and may converge fastly to high OOD-ness density with larger step size (i.e., $\epsilon = 0.5$). Besides, a minor Leapfrog step can result in outliers with small variance (i.e., the span of distributions is tiny), and a enormous Leapfrog step may lead to diverse but unstable distributions of outliers. As the visualization suggests, moderate values should be set for Leapfrog step $L$ and step size $\epsilon$.

Table 22: Ablation results on number of nearest ID clusters $N_{adj}$ (%). Comparison with competitive OOD detection baselines. Results are averaged across multiple runs.

| $N_{adj}$ | CIFAR-10 | | | | CIFAR-100 | | | |
|---|---|---|---|---|---|---|---|---|
| | FPR95↓ | AUROC↑ | AUPR↑ | ID-ACC↑ | FPR95↓ | AUROC↑ | AUPR↑ | ID-ACC↑ |
| 1/1 | $10.66_{\pm 0.94}$ | $\mathbf{97.15_{\pm 0.77}}$ | $94.80_{\pm 0.64}$ | $94.44_{\pm 0.23}$ | $47.42_{\pm 1.79}$ | $82.58_{\pm 0.36}$ | $74.91_{\pm 0.84}$ | $\mathbf{76.16_{\pm 0.13}}$ |
| 2/2 | $10.86_{\pm 0.56}$ | $96.96_{\pm 0.31}$ | $94.23_{\pm 0.48}$ | $94.46_{\pm 0.07}$ | $47.37_{\pm 0.50}$ | $83.11_{\pm 0.93}$ | $75.22_{\pm 0.38}$ | $76.09_{\pm 0.10}$ |
| 3/4(default) | $11.08_{\pm 0.35}$ | $97.00_{\pm 0.13}$ | $94.20_{\pm 0.64}$ | $94.61_{\pm 0.50}$ | $\mathbf{46.68_{\pm 1.44}}$ | $\mathbf{83.64_{\pm 0.64}}$ | $75.52_{\pm 1.30}$ | $76.12_{\pm 0.14}$ |
| 4(default)/10 | $\mathbf{10.48_{\pm 0.76}}$ | $97.11_{\pm 0.26}$ | $\mathbf{94.94_{\pm 0.86}}$ | $\mathbf{94.67_{\pm 0.15}}$ | $47.38_{\pm 0.30}$ | $83.58_{\pm 0.22}$ | $75.48_{\pm 0.21}$ | $76.01_{\pm 0.23}$ |
| 5/20 | $10.85_{\pm 1.07}$ | $96.86_{\pm 0.26}$ | $94.24_{\pm 0.92}$ | $94.50_{\pm 0.10}$ | $47.02_{\pm 1.30}$ | $83.48_{\pm 0.88}$ | $75.40_{\pm 0.56}$ | $75.96_{\pm 0.53}$ |

Table 23: Ablation results on synthesis rounds $R$ (%). Comparison with competitive OOD detection baselines. Results are averaged across multiple runs.

| $R$ | CIFAR-10 | | | | CIFAR-100 | | | |
|---|---|---|---|---|---|---|---|---|
| | FPR95↓ | AUROC↑ | AUPR↑ | ID-ACC↑ | FPR95↓ | AUROC↑ | AUPR↑ | ID-ACC↑ |
| 1 | $13.83_{\pm 0.62}$ | $95.51_{\pm 0.11}$ | $93.27_{\pm 0.97}$ | $94.60_{\pm 0.52}$ | $49.57_{\pm 0.10}$ | $81.89_{\pm 0.76}$ | $73.55_{\pm 1.16}$ | $75.87_{\pm 0.12}$ |
| 2 | $12.16_{\pm 1.10}$ | $96.74_{\pm 0.86}$ | $94.09_{\pm 0.12}$ | $94.66_{\pm 0.32}$ | $48.53_{\pm 0.32}$ | $82.35_{\pm 0.03}$ | $73.67_{\pm 1.41}$ | $75.64_{\pm 0.57}$ |
| 3 | $10.64_{\pm 0.41}$ | $96.93_{\pm 0.99}$ | $94.65_{\pm 0.43}$ | $94.60_{\pm 0.60}$ | $47.51_{\pm 0.07}$ | $82.42_{\pm 0.39}$ | $74.48_{\pm 0.17}$ | $75.91_{\pm 0.39}$ |
| 4 | $10.86_{\pm 0.17}$ | $96.99_{\pm 0.45}$ | $94.60_{\pm 0.15}$ | $\mathbf{94.72_{\pm 0.09}}$ | $47.20_{\pm 1.35}$ | $82.71_{\pm 0.28}$ | $74.92_{\pm 0.71}$ | $\mathbf{76.21_{\pm 0.16}}$ |
| 5(default) | $\mathbf{10.48_{\pm 0.76}}$ | $\mathbf{97.11_{\pm 0.26}}$ | $\mathbf{94.94_{\pm 0.86}}$ | $94.67_{\pm 0.15}$ | $\mathbf{46.68_{\pm 1.44}}$ | $\mathbf{83.64_{\pm 0.64}}$ | $\mathbf{75.52_{\pm 1.30}}$ | $76.12_{\pm 0.14}$ |

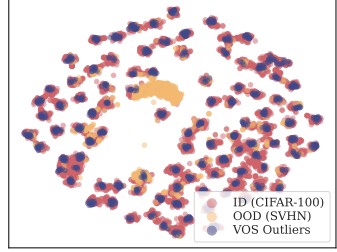 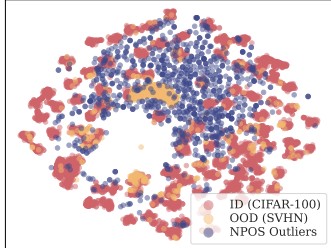 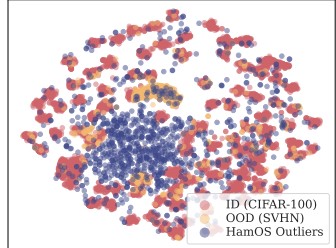

Figure 10: **t-SNE Visualization on CIFAR-100:** (a)VOS; (b)NPOS; (c)HamOS.

**Ablation on number of nearest neighbors $N_{adj}$.** To sample more representative outliers, we only generate outliers between though ID clusters close to each other. Specifically, we choose the $N_{adj}$ closest ID clusters w.r.t. to an ID cluster and calculate the corresponding midpoints as the start points of Markov chains. The results in Table 22 indicate that HamOS is not sensitive to the number of closest adjacent ID clusters considered to generate outliers. Note that we set $N_{adj} = \{1, 2, 3, 4, 5\}$ for CIFAR-10 and $N_{adj} = \{1, 2, 4, 10, 20\}$ for CIFAR-100, considering the number of ID classes.

**Ablation on synthesis rounds $R$.** As we formularize the outlier synthesis process as Markov chains, the length of Markov chains also impacts the quality of synthesized outliers. We ablate on the synthesis rounds $R$, reported in Table 23. The results show that too short of Markov chains (i.e., $R = 1, 2$) can't help achieve superior OOD performance, while greater $R$ (i.e., $> 2$) demonstrate efficacy, indicating the efficiency of HamOS for generating outliers.

**Ablation on Feature dimension.** In high-dimensional spaces, the Euclidean distance metric may be less discriminative, probably impairing the estimation of OOD-ness. The computational cost will also increase as the feature dimension scales. To assess the performance of HamOS with different feature dimensions, we conduct empirical analysis at dimensions 64, 256, and 512, alongside 4 representative baselines in Table 24. The results show that HamOS consistently achieves superior performance in almost all settings, demonstrating its stable performance across various feature dimensions. As the feature dimension scales up, HamOS only displays tiny performance degradation. Some baselines can be drastically influenced by the feature dimension, such as SSD+ which begins to collapse at 512 dimension on CIFAR-10 and PALM begins to collapse at 256 dimension on CIFAR-10 and 512 dimension on CIFAR-100. Surprisingly, PALM achieves high performance at 256 dimension on CIFAR-100, indicating that it requires careful tuning on the dimension to bring out the efficacy. Note that the dimension of the penultimate layer of the backbone is 512, so we scale up to 512 as a edge case.

Table 24: Ablation results on feature dimension (%). Comparison with competitive OOD detection baselines. Results are averaged across multiple runs.

| Dimension | Methods | CIFAR-10 | | | | CIFAR-100 | | | |
|---|---|---|---|---|---|---|---|---|---|
| | | FPR95↓ | AUROC↑ | AUPR↑ | ID-ACC↑ | FPR95↓ | AUROC↑ | AUPR↑ | ID-ACC↑ |
| 64 | SSD+ | 18.59 | 94.85 | 90.76 | 93.93 | 54.66 | 80.58 | 69.83 | **75.88** |
| | CIDER | 15.53 | 96.02 | 92.67 | 93.52 | 48.36 | 82.38 | 74.54 | 74.74 |
| | NPOS | 17.64 | 95.60 | 92.16 | 94.00 | 67.74 | 77.01 | 62.78 | 74.48 |
| | PALM | 44.36 | 84.02 | 76.43 | 93.87 | 66.95 | 78.89 | 63.55 | 74.45 |
| | **HamOS(ours)** | **12.66** | **96.16** | **93.47** | **94.51** | **47.74** | **82.57** | **75.25** | 75.21 |
| **128** **(Default)** | SSD+ | 18.49 | 94.85 | 90.88 | 93.95 | 54.03 | 80.64 | 69.73 | 75.63 |
| | CIDER | 16.28 | 95.76 | 92.36 | 93.98 | 49.64 | 81.77 | 73.22 | 75.09 |
| | NPOS | 14.39 | 96.61 | 93.35 | 93.95 | 51.41 | 81.02 | 72.49 | 74.53 |
| | PALM | 32.25 | 90.54 | 84.44 | 93.93 | 55.13 | 79.95 | 70.21 | 74.67 |
| | **HamOS(ours)** | **10.48** | **97.11** | **94.94** | **94.67** | **46.68** | **83.64** | **75.52** | **76.12** |
| 256 | SSD+ | 16.05 | 95.74 | 92.36 | **94.66** | 50.97 | 82.71 | 71.80 | **75.51** |
| | CIDER | 15.50 | 96.37 | 93.10 | 93.89 | 47.75 | 83.49 | 75.24 | 75.12 |
| | NPOS | 13.04 | 97.00 | 93.83 | 94.00 | 70.65 | 76.10 | 60.54 | 74.52 |
| | PALM | 92.74 | 68.18 | 42.41 | 10.00 | **46.81** | **83.62** | **75.48** | 73.69 |
| | **HamOS(ours)** | **10.81** | **97.03** | **94.18** | **94.66** | 47.26 | 83.61 | 75.36 | 75.28 |
| 512 | SSD+ | 85.59 | 66.83 | 46.64 | 93.46 | 51.70 | 81.67 | 70.58 | 74.97 |
| | CIDER | 13.38 | 96.69 | 93.30 | 93.72 | 49.04 | 81.70 | 73.46 | 75.60 |
| | NPOS | 15.22 | 96.27 | 92.79 | 94.33 | 69.09 | 77.14 | 61.01 | 74.94 |
| | PALM | 96.71 | 54.06 | 34.86 | 10.00 | 95.97 | 42.63 | 31.44 | 1.00 |
| | **HamOS(ours)** | **11.51** | **96.95** | **94.61** | **94.50** | **48.93** | **82.19** | **74.35** | **75.99** |

## G  DISCUSSIONS

**HamOS vs. Outlier Exposure.** Recall that our proposed HamOS generates virtual outliers solely based on the ID training data, while outlier exposure methods usually employ a large pool of natural outliers for regularization. For example, the Tiny-ImageNet (Le & Yang, 2015) is commonly used as the auxiliary dataset with CIFAR10/100 as ID datasets. The requirement for an auxiliary natural outlier dataset that can be even bigger than the ID dataset excessively undermines the feasibility of OE methods. Besides, the quality of the outliers is crucial for specific ID datasets, which should be diverse and informative, as demonstrated in previous works (Chen et al., 2021; Ming et al., 2022; Zhu et al., 2023b; Jiang et al., 2024a). Superior to OE methods, our HamOS framework can synthesize diverse and representative outliers without any requirement for additional datasets, enjoying significant flexibility and efficiency.

**HamOS vs. Contrastive-based Methods.** While the contrastive-based methods (Winkens et al., 2020; Tack et al., 2020; Sehwag et al., 2021; Sun et al., 2022; Ming et al., 2023b; Lu et al., 2024) mainly focus on learning differentiable ID embeddings, our HamOS can additionally introduce OOD supervision signals from the areas between ID clusters where numerous potentials OOD scenarios are explored through Markov chains. The empirical analysis in Table 12 demonstrates the efficacy of HamOS against training with ID contrastive losses singularly, as well as highlights the generality of HamOS, which is compatible with more advanced learning objectives for shaping the feature space.

**HamOS vs. Synthesis-based Methods.** Lying at the core of our HamOS framework, the HMC sampling algorithm distinguishes our method from previous synthesis-based methods (Du et al., 2022b; Tao et al., 2023) by sampling from Markov chains, which is underexplored for outlier synthesis. The synthesized outliers expand a vast area in the feature space, occupying various levels of OOD-ness, thanks to the property of HMC that can generate trajectories from low-density regions to high-density regions. In addition to Figure 7, We provide additional t-SNE visualization of the synthesized outliers with CIFAR-100 as an ID dataset in Figure 10. While VOS tends to generate erroneous outliers that lie within ID clusters and NPOS may generate gathered outliers that lack diversity, the outliers generated by our HamOS stretch through a vast area in the feature space without intruding on the ID clusters, which significantly enhances the model's detection capability. Here we analyze the advantages of HMC compared to traditional Gaussian sampling:

- HMC samples diverse and representative outliers. As a well-studied insight, the quality of outliers substantially affects the model's detection performance (Hendrycks et al., 2022), with diverse and representative outliers being demanded (Zhu et al., 2023b; Jiang et al., 2024a). HMC can traverse a large area in the latent space from low OOD-ness region

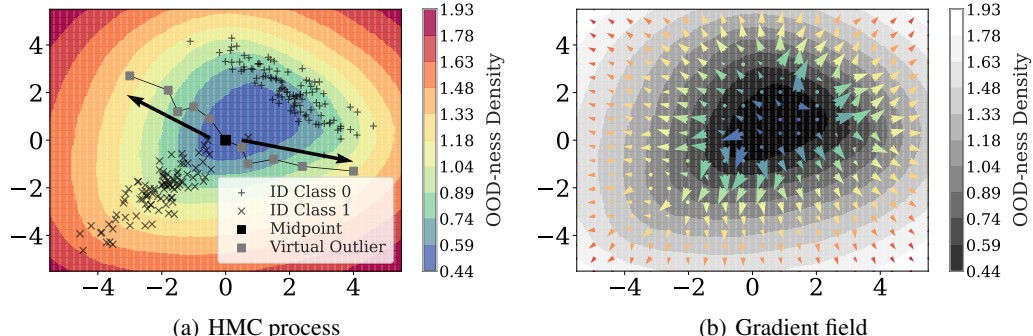

Figure 11: **Geometry analysis of the OOD-ness density estimation:** (a) the sampling process of HMC with the OOD-ness density;(b) the gradient field of the OOD-ness density.

to high OOD-ness region, thus generating diverse outliers representing different levels of OOD characteristics. However, Gaussian sampling, limited by its variance, always generates samples concentrated to the mean point, prohibiting its ability to explore the hyperspherical space.

- HMC samples more informative outliers compared to Gaussian sampling. With our crafted OOD-ness estimation, new outliers are generated with the gradient guidance, which explicitly points out the direction of the critical region that contains unknown OOD characteristics. The Gaussian sampling, however, stretches out to every direction evenly, resulting in generating outliers that may be less informative. The mix of outliers evenly distributed in every direction can hardly convey critical OOD characteristics to help the model learn differentiable representations.

**HamOS vs. Traditional Score-based Sampling.** HamOS displays numerous advantages compared to traditional score-based sampling algorithms, like score matching (Hyvärinen, 2005), Langevine dynamics (Welling & Teh, 2011), and diffusion models (Song et al., 2021), demonstrating the efficacy and efficiency of the framework. The advantages are outlined below:

- HMC exhibits a high sampling rate with low computational burden. Conventional score-based sampling may introduce extra learnable parameters which can increase the computational load. HMC can leverage the informative gradients of the potential energy function to guide the sampling process with Leapfrog integration for exploring a large area in the latent space, handling the complex target distribution in high dimensions. The direct guidance from the gradients results in a high acceptance rate of nearly 1.0, thereby keeping the computational burden practical. Parallel matrix operations can enhance the efficiency of the HMC sampling process.

- The estimation of the target unknown distribution enables the direct synthesis of OOD samples in the latent space. The objective of traditional score-based sampling is typically to converge to the target distribution guided by the ID training data, e.g., diffusion models are supposed to generate legitimate data, prohibiting the ability to generate samples from unknown distributions. HamOS, however, aims to collect diverse samples along the Markov chains, not requiring iterative sampling to converge. While other sampling algorithms struggle with the complex unknown distribution, such as Gaussian sampling that evenly explores all directions, HMC leverages explicit guidance to find OOD characteristics that can be directly employed to optimize the model for better detection performance.

- HamOS significantly exploits the information of ID priors for outlier synthesis by the class-conditional sampling between ID clusters. Traditional score-based sampling incorporates label information with the conditioned score model, which acts as a black box. In HamOS, with the OOD-ness estimation, outliers are explicitly sampled using the ID priors. Specifically, in the high-dimensional spherical space, midpoints of ID clusters serve as straightforward starting points of the Markov chains, guaranteeing the correctness of OOD synthesis. As the outliers are synthesized purely based on the ID data, HamOS has great generality to any dataset.

**On the design of OOD-ness density estimation.** We want to estimate the OOD-ness (or the likelihood to be OOD), indicating the probability of a sample from unknown distributions, based solely on the ID data. We start by defining the OOD-ness of a sample corresponding to a singular ID cluster in Equ (1). As we want to generate critical outliers between ID clusters, we further average the OOD-ness of a sample from a pair of ID clusters. This definition of OOD-ness suffices to calculate the potential energy, the gradient of which is employed to direct the sampling of Markov chains among the ID cluster pairings. Multiple Markov chains are generated within different ID cluster pairs, ultimately collected as a diverse mini-batch of outliers. The OOD-ness is measured solely by the $k$-NN distance without the assumption of any external factors. The micro-operation of HMC generates new samples with higher OOD-ness in each Markov chain, macroly leading to a diverse set of outliers that have varying degrees of OOD characteristics. We conduct further analysis of the OOD-ness density's geometry in Figure 11. As shown in Figure G, the Markov chains traverse from regions of low OOD-ness density to regions of high OOD-ness density, collecting a broad spectrum of outliers with diverse OOD characteristics. To delve deeper into the mechanism of the OOD-ness density, we plot the gradient field in Figure G, where the directions of arrows represent the direction of the density gradients and the scale of arrows represent the normalization of the density gradients. The gradient field further clarifies that points with high OOD-ness density tend to be proposed in the Markov chains. However, we impose no restrictions to the design of the OOD-ness density estimation, despite that KNN-based density already displays superior efficacy. More estimations of OOD-ness desnity can be explored; for example, the reciprocal of the ID probability density shares a similar geometry of our OOD-ness density. The flexibility of the OOD-ness density estimation design further highlights the generality of our proposed HamOS.

**Why does HamOS incorporate Cross-entropy loss?** Recall that the training objective of HamOS consists of three terms: the Cross-entropy loss, the ID contrastive loss, and the OOD discernment loss. The OOD detection requires the framework to handle the dual-task scheme, where the original classification task is managed by the classification head trained with CE loss, and the OOD detection task is dealt with by the projection head that maps the features onto the unit hypersphere for better ID-OOD differentiation, to which end we adopt the general framework of OE method (i.e., introduced in Appendix C.1). Our empirical results show that HamOS maintains ID accuracy significantly . In Table 1 and Table 2, in addition to achieving superior detection performance, HamOS also displays better ID classification performance than almost all regularization-based methods. HamOS manages to preserve the ID prediction ability by two components:

- The cross-entropy loss is adopted along with the fine-tuning phase to balance the optimization on the contrastive loss and the OOD discernment loss in Equation 8. This joint training scheme reaches a good equilibrium between the original classification task and OOD detection.

- The original classification head is preserved, which not only maintains the ID accuracy but also makes the HamOS framework compatible with many other post-hoc detection methods that manipulate features, logits, or softmax scores. As the classification and the detection are separated, they can rarely impair each other.

**On the quality of synthesized OOD samples.** The primary focus of this paper is to boost the OOD detection performance through outlier synthesis. We formalize the novel framework HamOS which models the synthesis process as Markov chains and generates virtual outliers with the guidance of the innovative potential energy function. With this in mind, the quality of the generated OOD samples is directly reflected by the detection performance and partially highlighted by the OOD scores in Figure 2, 4(b), 4(c). High variance of the OOD scores indicates diverse OOD characteristics that the generated samples have.

In previous works (Zhu et al., 2023b; Jiang et al., 2024a), task-specific metrics are adopted to validate the quality of outliers. The quality of synthesized samples may also affect ID performance. For example, in Pooladzandi et al. (2023), a subset of synthesized data is selected by CRAIG (Mirzasoleiman et al., 2020) and samples with low entropy are eliminated by an entropic filter, making the quality of the synthesized samples high enough to improve ID-ACC. There may be techniques that can be designed to boost the classification performance as well as the detection performance.

**Application of HamOS on other data modalities.** As HamOS only assumes a pretrained network that transforms the original high-dimensional raw data to the representation format that is more suit-

able for various tasks, such as classification, clustering, and generation. In NLP, numerous tasks involve learning the dense vector of embeddings, e.g., BERT and GPT produce contextual word embeddings for classification or autoregressive generation. Sentence-level or document-level representation learning is also practical in tasks like semantic search or document classification. Therefore, the HamOS framework can be adopted in many NLP tasks that require dense representation learning. In other modalities, such as graph-structured data or point-cloud data, HamOS can also be adopted since the data should be transformed into dense vector representation for various tasks, such as molecular structure modeling, protein design, and climate prediction. The superior performance of HamOS on the image data has been confirmed in this paper; nonetheless, its efficacy on various data modalities requires further investigation in future studies.

## H  BROADER IMPACT

**Cross-pollination opportunities.** The synthesis process of HamOS has many components analogous to score-based generative models, presenting significant prospects for cross-pollination. Here we outline some potentials for future research:

- The gradient score can be produced by a hierarchical model instead of the $k$-NN metric. The fundamental process of generative models is to turn a noise point into a legitimate point in an iterative manner, in which a score model is adopted to predict the gradient without sampling directly from the target distribution. Analogously, we can develop a score model, that may integrate the information from ID priors to produce gradients as better guidance for the outlier synthesis in each iteration. But this may introduce additional parameters, increasing the computational cost.

- Both HamOS and diffusion models adopt an iterative synthesis process which can be controlled by scheduling strategies. It is feasible to apply a schedule that can adjust dynamically based on the progress of outlier synthesis, which may potentially improve the quality of outliers by regulating the gradient scale. There are numerous schedules to choose from, such as linear, cosine, or any custom configuration. The adaptive scheduling strategy may also improve the versatility and adaptability of HamOS, accommodating diverse datasets, tasks, and data modalities.

- Additional techniques of generative models, such as classifier-free guidance, offer diverse research directions for HamOS. The classifier-free guidance technique is primarily adopted to introduce guidance information directly into the diffusion process without training an additional classifier. In HamOS, classifier-free guidance may help in lightweightly distilling external OOD information to the synthesis process of OOD samples, leading to a better balance between diversity and representativeness, thus elevating the quality of OOD samples.

- Conversely, the HamOS framework may serve as a beneficial prior work for improving the generative models. The non-parametric synthesis process may inspire more innovation in exploring and exploiting the latent space, leading to a high-quality, controllable, and efficient diffusion process. As HamOS facilitates learning a proper representation that is ID-OOD differentiable, there is potential to improve the fidelity of generation by shaping the latent space for diffusion models.

**Impact on foundation models.** Despite the substantial performance of foundation models using many data modalities, such as GPT, CLIP, and diffusion models, their trustworthiness and reliability remain underexplored. With the wide adoption of foundation models, the safe deployment and application of them are becoming more critical. The uncertainty estimation of the foundation models has attracted great attention, exploring how to make the outputs of the models more trustworthy (Jiang et al., 2024b; Zhu et al., 2024; Lee et al., 2024; Zhang et al., 2024).

OOD detection is becoming more essential as the foundation models are employed in increasing safety-critical applications, such as healthcare, finance, autonomous driving, and cybersecurity. The failure of self-uncertainty estimation may lead to dangerous outputs or risky decisions (Shi et al., 2024), as foundation models' overconfidence is not yet overcome (Groot & Valdenegro Toro, 2024). OOD detection acts as an indicator of when the LLMs/VLMs should be trusted, with erroneous

outputs, that don't align with training data, being detected before being displayed to users, such as hallucination detection (Farquhar et al., 2024) and unsolvable problem detection (Miyai et al., 2024).

HamOS, as a general framework, offers a new perspective to augment the training data with negative samples, hence improving the robustness and helping the model produce more proper representations. Recall that HamOS assumes a pretrained backbone network that projects the input data to the latent space, which aligns with typical representation learning schemes, endowing the nature of compatibility to the training of foundation models. Empirical analysis on small-to-large scale benchmarks has been done to validate the generality of the HamOS system. As vision-based tasks are widely deployed in real-world applications, such as autonomous driving and medical image diagnosis, it is more common to encounter OOD scenarios, henceforth we assess the efficacy and efficiency of HamOS on image tasks. This doesn't restrict the application of HamOS in other domains, such as NLP and graphs, which also adopt the representation learning paradigm.

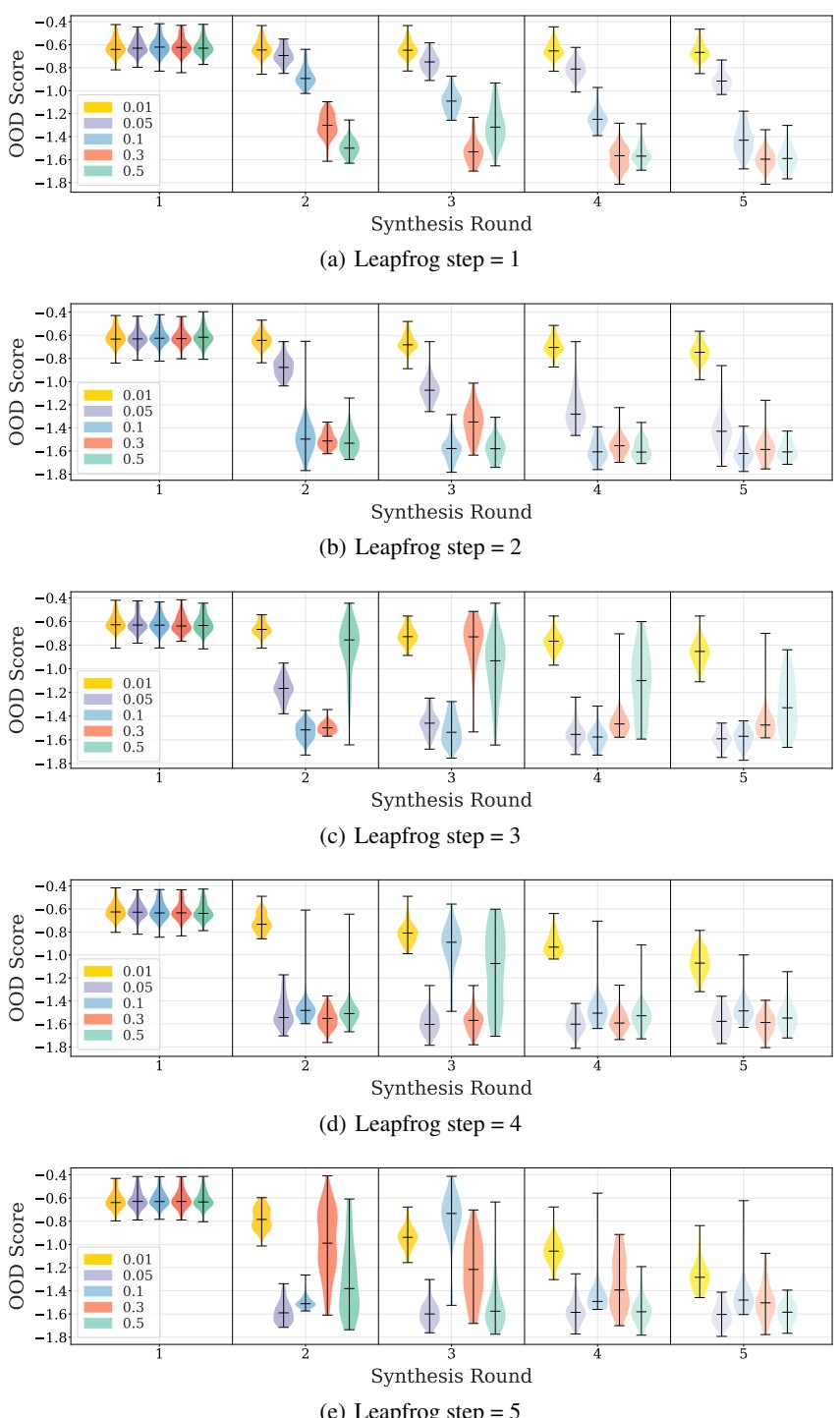

Figure 12: **Round-wise analysis on CIFAR-10 with different Leapfrog steps and step sizes:** the outliers are synthesized with different Leapfrog steps (i.e., $\{1, 2, 3, 4, 5\}$) and different step sizes (i.e., $\{0.01, 0.05, 0.1, 0.3, 0.5\}$).

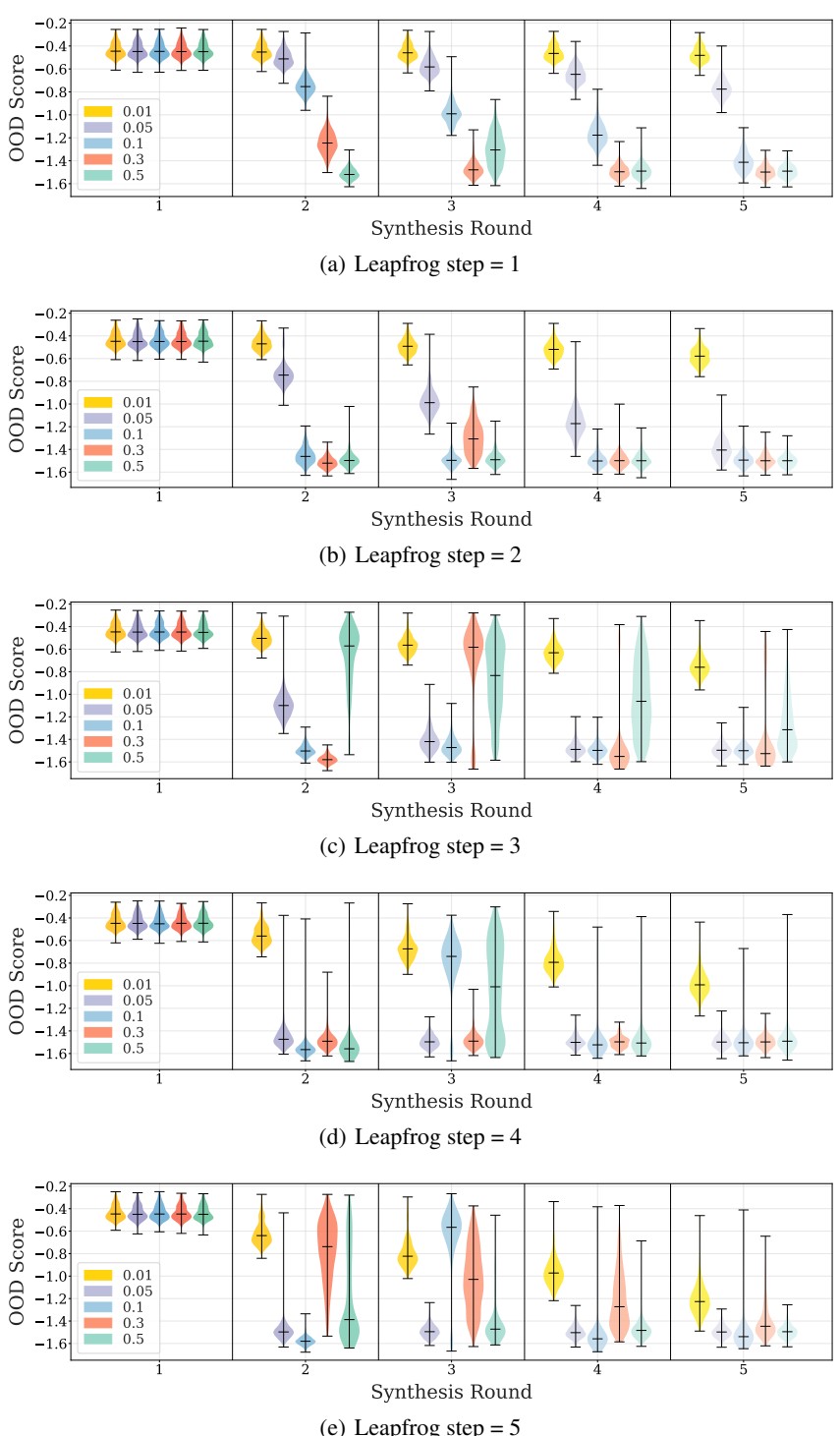

Figure 13: **Round-wise analysis on CIFAR-100 with different Leapfrog steps and step sizes:** the outliers are synthesized with different Leapfrog steps (i.e., $\{1, 2, 3, 4, 5\}$) and different step sizes (i.e., $\{0.01, 0.05, 0.1, 0.3, 0.5\}$).

