# OpenReview forum: "Outlier Synthesis via Hamiltonian Monte Carlo for Out-of-Distribution Detection"
_ICLR.cc/2025/Conference — ICLR 2025 Poster_

### Official Review · Reviewer_by3L · 2024-10-31

**Soundness:** 2
**Presentation:** 3
**Contribution:** 3
**Rating:** 5
**Confidence:** 5

**Summary:**

The paper proposes a novel framework called Hamiltonian Monte Carlo Outlier Synthesis (HamOS) for out-of-distribution (OOD) detection. The primary goal of HamOS is to address the limitations of current outlier synthesis approaches, which often rely on either extensive collections of natural OOD data or generative models that are computationally demanding.

HamOS uses Hamiltonian Monte Carlo (HMC) to generate diverse and representative virtual outliers directly from the feature space of in-distribution (ID) data. Unlike previous methods that use Gaussian sampling or generative models, HamOS utilizes Markov chains to sample new virtual outliers by traversing a broad area of the feature space. The process ensures that the synthesized outliers are well-distributed and possess varying degrees of OOD characteristics, which helps in improving OOD detection capabilities.

The framework is evaluated on standard benchmarks like CIFAR-10, CIFAR-100, and ImageNet-1K, and shows significant improvements in OOD detection metrics compared to state-of-the-art baselines. The proposed HamOS achieves a balance between diversity and representativeness of synthesized outliers.

**Strengths:**

1. Using HMC for generating diverse virtual outliers is a creative adaptation of a well-known sampling technique in statistics and machine learning. Unlike the conventional generative or Gaussian-based sampling methods, this approach allows the model to extensively traverse the latent feature space, thus creating diverse and representative OOD samples. This unique utilization of Markov Chain Monte Carlo (MCMC) methods for OOD detection offers an innovative approach that is distinct from prior generative and noise-based sampling strategies.

2. The concept of OOD-ness estimation to guide the sampling process is also a good addition. By focusing on hyperspherical embeddings and using distance metrics to determine the likelihood of being OOD, the paper contributes a perspective on how to quantify and synthesize OOD characteristics in feature space.

**Weaknesses:**

1. The paper's claims regarding the novelty of using Markov chains for outlier synthesis are overstated. Specifically, the statement in the first contribution on page 1 claims that outlier synthesis using Markov chains is a "new paradigm" for OOD detection, which is misleading. Markov chains are a well-established approach in sampling, and the application to outlier synthesis does not represent an entirely new paradigm, but rather a new technique within an existing category.

2. In the current era, large language models (LLMs) and multimodal foundational models have demonstrated remarkable performance in numerous domains, including computer vision, text understanding, and scientific knowledge discovery. The paper only evaluates HamOS on natural image datasets (e.g., CIFAR-10, CIFAR-100, ImageNet-1K), which raises concerns about its significance and practical applicability in other fields.

3. The concept of "OOD-ness" is central to the proposed approach, but its definition remains unclear. Specifically, it is not evident how OOD-ness is formally quantified or measured, which complicates understanding how diverse outliers are synthesized. There is a lack of clarity regarding whether OOD-ness is defined solely based on distance metrics (such as k-nearest neighbors) or if other factors are considered.

4. The advantages of Hamiltonian Monte Carlo (HMC) over other sampling methods, such as Gaussian sampling or Random Walk MCMC, are not sufficiently discussed. The reader is left wondering why HMC is particularly suitable for OOD detection and how it performs better than other, potentially simpler, methods.

5. In Table 3, HamOS shows low inter-class dispersion, which is concerning as it suggests poor separation between different ID classes in the feature space. This could potentially affect the performance of the method on in-distribution (ID) classification, as class separation is critical for robust classification.

**Questions:**

1. Could you provide additional context that clearly distinguishes this work from existing generative or sampling-based methods for OOD detection? Specifically, how does your approach go beyond being a new method and qualify as a new paradigm?

2. In the era of large language models (LLMs) and multimodal foundational models, what role do you envision for OOD detection, especially for image data? How does your proposed method relate to or complement the capabilities of LLMs?

3. Can the proposed HamOS be adapted to other domains, such as text (e.g., natural language processing) or scientific discovery? If feasible, could you elaborate on the generalization of HamOS to these other types of data? Providing a discussion or a hypothetical example would help establish the broader applicability of your work.

4. How does HMC sampling compare to other sampling methods, such as Gaussian sampling or Random Walk MCMC, in the context of OOD detection? Can you provide either a theoretical or empirical comparison to justify the choice of HMC?

---

> ### Author Response · Authors · 2024-11-19
> **Response to Reviewer by3L (1/6)**
>
> Thank you for your time devoted in reviewing this paper and your insightful comments. We are very grateful that you find our method **creative**, **unique**, and **innovative**. Please find our response to your concerns below:
>
> ## 1. What qualifies HamOS as a new paradigm. [W1, Q1]
>
> **A1**: Thank you for pointing out the potential for misunderstanding. Here we provide an additional explanation that clearly distinguishes our method from previous methods.
>
> To the best of our knowledge, almost all previous works that synthesize outliers either adopt parameter-heavy generative models or Gaussian sampling.
>
> For generative model-based methods, some works [1,2,3,4,5] train an additional generative model (e.g., GAN) to produce outliers, whereas others [6,7,8] leverage the off-the-shelf pretrained diffusion model. The introduction of generative models significantly increases the computational load, impractical for large-scale scenarios such as ImageNet-1K. The heavy computational cost attributes the synthesized outliers to be less diverse and representative.
>
> For Gaussian sampling-based approaches [9,10,11], the general idea is to obtain samples that are in low-likelihood regions, making applying Gaussian noise to boundary ID samples a common practice. However, Gaussian sampling, limited by its variance, generates samples concentrated around the mean point, prohibiting its ability to explore the hyperspherical space. The symmetry property of Gaussian sampling makes it stretch out in every direction evenly, lacking guidance information.
>
> HamOS offers a brand-new perspective to outlier synthesis by modeling the synthesis process as Markov chains, serving as a novel framework that generates outliers based solely on the ID data. Here are the two main innovations:
>
> - HamOS provides an innovative estimation of the OOD likelihood (Equation 1 & 2). The gradient of the potential energy function offers explicit guidance for generating outliers that differ from previous ones. Generative models can also generate outliers with specific objectives; nevertheless, they necessitate substantial additional training costs, which restrict their application in resource-constrained situations. Gaussian sampling-based methods can only generate outliers around ID boundary samples, without explicitly exploring the complex latent space for diverse OOD characteristics.
>
> - HamOS adopts the Markov chain Monte Carlo process which enables it to traverse a wide range of regions in the latent space. By meandering with the guidance of the potential energy function, outliers with different OOD characteristics are collected to form the diverse and representative mini-batch. The diversity and representativeness of outliers in turn significantly reduce the amount of needed samples, e.g., in default configurations, HamOS only generates 20 outliers per class, whereas VOS and NPOS generate 200 and 400 per class respectively, demonstrating the HamOS's efficiency.
>
> We updated this discussion in the revision. We also revise "new paradigm" to "new framework" for clarity.

---

> ### Author Response · Authors · 2024-11-19
> **Response to Reviewer by3L (2/6)**
>
> ## 2. Significance of HamOS to LLMs and multimodal foundation models. [W2, Q2]
>
> **A2**: Brilliant point! We provide a comprehensive overview of the relations between HamOS and the foundation models and how HamOS can assist in solving problems in other domains.
>
> Despite the substantial performance of foundation models using many data modalities, such as GPT, CLIP, and diffusion models, their trustworthiness and reliability remain underexplored. With the wide adoption of foundation models, the safe deployment and application of them are becoming more critical. The uncertainty estimation of the foundation models has attracted great attention, exploring how to make the outputs of the models more trustworthy [12,13,14].
>
> OOD detection is crucial for many aspects of foundation models. In the phase of data collection, it is essential to identify samples from unknown distributions among the extensive web-sourced data to ensure that dangerous content is filtered out rather than being fed into the model. The remarkable performance of foundation models significantly relies on high-quality pretraining data which is acquired through preprocessing raw data by OOD detectors. During the training period, robust training schemes from the OOD detection community can be adopted to enhance the model's robustness and generalization to new distributions and domains. During inference, toxic input from users should also be blocked out via OOD detection to prevent any induced poison output from the model.
>
> HamOS, as a general framework, offers a new perspective to augment the training data with negative samples, hence improving the robustness and helping the model produce more proper representations. Recall that HamOS assumes a pretrained backbone network that projects the input data to the latent space, which aligns with typical representation learning schemes, endowing the nature of compatibility to the training of foundation models. Empirical analysis on small-to-large scale benchmarks has been done to validate the generality of the HamOS system. As vision-based tasks are widely deployed in real-world applications, such as autonomous driving and medical image diagnosis, it is more common to encounter OOD scenarios, henceforth we assess the efficacy and efficiency of HamOS on image tasks. This doesn't restrict the application of HamOS in other domains, such as NLP and graphs, which also adopt the representation learning paradigm.
>
> All in all, HamOS may act as a general training framework for the safety and reliability of foundation models. Thanks again for bringing up such an inspiring point! We updated this discussion in the revision.

---

> ### Author Response · Authors · 2024-11-19
> **Response to Reviewer by3L (3/6)**
>
> ## 3. Applicability of HamOS in other fields, such as text, scientific discovery. [W2, Q3]
>
> **A3**: Another great point! As HamOS only assumes a pretrained network that transforms the original high-dimensional raw data to the representation format that is more suitable for various tasks, such as classification, clustering, and generation. In NLP, numerous tasks involve learning the dense vector of embeddings, e.g., BERT and GPT produce contextual word embeddings for classification or autoregressive generation. Sentence-level or document-level representation learning is also practical in tasks like semantic search or document classification. Therefore, the HamOS framework can be adopted in many NLP tasks that require dense representation learning. In other modalities, such as graph-structured data or point-cloud data, HamOS can also be adopted since the data should be transformed into dense vector representation for various tasks, such as molecular structure modeling, protein design, and climate prediction. The superior performance of HamOS on the image data has been confirmed in this paper; nonetheless, its efficacy on various data modalities requires further investigation in future studies. We included this discussion in the revision for the general applicability of HamOS.
>
> ## 4. The definition of "OOD-ness" is unclear. [W3]
>
> **A4**: We are glad you mentioned the potential misunderstanding of the OOD-ness definition! Here we provide clearer explanation about the definition of OOD-ness.
>
> We want to estimate the OOD-ness (or the likelihood to be OOD), indicating the probability of a sample from unknown distributions, based solely on the ID data. We start by defining the OOD-ness of a sample corresponding to a singular ID cluster in Equation 1. As we want to generate critical outliers between ID clusters, we further average the OOD-ness of a sample from a pair of ID clusters. This definition of OOD-ness suffices to calculate the potential energy, the gradient of which is employed to direct the sampling of Markov chains among the ID cluster pairings. Multiple Markov chains are generated within different ID cluster pairs, ultimately collected as a diverse mini-batch of outliers. The OOD-ness is measured solely by the $k$-NN distance without the assumption of any external factors. The micro-operation of HMC generates new samples with higher OOD-ness in each Markov chain,  macroly leading to a diverse set of outliers that have varying degrees of OOD characteristics. We added a clearer explanation about OOD-ness to the revision.

---

> ### Author Response · Authors · 2024-11-19
> **Response to Reviewer by3L (4/6)**
>
> ## 5. Advantages of HMC over other sampling algorithms. [W4, Q4]
>
> **A5**: We appreciate your insightful question! Here we first clarify the superiority of HMC against the Gaussian sampling; then analyze the advantages of HMC with random walk MCMC as baselines. The analysis is supported by comprehensive experiments.
>
> HMC sampling outperforms Gaussian sampling for mainly two reasons:
>
> 1. HMC samples diverse and representative outliers. As a well-studied insight, the quality of outliers substantially affects the model's detection performance [15], with diverse and representative outliers being demanded [16,17]. HMC can traverse a large area in the latent space from low OOD-ness region to high OOD-ness region, thus generating diverse outliers representing different levels of OOD characteristics. However, Gaussian sampling, limited by its variance, always generates samples concentrated to the mean point, prohibiting its ability to explore the hyperspherical space.
>
> 2. HMC samples more informative outliers compared to Gaussian sampling. With our crafted OOD-ness estimation, new outliers are generated with the gradient guidance, which explicitly points out the direction of the critical region that contains unknown OOD characteristics. The Gaussian sampling, however, stretches out to every direction evenly, resulting in generating outliers that may be less informative. The mix of outliers evenly distributed in every direction can hardly convey critical OOD characteristics to help the model learn differentiable representations.
>
> The comprehensive empirical analysis shows that HMC exhibits improved performance than Gaussian sampling adopted by VOS and NPOS in Table 1,2,5.
>
> Here we provide comparison between other score-based sampling algorithms. We would like to remind you that we have conducted empirical analysis of several score-based sampling algorithms in Section 4.3 with random walk sampling as the baseline. The algorithms are introduced briefly in Appendix C.3. The empirical results are shown in Figure 6(a) with the full results displayed in Table 9 in Appendix F.1.
>
> We have specifically compared 4 score-based sampling algorithms, HMC (default), MALA, mMALA, and RMHMC. The empirical results show that the proposed HamOS framework is consistently effective with different sampling algorithms compared to the random walk baseline and other OOD detection baselines. There are slight performance differences between the score-based algorithms which are mainly due to the distinct characteristics of each algorithm. Different algorithms may exhibit different behaviors under various circumstances, suggesting that HamOS can be tuned for different scenarios indicating its remarkable versatility. Moreover, Langevin dynamics-based algorithms (MALA and mMALA) are more time-efficient compared to HMC-based algorithms, since they only calculate the gradient once per round instead of leveraging the Leagfrog discretization for multi-step updates.
>
> The HMC algorithm and its variants display numerous advantages compared to traditional score-based sampling algorithms, like score matching [18], Langevine dynamics [19], and diffusion models [20], contributing to the efficacy and efficiency of the HamOS framework. The advantages are outlined below:
>
> - HMC exhibits a high sampling rate with low computational burden. HMC can leverage the informative gradients of the potential energy function to guide the sampling process with Leapfrog integration for exploring a large area in the latent space, handling the complex target distribution in high dimensions. The direct guidance from the gradients results in a high acceptance rate of nearly 1.0, thereby keeping the computational burden practical. Parallel matrix operations can enhance the efficiency of the HMC sampling process.
>
> - The meticulously crafted estimation of the target unknown distribution enables the direct synthesis of OOD samples in the latent space. While other sampling algorithms struggle with the complex unknown distribution, such as Gaussian sampling that evenly explores all directions, HMC leverages explicit guidance to find OOD characteristics that can be directly employed to optimize the model for better detection performance.
>
> - Third, HamOS significantly exploits the information of ID priors by the class-conditional sampling between ID clusters. In the high-dimensional spherical space, midpoints of ID clusters serve as straightforward anchors as the roots of the Markov chains, guaranteeing the correctness of OOD synthesis. As the outliers are synthesized purely based on the ID data, HamOS has great generality to any dataset.
>
> In conclusion, our proposed HamOS framework not only demonstrates flexibility with various score-based sampling algorithms, distinguishing its versatility but also possesses superiority compared with traditional sampling algorithms. This discussion has been incorporated into the new revision.

---

> ### Author Response · Authors · 2024-11-19
> **Response to Reviewer by3L (5/6)**
>
> ## 6. Low inter-class dispersion suggests poor ID accuracy. [W5]
>
> **A6**:  We are glad you bring up such an inspiring question. The inter-class dispersion only serves as a restricted viewpoint for analysing the differentiability between ID clusters. The most straightforward way to validate the ID classification performance is the ID-ACC.
>
> The inter-class dispersion doesn't display a direct correlation with the ACC. For example, CIDER outstrips SSD+ by a large margin on the inter-class dispersion (seen in Table 3) while attaining approximately equivalent ID accuracy as SSD+(see to Table 1). Although HamOS doesn't exhibit the highest inter-class dispersion, it indeed surpassess all the methods compared in Table 3 by ID-ACC.
>
> The inter-class dispersion doesn't exhibit a direct correlation with the OOD performance. In the hyperspherical space, the ID-OOD separation acts as a direct metric for assessing the performance of OOD detection, wherein HamOS significantly outperforms the baselines. Recall that the inter-class dispersion is optimized by the contrastive loss term in Equation 8. The contrastive loss and the OOD discernment loss act synergistically to learn an appropriate representation that exhbits enhanced ID-OOD differentiability, while maintaining sufficient inter-class dispersion for detection.
>
> Here we provide further discussion about the relation between ID classification and OOD detection. In the context of OOD detection, the ID accuracy should also be preserved. We managed to enhance the detection performance without sacrificing the classification capability by mainly two components.
>
> - The cross-entropy loss is adopted along with the fine-tuning phase to balance the optimization on the contrastive loss and the OOD discernment loss in Equation 8. This joint training scheme reaches a good equilibrium between the original classification task and OOD detection.
>
> - The original classification head is preserved, which not only maintains the ID accuracy but also makes the HamOS framework compatible with many other post-hoc detection methods that manipulate features, logits, or softmax scores. As the classification and the detection are separated, they can rarely impair each other.
>
> Our empirical results show that HamOS maintains ID accuracy significantly. In Table 1 and Table 2, in addition to achieving superior detection performance, HamOS also displays better ID classification performance than almost all regularization-based methods. The empirical study demonstrates the superiority of our 'Y-head' HamOS framework.
>
> Thanks again for such constructive comments. We incorporated the discussion in the revision.

---

> ### Author Response · Authors · 2024-11-19
> **Response to Reviewer by3L (6/6)**
>
> [1] Kimin Lee, Honglak Lee, Kibok Lee, and Jinwoo Shin. Training confidence-calibrated classifiers for detecting out-of-distribution samples. In Proceedings of the 6th International Conference on Learning Representations, 2018.
>
> [2] Kumar Sricharan and Ashok Srivastava. Building robust classifiers through generation of confident out of distribution examples. In Advances in Neural Information Processing Systems 31 workshop on bayesian deep learning, 2018.
>
> [3] Sachin Vernekar, Ashish Gaurav, Vahdat Abdelzad, Taylor Denouden, Rick Salay, and K. Czarnecki. Out-of-distribution detection in classifiers via generation. In Advances in Neural Information Processing Systems 32 workshop on safety and robustness in decision making, 2019.
>
> [4] Felix Moller, Diego Botache, Denis Huseljic, Florian Heidecker, Maarten Bieshaar, and Bernhard Sick. Out-of-distribution detection and generation using soft brownian offset sampling and autoencoders. In Proceedings of the 34th IEEE Conference on Computer Vision and Pattern Recognition Workshops, pp. 46–55, 2021.
>
> [5] Masoud Pourreza, Bahram Mohammadi, Mostafa Khaki, Samir Bouindour, Hichem Snoussi, and Mohammad Sabokrou. G2d: Generate to detect anomaly. In Proceedings of the IEEE/CVF Winter Conference on Applications of Computer Vision, pp. 2002–2011, 2021.
>
> [6] Xuefeng Du, Yiyou Sun, Jerry Zhu, and Yixuan Li. Dream the impossible: Outlier imagination with diffusion models. In Advances in Neural Information Processing Systems 36, 2023.
>
> [7] Jiahui Liu, Xin Wen, Shizhen Zhao, Yingxian Chen, and Xiaojuan Qi. Can ood object detectors learn from foundation models? In Proceedings of the 18th European Conference on Computer Vision, 2024.
>
> [8] Jiankang Chen, Ling Deng, Zhiyong Gan, Wei-Shi Zheng, and Ruixuan Wang. Fodfom: Fake outlier data by foundation models creates stronger visual out-of-distribution detector. In Proceedings of the 32nd ACM International Conference on Multimedia, pp. 1981–1990, 2024.
>
> [9] Xuefeng Du, Zhaoning Wang, Mu Cai, and Yixuan Li. Vos: Learning what you don’t know by virtual outlier synthesis. Proceedings of the 10th International Conference on Learning Representations, 2022.
>
> [10] Leitian Tao, Xuefeng Du, Jerry Zhu, and Yixuan Li. Non-parametric outlier synthesis. In Proceedings of the 11th International Conference on Learning Representations, 2023.
>
> [11] Hossein Mirzaei and Mackenzie W. Mathis. Adversarially robust out-of-distribution detection using lyapunov-stabilized embeddings. arXiv preprint arXiv:2410.10744, 2024.
>
> [12] Xue Jiang, Feng Liu, Zhen Fang, Hong Chen, Tongliang Liu, Feng Zheng, and Bo Han. Negative label guided OOD detection with pretrained vision-language models. In Proceedings of the 12th International Conference on Learning Representations, 2024.
>
> [13] Armando Zhu, Jiabei Liu, Li Keqin, Shuying Dai, Bo Hong, Peng Zhao, and Changsong Wei. Exploiting diffusion prior for out-of-distribution detection. Irish Interdisciplinary Journal of Science Research, 08:171–185, 2024.
>
> [14] Seulbi Lee, Jihyo Kim, and Sangheum Hwang. Reflexive guidance: Improving oodd in vision-language models via self-guided image-adaptive concept generation. arXiv preprint arXiv:2410.14975, 2024.
>
> [15] Dan Hendrycks, Steven Basart, Mantas Mazeika, Andy Zou, Joseph Kwon, Mohammadreza Mostajabi, Jacob Steinhardt, and Dawn Song. Scaling out-of-distribution detection for real-world settings. In Proceedings of the 39th International Conference on Machine Learning, volume 162, pp. 8759–8773, 2022.
>
> [16] Jianing Zhu, Geng Yu, Jiangchao Yao, Tongliang Liu, Gang Niu, Masashi Sugiyama, and Bo Han. Diversified outlier exposure for out-of-distribution detection via informative extrapolation. In Advances in Neural Information Processing Systems 36, 2023.
>
> [17] Wenyu Jiang, Hao Cheng, MingCai Chen, Chongjun Wang, and Hongxin Wei. DOS: Diverse outlier sampling for out-of-distribution detection. In Proceedings of the 12th International Conference on Learning Representations, 2024.
>
> [18] Aapo Hyvärinen. Estimation of non-normalized statistical models by score matching. Journal of Machine Learning Research, 6(24):695–709, 2005.
>
> [19] Max Welling and Yee Whye Teh. Bayesian learning via stochastic gradient langevin dynamics. In Proceedings of the 28th International Conference on Machine Learning, pp. 681–688, 2011.
>
> [20] Yang Song, Jascha Sohl-Dickstein, Diederik P Kingma, Abhishek Kumar, Stefano Ermon, and Ben Poole. Score-based generative modeling through stochastic differential equations. In Proceedings of the 9th International Conference on Learning Representations, 2021.

---

> ### Author Response · Authors · 2024-11-22
> **Kind reminder to Reviewer by3L**
>
> Dear Reviewer by3L,
>
> We sincerely appreciate your valuable comments! We understand that you may be too busy to check our rebuttal. We discussed the HamOS's significance to foundation models (see Appendix H) and the potential to apply HamOS to other data modalities (see Appendix G). We also provided the advantages of HamOS compared to other sampling algorithms (see Appendix G). The discussion about inter-class dispersion (see Section 4.2 and Appendix F.1) and the definition of "OOD-ness" is also detailed.
>
> We sincerely hope that the refinement can address your concerns. Please let us know if you still have any further concerns and we will be open to all possible discussions.
>
> Best regards,
>
> Authors of Submission 2198

---

> > ### Comment · Reviewer_by3L · 2024-11-23
> >
> > Dear authors,
> >
> > Thank you for your comprehensive response. Regarding foundation models, my main concern lies in the practical significance of Out-of-Distribution (OOD) detection. As we know, large language models (LLMs) such as GPT-4 and Gemini have been trained on vast and diverse datasets, covering nearly all classes of data across the world. In this context, the relevance of OOD detection becomes less meaningful, as these models are capable of generalizing to a wide range of inputs. Since they can handle novel or unseen scenarios with remarkable proficiency, the need for explicit OOD detection may no longer be as critical as it once was in earlier machine learning paradigms.

---

> > > ### Author Response · Authors · 2024-11-24
> > > **Response to further concern (1/2)**
> > >
> > > Thank you for your insightful comments regarding the practical significance of Out-of-Distribution (OOD) detection in the context of foundation models.
> > >
> > > Despite achieving remarkable performance, foundation models are not infallible as they still lack robustness when operating with new domains. Foundation models, trained at scale on huge data collections, provide high-quality representations that generalize well to a wide range of inputs. However, we don't share unanimity with you on your claim that the vast dataset covers nearly all classes of data across the world. As the data distribution evolves over time, it is infeasible to generalize to the infinite future data with finite history data [1]. An explicit OOD detector serves as a filter that identifies data the current model can handle correctly.
> > >
> > > OOD detection is becoming even more essential as the foundation models are employed in increasing safety-critical applications, such as healthcare, finance, autonomous driving, and cybersecurity. The failure of self-uncertainty estimation may lead to dangerous outputs or risky decisions [3], as foundation models' overconfidence is not yet overcome [4]. OOD detection acts as an indicator of when the LLMs/VLMs should be trusted, with erroneous outputs, that don't align with training data, being detected before being displayed to users, such as hallucination detection [5] and unsolvable problem detection [6].
> > >
> > > As the data distributions evolve over time, OOD detection plays a crucial role in building complex LLM/VLM systems, triggering continual learning when necessary [7, 8]. The foundation models should move forward to adapt to new domains, for their knowledge of previous data is soon running out of date. Detecting unknown classes is a critical pre-step for the foundation models before training on the new data collections. Moreover, OOD detection is especially important for avoiding unreliable outputs in many downstream tasks that require post-training on specific data that may deteriorate the model's generality due to catastrophic forgetting [9].
> > >
> > > HamOS can be incorporated into the complicated LLM/VLM systems for robust training as a general framework that augments the training data with virtual negative samples. Recall that HamOS requires only a pretrained backbone that transforms the raw data into representations, which perfectly aligns with the foundation models [10]. The non-parametric outlier synthesis, which incorporates no external information, also gives rise to the great generality of the proposed HamOS. In this paper, comprehensive empirical analysis has been conducted on small-to-large scale benchmarks to demonstrate the great scalability of HamOS.
> > >
> > > We would like to introduce a few case studies to assist you in understanding the significance of OOD detection for foundation models:
> > >
> > > - Cybersecurity [11]: Since log file analysis is a common approach to detect adversarial attacks, new, foundation models should adapt appropriately to new attack methods as new vulnerabilities are found and exploited continuously.
> > >
> > > - Healthcare [12]: Though LLMs/VLMs have been helpful medical assistants in suggesting diagnoses, new types of diseases or personalized diseases may incur over time. The foundation model ought to detect diseases that differ from previous ones.
> > >
> > > We agree that the landscape of LLM/VLM is rapidly evolving, and our work contributes to the ongoing effort to develop robust mechanisms that ensure the safe and reliable deployment of foundation models, including future generations of LLMs/VLMs. We appreciate your feedback and believe our framework offers a valuable contribution to the field. The discussion has been updated in the revision.

---

> > > ### Author Response · Authors · 2024-11-24
> > > **Response to further concern (2/2)**
> > >
> > > ## References
> > >
> > > [1] Huaxiu Yao, Caroline Choi, Bochuan Cao, Yoonho Lee, Pang Wei W Koh, and Chelsea Finn. Wild-time: A benchmark of in-the-wild distribution shift over time. In Advances in Neural Information Processing Systems 35, 2022.
> > >
> > > [2] Miao Xiong, Zhiyuan Hu, Xinyang Lu, YIFEI LI, Jie Fu, Junxian He, and Bryan Hooi. Can LLMs express their uncertainty? an empirical evaluation of confidence elicitation in LLMs. In Proceedings of the 12th International Conference on Learning Representations, 2024.
> > >
> > > [3] Liang Shi, Boyu Jiang, and Feng Guo. Scvlm: a vision-language model for driving safety critical event understanding. arXiv preprint arXiv: 2410.00982, 2024.
> > >
> > > [4] Tobias Groot and Matias Valdenegro Toro. Overconfidence is key: Verbalized uncertainty evaluation in large language and vision-language models. In Proceedings of the 4th Workshop on Trustworthy Natural Language Processing, 2024.
> > >
> > > [5] Sebastian Farquhar, Jannik Kossen, Lorenz Kuhn, and Yarin Gal. Detecting hallucinations in large language models using semantic entropy. Nature, 630(8017):625–630, 2024.
> > >
> > > [6] Atsuyuki Miyai, Jingkang Yang, Jingyang Zhang, Yifei Ming, Qing Yu, Go Irie, Yixuan Li, Hai Li, Ziwei Liu, and Kiyoharu Aizawa. Unsolvable problem detection: Evaluating trustworthiness of vision language models. arXiv preprint arXiv: 2403.20331, 2024.
> > >
> > > [7] Gyuhak Kim, Sepideh Esmaeilpour, Changnan Xiao, and Bing Liu. Continual learning based on ood detection and task masking. In Proceedings of the 35th IEEE Conference on Computer Vision and Pattern Recognition Workshops, 2022.
> > >
> > > [8] Viet Dao, Van-Cuong Pham, Quyen Tran, Thanh-Thien Le, Linh Van Ngo, and Thien Huu Nguyen. Lifelong event detection via optimal transport. In Proceedings of the Conference on Empirical Methods in Natural Language Processing, pp. 12610–12621, 2024.
> > >
> > > [9] Yun Luo, Zhen Yang, Fandong Meng, Yafu Li, Jie Zhou, and Yue Zhang. An empirical study of catastrophic forgetting in large language models during continual fine-tuning. arXiv preprint arXiv: 2308.08747, 2024.
> > >
> > > [10] Atsuyuki Miyai, Jingkang Yang, Jingyang Zhang, Yifei Ming, Yueqian Lin, Qing Yu, Go Irie, Shafiq Joty, Yixuan Li, Hai Li, Ziwei Liu, Toshihiko Yamasaki, and Kiyoharu Aizawa. Generalized out-of-distribution detection and beyond in vision language model era: A survey. arXiv preprint arXiv: 2407.21794, 2024.
> > >
> > > [11] Egil Karlsen, Xiao Luo, Nur Zincir-Heywood, and Malcolm Heywood. Benchmarking large language models for log analysis, security, and interpretation. Journal of Network and Systems Management, 32(3):59, 2024.
> > >
> > > [12] Dandan Wang and Shiqing Zhang. Large language models in medical and healthcare fields: applications, advances, and challenges. Artificial Intelligence Review, 57(11):299, 2024.

---

> > > ### Author Response · Authors · 2024-11-27
> > > **Kind notification to Reviewer by3L**
> > >
> > > Dear Reviewer by3L,
> > >
> > > We sincerely value your constructive feedback! We understand that you might be too busy to check our response. We would like to provide an overview of the changes we made based on your advice and guidance:
> > >
> > > - Discussed the significance of OOD detection to foundation models, as well as the applicability of HamOS (see Appendix H).
> > > - Discussed the potential to apply HamOS to other data modalities (see Appendix G).
> > > - Clarified the advantages of HamOS compared to other sampling algorithms (see Appendix G).
> > > - Clarified the inter-class dispersion (see Section 4.2 and Appendix F.1).
> > > - Clarified the definition of "OOD-ness" (see Appendix G).
> > >
> > > We genuinely hope that the improvement will allay your concerns. If you have any more concerns, please let us know, and we'll be happy to discuss them.
> > >
> > > Best regards,
> > >
> > > Authors of Submission 2198

---

### Official Review · Reviewer_HEF3 · 2024-11-01

**Soundness:** 3
**Presentation:** 3
**Contribution:** 2
**Rating:** 8
**Confidence:** 3

**Summary:**

This paper used Spherical HMC to generate OOD data and then from that figure out which data is ID and which data is OOD.

**Strengths:**

Uses Spherical HMC to better sample from the energy basin to sample OOD points.

Principled and theoretically sound way to generate OOD samples and consistently outperforms other methods on the compared tasks and datasets.

**Weaknesses:**

In Fig 7 TNSE HMC is working a lot better to sample the OOD regions compared to NPOS, but in Figure 10 NPOS actually seems to be performing just as well or maybe even better as Spherical HMC.

One thing that is not clear to me is how we can understand the quality/utility difference between the OOD points sampled between the two methods other than the effect of outlier detection, but this might be out of the scope of the paper.

**Questions:**

I know HMC can be a bit computationally heavy, can the authors talk about this a bit? It is not immediately obvious to me if one should care about computational load in this setting, but it would be good to discuss runtimes or theoretical cost vs other methods non the less.

Do the authors think something like Spherical RMHMC would do a better job or is Spherical HMC (SpHMC) good enough?

I am wondering if the authors have any idea about the quality of the generated samples? Can they be used to boost the ID or OOD classification accuracy of the underlying models? I ask this because there was a paper that used MCMC to sample from an EBM and then train on those samples. They found this reduced performance unless the images were of high quality. It would be interesting if the samples generated via SpHMC are of high enough quality and diverse enough to actually directly boost acc performance of the model.

[1 ]Pooladzandi, Omead, et al. "Generating High Fidelity Synthetic Data via Coreset selection and Entropic Regularization." arXiv preprint arXiv:2302.00138 (2023). https://arxiv.org/abs/2302.00138

---

> ### Author Response · Authors · 2024-11-19
> **Response to Reviewer HEF3 (1/3)**
>
> Thank you for your time devoted to reviewing this paper and your insightful comments! We are delighted that you find our method **principled** and **theoretically sound**. Please find our response to your concerns below:
>
> ## 1. NPOS seems to perform well in Figure 10. [W1]
>
> **A1**: Great observation! Here we provide further clarification about the visualization.
>
> We utilized the t-SNE technique to cast the high-dimensional OOD samples to 2-dimension and visualize them in Figure 7 and Figure 10 corresponding to CIFAR-10 and CIFAR-100 datasets respectively. The primary purpose of t-SNE visualization is to intuitively display the synthesized OOD samples' quality. Due to the loss of most characteristics during compression, the t-SNE can only provide a limited perspective toward inspecting the OOD samples.
>
> The visualizations generally show that HamOS can synthesize more diverse and representative outliers, which are distributed more widely, than VOS and NPOS. As can be seen in Figure 10, NPOS generates samples that mainly occupy the upper right region while leaving the lower left region nearly blank. HamOS generates OOD samples that mainly occupy the lower left region while some samples are spread through the upper right region as well as the cracks between ID clusters. Besides, HamOS generates samples that occupy more real outlier spots (i.e., yellow dots) than NPOS and VOS, demonstrating its superiority in generating more diverse outliers. Most importantly, HamOS outperforms NPOS on almost all benchmarks, demonstrating the effectiveness of HamOS.
>
> ## 2. The quality/utility of the generated OOD samples. [W2, Q3]
>
> **A2**: Thank you for bringing up such an inspiring point. As a well-studied insight, the quality of outliers significantly affects the model's detection performance [1], with diverse and representative outliers are demanded [2,3]. The primary focus of this paper is to boost the OOD detection performance through outlier synthesis. We formalize the novel framework HamOS which models the synthesis process as Markov chains and generates virtual outliers with the guidance of the innovative potential energy function. With this in mind, the quality of the generated OOD samples is directly reflected by the detection performance and partially highlighted by the OOD scores in Figure 2, 4(b)(c). High variance of the OOD scores indicates diverse OOD characteristics that the generated samples have.
>
> In previous works [2,3], task-specific metrics are adopted to validate the quality of outliers. In the context of outlier synthesis based solely on ID data, we investigated the underexplored Markov chain monte carlo for generating more diverse and representative outliers, which can significantly boost the detection performance as shown by our empirical analysis.
>
> The effect of outliers can also be influenced by how the outliers are involved in the training phase. In VOS, the energy of outliers is optimized to be higher; in NPOS, an additional binary classifier is trained to discriminate between ID and OOD. In HamOS, given our attempt to produce proper embeddings in the hyperspherical space, we optimize to push the outliers away from any ID clusters, as shown in Equation 7 of Section 3.3. While this simple practice demonstrates superior performance, more research space is left for finding advanced training schemes to engage the OOD supervision signals in future work.
>
> ## 3. Time cost compared to other methods. [Q1]
>
> **A3**: Another excellent point! We have evaluated the computational efficiency of HamOS in Appendix F.1 and present the time cost to synthesize a mini-batch of OOD samples for VOS, NPOS, and HamOS in Table 15. Though the HMC sampling algorithm typically incurs more computational cost than simpler Gaussian sampling, our method still keeps the time cost close to VOS and NPOS and even surpasses NPOS on ImageNet-1K. The computational efficiency results from two factors:
>
> - **Reduced amount of outliers.** Since HamOS can generate diverse and representative outliers, it necessitates fewer outliers to achieve superior performance than VOS and NPOS. In default configurations, HamOS only generates 20 OOD samples per class, whereas VOS samples 200 outliers and NPOS samples 400 outliers per class from Gaussian distribution. The low requirement of virtual outliers enables HamOS to perform efficiently, particularly in large-scale scenarios.
>
> - **Parallel matrix operations.** The Markov chains are generated parallelly across all classes using matrix operations. Additionally, the advanced similarity search technique also boosts the calculation of $k$-NN scores. This can be more straightforwardly verified by consulting the code in the supplementary material.
>
> Ultimately, HamOS can achieve superior performance against VOS and NPOS, as well as keeping the time cost feasible. We updated the time cost analysis of HamOS in the main context to ensure that readers have a clear understanding of HamOS's efficiency.

---

> ### Author Response · Authors · 2024-11-19
> **Response to Reviewer HEF3 (2/3)**
>
> ## 4. Effect of variants of HMC. [Q2]
>
> **A4**: Great point! Riemann Manifold HMC extends HMC by incorporating the position-dependent covariance matrix of the auxiliary momentum $\zq$, making RMHMC suitable for situations where the target distribution is concentrated on a low-dimensional manifold within the latent space. The Spherical RMHMC further extends the RMHMC with the constraint of the unit sphere. As a versatile framework, HamOS can be integrated with various score-based sampling algorithms, with the flexibility to adopt different sampling algorithms under different scenarios.
>
> We would like to remind you that we have conducted empirical analysis of several score-based sampling algorithms in Section 4.3 with random walk sampling as the baseline. The algorithms are introduced briefly in Appendix C.3. The empirical results are shown in Figure 6(a) with the full results displayed in Table 9 in Appendix F.1.
>
> We have specifically compared 4 score-based sampling algorithms, HMC (default), MALA, mMALA, and RMHMC. The empirical results show that the proposed HamOS framework is consistently effective with different sampling algorithms compared to the random walk baseline and other OOD detection baselines. There are slight performance differences between the score-based algorithms which are mainly due to the distinct characteristics of each algorithm. Different algorithms may exhibit different behaviors under various circumstances, suggesting that HamOS can be tuned for different scenarios indicating its remarkable versatility. Moreover, Langevin dynamics-based algorithms (MALA and mMALA) are more time-efficient compared to HMC-based algorithms, since they only calculate the gradient once per round instead of leveraging the Leagfrog discretization for multi-step updates.
>
> The HMC algorithm and its variants display numerous advantages compared to traditional score-based sampling algorithms, like score matching [4], Langevine dynamics [5], and diffusion models [6], contributing to the efficacy and efficiency of the HamOS framework. The advantages are outlined below:
>
> - HMC exhibits a high sampling rate with low computational burden. HMC can leverage the informative gradients of the potential energy function to guide the sampling process with Leapfrog integration for exploring a large area in the latent space, handling the complex target distribution in high dimensions. The direct guidance from the gradients results in a high acceptance rate of nearly 1.0, thereby keeping the computational burden practical. Parallel matrix operations can enhance the efficiency of the HMC sampling process.
>
> - The meticulously crafted estimation of the target unknown distribution enables the direct synthesis of OOD samples in the latent space. While other sampling algorithms struggle with the complex unknown distribution, such as Gaussian sampling that evenly explores all directions, HMC leverages explicit guidance to find OOD characteristics that can be directly employed to optimize the model for better detection performance.
>
> - Third, HamOS significantly exploits the information of ID priors by the class-conditional sampling between ID clusters. In the high-dimensional spherical space, midpoints of ID clusters serve as straightforward anchors as the roots of the Markov chains, guaranteeing the correctness of OOD synthesis. As the outliers are synthesized purely based on the ID data, HamOS has great generality to any dataset.
>
> In conclusion, our proposed HamOS framework not only demonstrates flexibility with various score-based sampling algorithms, distinguishing its versatility but also possesses superiority compared with traditional sampling algorithms. This discussion has been incorporated into the new revision.

---

> ### Author Response · Authors · 2024-11-19
> **Response to Reviewer HEF3 (3/3)**
>
> ## 5. Effect of OOD samples on ID accuracy. [Q3]
>
> **A5**: We are glad you bring up such an inspiring question. The quality/utility of synthesized samples is validated depending on what task they are generated for.
>
> As the primary purpose of this paper is to synthesize outliers to boost OOD detection performance, the synthesis process is designed specifically so that outliers with diverse levels of OOD characteristics are generated. In the paper you referenced [7], a subset of synthesized data is selected by CRAIG [8] and samples with low entropy are eliminated by an entropic filter, making the quality of the synthesized samples high enough to improve ACC. As the objectives are different, different techniques are applied to guarantee that high-quality samples are generated for their specific task. There may be techniques that can be designed to boost the classification performance as well as the detection performance, which we think is a potential avenue for future research.
>
> In the context of OOD detection, the ID accuracy should also be preserved. We managed to enhance the detection performance without sacrificing the classification capability by mainly two components.
>
> - The cross-entropy loss is adopted along with the fine-tuning phase to balance the optimization on the contrastive loss and the OOD discernment loss in Equation 8. This joint training scheme reaches a good equilibrium between the original classification task and OOD detection.
>
> - The original classification head is preserved, which not only maintains the ID accuracy but also makes the HamOS framework compatible with many other post-hoc detection methods that manipulate features, logits, or softmax scores. As the classification and the detection are separated, they can rarely impair each other.
>
> Our empirical results show that HamOS maintains ID accuracy significantly . In Table 1 and Table 2, in addition to achieving superior detection performance, HamOS also displays better ID classification performance than almost all regularization-based methods. The empirical study demonstrates the superiority of our 'Y-head' HamOS framework.
>
> Thanks again for such constructive comments. We will cite the paper and incorporate the discussion in the revision with the hope of inspiring more future works.
>
> [1] Dan Hendrycks, Steven Basart, Mantas Mazeika, Andy Zou, Joseph Kwon, Mohammadreza Mostajabi, Jacob Steinhardt, and Dawn Song. Scaling out-of-distribution detection for real-world settings. In Proceedings of the 39th International Conference on Machine Learning, volume 162, pp. 8759–8773, 2022.
>
> [2] Jianing Zhu, Geng Yu, Jiangchao Yao, Tongliang Liu, Gang Niu, Masashi Sugiyama, and Bo Han. Diversified outlier exposure for out-of-distribution detection via informative extrapolation. In Advances in Neural Information Processing Systems 36, 2023.
>
> [3] Wenyu Jiang, Hao Cheng, MingCai Chen, Chongjun Wang, and Hongxin Wei. DOS: Diverse outlier sampling for out-of-distribution detection. In Proceedings of the 12th International Conference on Learning Representations, 2024.
>
> [4] Aapo Hyvärinen. Estimation of non-normalized statistical models by score matching. Journal of Machine Learning Research, 6(24):695–709, 2005.
>
> [5] Max Welling and Yee Whye Teh. Bayesian learning via stochastic gradient langevin dynamics. In Proceedings of the 28th International Conference on Machine Learning, pp. 681–688, 2011.
>
> [6] Yang Song, Jascha Sohl-Dickstein, Diederik P Kingma, Abhishek Kumar, Stefano Ermon, and Ben Poole. Score-based generative modeling through stochastic differential equations. In Proceedings of the 9th International Conference on Learning Representations, 2021.
>
> [7] Omead Pooladzandi, Pasha Khosravi, Erik Nijkamp, and Baharan Mirzasoleiman. Generating high fidelity synthetic data via coreset selection and entropic regularization. In Advances in Neural Information Processing Systems 36 workshop on synthetic data for empowering ML research, 2023.
>
> [8] Baharan Mirzasoleiman, Jeff Bilmes, and Jure Leskovec. Coresets for data-efficient training of machine learning models. In Proceedings of the 37th International Conference on Machine Learning, volume 119, pp. 6950–6960, 2020.

---

> > ### Comment · Reviewer_HEF3 · 2024-11-22
> > **Makes Sense**
> >
> > The authors answered my questions sufficiently.
> >
> > I had missed the RMHMC comparison in Table 9. There are other papers that talk about how RMHMC do better for OOD, as it allows you to reweight the grads by the curvature, so naturally you can traverse the space better. In the experiments with RMHMC did y'all do the spherical equivalent of RMHMC.
> >
> > This is another paper that might be good to cite. They use a Hat EBM (https://arxiv.org/abs/2210.16486) and see that it does particularly well at identifying OOD samples. They look at OOD detection for CIFAR100 which aligns with Table 1 in your paper. If I compare the AUROC values HamOS is getting 83% vs Hat EBM which is getting 87% where higher is better. This makes the results presented in the paper a little confusing to me. It would be good to incorporate/note/etc the Hat EBM. Perhaps it is not an apples-to-apples comparison. Could you chat on this during the discussion period? I know there isn't much time left.
> >
> > I appreciate the authors going into many details during the rebuttals. I increase my score by 1.

---

> > > ### Author Response · Authors · 2024-11-23
> > > **Great thanks and discussion of additional questions**
> > >
> > > Dear Reviewer HEF3,
> > >
> > > Great thanks for your timely feedback and for raising the score! We are glad that our discussion addressed your concerns. Here we provide clarification for your further concerns.
> > >
> > > > In the experiments with RMHMC did y'all do the spherical equivalent of RMHMC.
> > >
> > > **A1**: Yes, we adopted the Spherical RMHMC compared to Spherical HMC. Specifically, we modified from Spherical HMC to form the Spherical RMHMC by conditioning the covariance matrix on previous states in Equation (13), thus dynamically traversing the latent space. Equation (3) guarantees the spherical equivalence, which ensures the newly proposed samples lie on the unit sphere.
> > >
> > > > Discussion on another related work
> > >
> > > **A2**: We appreciate your mentioning another related work Hat EBM [1]. Hat EBM incorporates the generator into its forward pass, thus enabling explicit modeling of the generator's outputs. As for your concern about the performance, the experiment in Hat EBM uses CIFAR-10 as ID data and CIFAR-100 as OOD data, which aligns with our setting in Table 6. Our HamOS achieves 91.36\% on AUROC in this setting. As the backbones and training configurations differ a lot, more experiments should be conducted so that the two methods are comparable. Despite this, we included Hat EBM in the Related Work for it offers a new architecture perspective.
> > >
> > > Thanks again for your valuable feedback!
> > >
> > > Best regards,
> > >
> > > Authors of Submission 2198
> > >
> > > [1] Mitch Hill, Erik Nijkamp, Jonathan Mitchell, Bo Pang, and Song-Chun Zhu. Learning probabilistic models from generator latent spaces with hat ebm. In Advances in Neural Information Processing Systems 36, 2024.

---

> ### Author Response · Authors · 2024-11-22
> **Kind reminder to Reviewer HEF3**
>
> Dear Reviewer HEF3,
>
> We sincerely appreciate your valuable comments! We understand that you may be too busy to check our rebuttal. We have added discussions about the computational cost of HamOS (see Section 4.3 and Appendix F.1) and the analysis of variants of HMC, including SpRMHMC (see Appendix G). We also clarified the visualization in Figure 10 and discussed the quality/utility of synthesized samples and their effect on ID-ACC (see Appendix G).
>
> We sincerely hope that the refinement can address your concerns. Please let us know if you still have any further concerns and we will be open to all possible discussions.
>
> Best regards,
>
> Authors of Submission 2198

---

### Official Review · Reviewer_5Vcs · 2024-11-03

**Soundness:** 4
**Presentation:** 3
**Contribution:** 3
**Rating:** 8
**Confidence:** 3

**Summary:**

The paper proposes HamOS (Hamiltonian Monte Carlo Outlier Synthesis), for out-of-distribution (OOD) detection that uses Hamiltonian Monte Carlo sampling to generate synthetic outliers from in-distribution data. The key innovation is using HMC to sample diverse outliers in a learned hyperspherical feature space, guided by an OOD-ness density estimation based on k-nearest neighbors. The framework combines classification, contrastive learning, and OOD discernment losses to learn representations that better separate ID and OOD data.

**Strengths:**

- This provides a novel application of HMC for outlier synthesis, and provides an elegant integration of hyperspherical embeddings with HMC sampling. Novel synthesis framework avoids the need for access to real outlier data.
- The benchmarks studied are comprehensive, written up in a well-structured experimental section. SoTA performance is achieved.
- The code in the supplementary section is cleanly integrated, kudos!
- Visualizations are intuitive and informative, well presented!

**Weaknesses:**

The paper presents Hamiltonian Monte Carlo (HMC) for sampling outliers in latent space, sharing similarities with score-based generative models. It defines an OOD-ness energy function: $U_{OOD}(z; Z_u, Z_v) = -\log P_{OOD}(z; Z_u, Z_v) = -\log \sum_{i=u,v} P_{OOD}(z; Z_i) + \text{constant}$, which is analogous to score models' $\nabla_x \log p(x)$ and HamOS's $\nabla_z U_{OOD}(z)$

The HMC sampling procedure follows $q^{(ℓ+1/2)} = q^{(ℓ)} - \frac{\epsilon}{2}(I_d - z^{(ℓ)}(z^{(ℓ)})^\top)\nabla_{z^{(ℓ)}} U_{OOD}(z^{(ℓ)})$, similar to Langevin dynamics $x_{t+1} = x_t + \frac{\epsilon}{2}\nabla_x \log p(x_t) + \sqrt{\epsilon}z_t$.

The main differentiator - HamOS's objective function $L_{HamOS} = L_{CE} + L_{ID-con} + \lambda_d L_{OOD-disc}$ combines classification, contrastive learning, and OOD detection, differentiating it from traditional diffusion models by sampling between ID clusters to find OOD regions rather than sampling from noise to data distribution.

To improve the paper, the authors should add a section comparing their work to score-based sampling, properly cite relevant prior work on gradient-based sampling in latent spaces, and explain their advantages over traditional score-based approaches. These advantages include computational efficiency, direct optimization for OOD detection, and class-conditional sampling between clusters. The paper could also discuss potential cross-pollination opportunities, such as incorporating score network architectures for OOD-ness estimation, adopting noise scheduling strategies from diffusion models, and utilizing classifier-free guidance techniques.

**Questions:**

- How does the performance of HamOS scale with feature dimension? Hyperspherical geometry might become problematic at higher dims.
- How sensitive is the method to choosing initial midpoints between the clusters?
- Could you provide more theoretical insight into what hard margin to choose and a principled way to select values for different datasets?
- What makes HMC-based sampling more effective than simpler Gaussian sampling approaches?
- See, above "weaknesses". I'd be willing to revise score upwards, provided this is addressed.

---

> ### Author Response · Authors · 2024-11-19
> **Response to Reviewer 5Vcs (1/5)**
>
> Thank you for your time devoted to reviewing this paper and your constructive suggestions. We are greatly encouraged that you find our method **novel** and **elegant** with **comprehensive** empirical results. Please find our response to your concerns below:
>
> ## 1. Advantages over other score-based sampling algorithms. [W1, Q5]
>
> **A1**: Excellent point! HamOS displays numerous advantages compared to traditional score-based sampling, such as score-based generative models. Besides, as a general framework, HamOS can be integrated with various score-based sampling algorithms, with the flexibility to adopt different score-based sampling algorithms in diverse contexts.
>
> HamOS displays numerous advantages compared to traditional score-based sampling algorithms, like score matching [1], Langevine dynamics [2], and diffusion models [3], demonstrating the efficacy and efficiency of the framework. The advantages are outlined below:
>
> - HMC exhibits a high sampling rate with low computational burden. Conventional score-based sampling may introduce extra learnable parameters which can increase the computational load. HMC can leverage the informative gradients of the potential energy function to guide the sampling process with Leapfrog integration for exploring a large area in the latent space, handling the complex target distribution in high dimensions. The direct guidance from the gradients results in a high acceptance rate of nearly 1.0, thereby keeping the computational burden practical. Parallel matrix operations can enhance the efficiency of the HMC sampling process.
>
> - The estimation of the target unknown distribution enables the direct synthesis of OOD samples in the latent space. The objective of traditional score-based sampling is typically to converge to the target distribution guided by the ID training data, e.g., diffusion models are supposed to generate legitimate data, prohibiting the ability to generate samples from unknown distributions. HamOS, however, aims to collect diverse samples along the Markov chains, not requiring iterative sampling to converge. While other sampling algorithms struggle with the complex unknown distribution, such as Gaussian sampling that evenly explores all directions, HMC leverages explicit guidance to find OOD characteristics that can be directly employed to optimize the model for better detection performance.
>
> - HamOS significantly exploits the information of ID priors for outlier synthesis by the class-conditional sampling between ID clusters. Traditional score-based sampling incorporates label information with the conditioned score model, which acts as a black box. In HamOS, with the OOD-ness estimation, outliers are explicitly sampled using the ID priors. Specifically, in the high-dimensional spherical space, midpoints of ID clusters serve as straightforward starting points of the Markov chains, guaranteeing the correctness of OOD synthesis. As the outliers are synthesized purely based on the ID data, HamOS has great generality to any dataset.
>
> We would like to remind you that we have conducted empirical analysis of several score-based sampling algorithms in Section 4.3 with random walk sampling as the baseline. The algorithms are introduced briefly in Appendix C.3. The empirical results are shown in Figure 6(a) with the full results displayed in Table 9 in Appendix F.1.
>
> We have specifically compared 4 score-based sampling algorithms, HMC (default), MALA, mMALA, and RMHMC. The empirical results show that the proposed HamOS framework is consistently effective with different sampling algorithms compared to the random walk baseline and other OOD detection baselines. There are slight performance differences between the score-based algorithms which are mainly due to the distinct characteristics of each algorithm. Different algorithms may exhibit different behaviors under various circumstances, suggesting that HamOS can be tuned for different scenarios indicating its remarkable versatility. Moreover, Langevin dynamics-based algorithms (MALA and mMALA) are more time-efficient compared to HMC-based algorithms, since they only calculate the gradient once per round instead of leveraging the Leagfrog discretization for multi-step updates.
>
> In conclusion, our proposed HamOS framework not only demonstrates flexibility with various score-based sampling algorithms, distinguishing its versatility but also possesses superiority compared with traditional sampling algorithms. This discussion has been incorporated into the new revision. By the way, if we misunderstood any of your concerns, please kindly point it out.

---

> ### Author Response · Authors · 2024-11-19
> **Response to Reviewer 5Vcs (2/5)**
>
> ## 2. Potential cross-pollination opportunities. [W2, Q5]
>
> **A2**: Thank you for such an inspiring suggestion! The synthesis process of HamOS has many components analogous to score-based generative models, presenting significant prospects for cross-pollination. Here we outline some ideas:
>
> - The gradient score can be produced by a hierarchical model instead of the $k$-NN metric. The fundamental process of generative models is to turn a noise point into a legitimate point in an iterative manner, in which a score model is adopted to predict the gradient without sampling directly from the target distribution. Analogously, we can develop a score model, that may integrate the information from ID priors to produce gradients as better guidance for the outlier synthesis in each iteration. But this may introduce additional parameters, increasing the computational cost.
>
> - Both HamOS and diffusion models adopt an iterative synthesis process which can be controlled by scheduling strategies. It is feasible to apply a schedule that can adjust dynamically based on the progress of outlier synthesis, which may potentially improve the quality of outliers by regulating the gradient scale. There are numerous schedules to choose from, such as linear, cosine, or any custom configuration. The adaptive scheduling strategy may also improve the versatility and adaptability of HamOS, accommodating diverse datasets, tasks, and data modalities.
>
> - Additional techniques of generative models, such as classifier-free guidance, offer diverse research directions for HamOS. The classifier-free guidance technique is primarily adopted to introduce guidance information directly into the diffusion process without training an additional classifier. In HamOS, classifier-free guidance may help in lightweightly distilling external OOD information to the synthesis process of OOD samples, leading to a better balance between diversity and representativeness, thus elevating the quality of OOD samples.
>
> - Conversely, the HamOS framework may serve as a beneficial prior work for improving the generative models. The non-parametric synthesis process may inspire more innovation in exploring and exploiting the latent space, leading to a high-quality, controllable, and efficient diffusion process.  As HamOS facilitates learning a proper representation that is ID-OOD differentiable, there is potential to improve the fidelity of generation by shaping the latent space for diffusion models.
>
> Thanks again for bringing up such an exhilarating point! We included this discussion in the revision, viewing it as a valuable avenue for future research.

---

> ### Author Response · Authors · 2024-11-19
> **Response to Reviewer 5Vcs (3/5)**
>
> ## 3. Performance of HamOS with higher feature dimensions. [Q1]
>
> **A3**: We appreciate your mention about that! Here we provide additional empirical study on different feature dimensions.
>
> In high-dimensional spaces, the Euclidean distance metric may be less discriminative, probably impairing the estimation of OOD-ness. The computational cost will also increase as the feature dimension scales. To assess the performance of HamOS with different feature dimensions, we conduct empirical analysis at dimensions 64, 256, and 512, alongside 4 representative baselines in **Table 1 & 2 below**. The results show that HamOS consistently achieves superior performance in almost all settings, demonstrating its stable performance across various feature dimensions. As the feature dimension scales up, HamOS only displays tiny performance degradation. Some baselines can be drastically influenced by the feature dimension, such as SSD+ which begins to collapse at 512 dimension on CIFAR-10 and PALM begins to collapse at 256 dimension on CIFAR-10 and 512 dimension on CIFAR-100. Surprisingly, PALM achieves high performance at 256 dimension on CIFAR-100, indicating that it requires careful tuning on the dimension to bring out the efficacy. Note that the dimension of the penultimate layer of the backbone is 512, so we scale up to 512 as a edge case.
>
>
> **Table 1. Ablation results on feature dimension with CIFAR-10 as ID dataset (\%).**
>
> | Method | FPR95↓ | AUROC↑ | AUPR↑ | ID-ACC↑ |
> |---|---|---|---|---|
> | **Dim. 64** |
> | SSD+ | 18.59 | 94.85 | 90.76 | 93.93 |
> | CIDER | 15.53 | 96.02 | 92.67 | 93.52 |
> | NPOS | 17.64 | 95.60 | 92.16 | 94.00 |
> | PALM | 44.36 | 84.02 | 76.43 | 93.87 |
> | **HamOS(ours)**| **12.66** | **96.16** | **93.47** | **94.51** |
> | **Dim. 128 (Default)** |
> | SSD+ | 18.49 | 94.85 | 90.88 | 93.95 |
> | CIDER | 16.28 | 95.76 | 92.36 | 93.98 |
> | NPOS | 14.39 | 96.61 | 93.35 | 93.95 |
> | PALM | 32.25 | 90.54 | 84.44 | 93.93 |
> | **HamOS(ours)** | **10.48** | **97.11** | **94.94** | **94.67** |
> | **Dim. 256** |
> | SSD+ | 16.05 | 95.74 | 92.36 | **94.66** |
> | CIDER | 15.50 | 96.37 | 93.10 | 93.89 |
> | NPOS | 13.04 | 97.00 | 93.83 | 94.00 |
> | PALM | 92.74 | 68.18 | 42.41 | 10.00 |
> | **HamOS(ours)** | **10.81** | **97.03** | **94.18** | **94.66** |
> | **Dim. 512** |
> | SSD+ | 85.59 | 66.83 | 46.64 | 93.46 |
> | CIDER | 13.38 | 96.69 | 93.30 | 93.72 |
> | NPOS | 15.22 | 96.27 | 92.79 | 94.33 |
> | PALM | 96.71 | 54.06 | 34.86 | 10.00 |
> | **HamOS(ours)** | **11.51** | **96.95** | **94.61** | **94.50** |
>
> **Table 2. Ablation results on feature dimension with CIFAR-100 as ID dataset (\%).**
>
> | Method | FPR95↓ | AUROC↑ | AUPR↑ | ID-ACC↑ |
> |---|---|---|---|---|
> | **Dim. 64** |
> | SSD+ | 54.66 | 80.58 | 69.83 | **75.88** |
> | CIDER | 48.36 | 82.38 | 74.54 | 74.74 |
> | NPOS | 67.74 | 77.01 | 62.78 | 74.48 |
> | PALM | 66.95 | 78.89 | 63.55 | 74.45 |
> | **HamOS(ours)** | **47.74** | **82.57** | **75.25** | 75.21 |
> | **Dim. 128 (Default)** |
> | SSD+ | 54.03 | 80.64 | 69.73 | 75.63 |
> | CIDER | 49.64 | 81.77 | 73.22 | 75.09 |
> | NPOS | 51.41 | 81.02 | 72.49 | 74.53 |
> | PALM | 55.13 | 79.95 | 70.21 | 74.67 |
> | **HamOS(ours)** | **46.68** | **83.64** | **75.52** | **76.12** |
> | **Dim. 256** |
> | SSD+ | 50.97 | 82.71 | 71.80 | **75.51** |
> | CIDER | 47.75 | 83.49 | 75.24 | 75.12 |
> | NPOS | 70.65 | 76.10 | 60.54 | 74.52 |
> | PALM | **46.81** | **83.62** | **75.48** | 73.69 |
> | **HamOS(ours)** | 47.26 | 83.61 | 75.36 | 75.28 |
> | **Dim. 512** |
> | SSD+ | 51.70 | 81.67 | 70.58 | 74.97 |
> | CIDER | 49.04 | 81.70 | 73.46 | 75.60 |
> | NPOS | 69.09 | 77.14 | 61.01 | 74.94 |
> | PALM | 95.97 | 42.63 | 31.44 | 1.00 |
> | **HamOS(ours)** | **48.93** | **82.19** | **74.35** | **75.99** |

---

> ### Author Response · Authors · 2024-11-19
> **Response to Reviewer 5Vcs (4/5)**
>
> ## 4. Sensitivity of HamOS to choosing initial midpoints. [Q2]
>
> **A4**: Another great point! Here we provide additional empirical analysis on the sensitivity of the initial midpoints.
>
> As our primary goal is to synthesize diverse and representative outliers with the estimation of the target unknown distribution produced by the $k$-NN distance metric, we iteratively sample data points from low OOD-ness region to high OOD-ness region. To this end, we choose the midpoints of ID cluster pairs with relatively low $k$-NN distances from ID clusters, while remaining confusing and pivotal for the two ID clusters.
>
> To examine the sensitivity of HamOS to the choice of midpoints, we apply Gaussian noise to the initial midpoints at different levels as small perturbations and display the results in **Table 3 & 4 below**. The scale of Gaussian noise is controlled by the standard deviation $\sigma$.  The results in Table 2 show that the performance of HamOS continuously deteriorates as the amount of Gaussian noise to the midpoints increases, implying that the choice of midpoints between ID clusters is essential. Intuitively, perturbations on the midpoints may compel the initial states into areas that are of high OOD-ness, undermining the diversity of outliers. The perturbed midpoints may also reside in the ID clusters, bringing down the acceptance rate.
>
> **Table 3. Perturbation on the initial midpoints with CIFAR-10 as ID dataset (\%).**
>
> | $\sigma$ | FPR95↓ | AUROC↑ | AUPR↑ | ID-ACC↑ |
> |---|---|---|---|---|
> | 0.0 (default) | **10.48** | **97.11** | **94.94** | 94.67 |
> | 0.001 | 10.53 | 96.96 | 94.38 | 94.48 |
> | 0.01 | 10.82 | 96.50 | 94.89 | 94.34 |
> | 0.05 | 11.08 | 96.49 | 94.04 | 94.44 |
> | 0.1 | 12.61 | 96.16 | 93.41 | **94.72** |
> | 0.2 | 11.83 | 96.69 | 94.12 | 94.33 |
>
> **Table 4. Perturbation on the initial midpoints with CIFAR-100 as ID dataset (\%).**
>
> | $\sigma$ | FPR95↓ | AUROC↑ | AUPR↑ | ID-ACC↑ |
> |---|---|---|---|---|
> | 0.0 (default) | **46.68** | **83.64** | **75.52** | **76.12** |
> | 0.001 | 47.18 | 83.63 | 74.87 | 75.64 |
> | 0.01 | 47.64 | 82.67 | 74.42 | 75.73 |
> | 0.05 | 47.43 | 82.55 | 74.02 | 75.63 |
> | 0.1 | 47.64 | 82.90 | 74.62 | 76.06 |
> | 0.2 | 48.11 | 82.74 | 74.19 | 75.76 |
>
> ## 5. Principled way to select hard margin values. [Q3]
>
> **A5**: We are glad you bring that up! We wish to notify you that we have performed an ablation study on the hard margin presented in Table 18 in Appendix F.2. As the results suggest, simply setting the hard margin $\delta$ to 0.1 can help achieve superior performance on CIFAR-10 and CIFAR-100. We also fixed the hard margin to 0.1 to attain SoTA performance on the large-scale ImageNet-1K. Our empirical analysis indicates that 0.1 is the optimal choice for the hard margin $\delta$ across different datasets. We deem that the the straightforward selection of value for the hard margin is due to its direct application to the log-likelihood which possesses universal geometric properties across various datasets as the feature embeddings are projected to the unit sphere. Intuitively, a lower hard margin will decrease the sampling acceptance rate, thus compromising the efficiency of HamOS, whereas a higher hard margin encourages erroneous outliers that have high ID probabilities conveying spurious OOD characteristics.

---

> ### Author Response · Authors · 2024-11-19
> **Response to Reviewer 5Vcs (5/5)**
>
> ## 6. Why HMC sampling is effective than Gaussian sampling? [Q4]
>
> **A6**: HMC sampling outperforms Gaussian sampling for mainly two reasons:
>
> - HMC samples diverse and representative outliers. As a well-studied insight, the quality of outliers substantially affects the model's detection performance [4], with diverse and representative outliers being demanded [5,6]. HMC can traverse a large area in the latent space from low OOD-ness region to high OOD-ness region, thus generating diverse outliers representing different levels of OOD characteristics. However, Gaussian sampling, limited by its variance, always generates samples concentrated to the mean point, prohibiting its ability to explore the hyperspherical space.
>
> - HMC samples more informative outliers compared to Gaussian sampling. With our crafted OOD-ness estimation, new outliers are generated with the gradient guidance, which explicitly points out the direction of the critical region that contains unknown OOD characteristics. The Gaussian sampling, however, stretches out to every direction evenly, resulting in generating outliers that may be less informative. The mix of outliers evenly distributed in every direction can hardly convey critical OOD characteristics to help the model learn differentiable representations.
>
> We updated this discussion to the revision to distinguish our HamOS from previous sampling algorithms.
>
> [1] Aapo Hyvärinen. Estimation of non-normalized statistical models by score matching. Journal of Machine Learning Research, 6(24):695–709, 2005.
>
> [2] Max Welling and Yee Whye Teh. Bayesian learning via stochastic gradient langevin dynamics. In Proceedings of the 28th International Conference on Machine Learning, pp. 681–688, 2011.
>
> [3] Yang Song, Jascha Sohl-Dickstein, Diederik P Kingma, Abhishek Kumar, Stefano Ermon, and Ben Poole. Score-based generative modeling through stochastic differential equations. In Proceedings of the 9th International Conference on Learning Representations, 2021.
>
> [4] Dan Hendrycks, Steven Basart, Mantas Mazeika, Andy Zou, Joseph Kwon, Mohammadreza Mostajabi, Jacob Steinhardt, and Dawn Song. Scaling out-of-distribution detection for real-world settings. In Proceedings of the 39th International Conference on Machine Learning, pp. 8759–8773, 2022.
>
> [5] Jianing Zhu, Geng Yu, Jiangchao Yao, Tongliang Liu, Gang Niu, Masashi Sugiyama, and Bo Han. Diversified outlier exposure for out-of-distribution detection via informative extrapolation. In Advances in Neural Information Processing Systems 36, 2023.
>
> [6] Wenyu Jiang, Hao Cheng, MingCai Chen, Chongjun Wang, and Hongxin Wei. DOS: Diverse outlier sampling for out-of-distribution detection. In Proceedings of the 12th International Conference on Learning Representations, 2024.

---

> ### Author Response · Authors · 2024-11-22
> **Kind reminder to Reviewer 5Vcs**
>
> Dear Reviewer 5Vcs,
>
> We sincerely appreciate your valuable comments! We understand that you may be too busy to check our rebuttal. We added the comparison with other score-based sampling algorithms (see Appendix G) and provided discussions about the potential of cross-pollination (see Appendix H). We also added an empirical study on the feature dimension (see Appendix F.2) and sensitivity of the initial midpoints (see Appendix F.1). We clarified the principled way to select the hard margin. The advantages of HMC over Gaussian sampling are also detailed.
>
> We sincerely hope that the refinement can address your concerns. Please let us know if you still have any further concerns and we will be open to all possible discussions.
>
> Best regards,
>
> Authors of Submission 2198

---

> > ### Comment · Reviewer_5Vcs · 2024-11-24
> >
> > Dear Authors, thank you for your detailed reply. I appreciate your commitment to your work, and the thorougness of the additional experimentation. I have revised your score upwards. Best of luck with your acceptance!

---

> > > ### Author Response · Authors · 2024-11-25
> > > **Great thanks!**
> > >
> > > Dear Reviewer 5Vcs,
> > >
> > > Thank you for your positive insightful review and for increasing our score!  We are delighted that our revisions have effectively addressed your concerns. We truly appreciate your support and guidance throughout this process.
> > >
> > > Best regards,
> > >
> > > The Authors of Submission 2198

---

### Official Review · Reviewer_hv7F · 2024-11-04

**Soundness:** 3
**Presentation:** 2
**Contribution:** 3
**Rating:** 6
**Confidence:** 4

**Summary:**

Many existing OOD detection methods rely on a large, high-quality set of natural outliers. To address limitations with synthetic outliers, this paper introduces the Hamiltonian Monte Carlo Outlier Synthesis (HamOS) framework, which uses Markov chains to generate diverse and representative outliers based solely on in-distribution data. HamOS achieves efficient and high-quality synthesis, showing superior performance over state-of-the-art baselines on standard and large-scale benchmarks.

**Strengths:**

- The motivation behind this method is clear and compelling, focusing on synthesizing effective virtual OOD data to enhance the robustness of machine learning models.
- Unlike traditional approaches that rely heavily on large, high-quality pools of natural outliers, this method leverages the technical innovation of Markov chains to generate diverse and representative outliers from only in-distribution data.
- This framework not only provides high-quality synthetic outliers efficiently but also demonstrates superior performance across multiple datasets, validating its effectiveness and efficiency compared to state-of-the-art methods in OOD detection.

**Weaknesses:**

- There are several detailed errors in the article that affect the reader's experience. For example, the caption of Figure 1 mentions ImageNet.
- The manuscript's explanations are difficult to follow, especially in the method section regarding how OOD samples are synthesized and how HMC is utilized. This is particularly challenging in the part involving Equation 3.

**Questions:**

- How does the proposed method compare to NPOS and VOS in terms of time cost?

- How stable are the experimental results (e.g., what is the standard deviation)?

---

> ### Author Response · Authors · 2024-11-19
> **Response to Reviewer hv7F**
>
> Thank you for your time devoted to reviewing this paper and your constructive comments! We are delighted that you find our motivation **clear** and **compelling** and our method to be **efficient** and **effective**. Please find our response to your concerns below:
>
> ## 1. Detailed errors, such as ImageNet in Figure 1. [W1]
>
> **A1**: Thank you for the suggestion! Regarding Figure 1, we intended to show HamOS's superior performance against the baselines. The x-axis represents the FPR95 on CIFAR-10, the y-axis represents the FPR95 on CIFAR-100, and the size of the dots indicates the AUROC on ImageNet-1K. We described the figure precisely in the main context to avoid confusion in the revised version. We have double-checked for any typographical errors to enhance the overall readability and clarity.
>
> ## 2. Difficult to follow the part involving Equation 3. [W2]
>
> **A2**: We are glad you bring that up! The outlier synthesis adopts the HMC procedure, which is explicitly detailed in Appendix C.3 to assist readers to understand how the samples are synthesized along with Markov chains.
>
> The key point of synthesizing OOD samples via HMC is to formulate the potential energy function, i.e., Equation 2. It is responsible for guiding the Markov chains to traverse extensively in embedding space and generate diverse and representative OOD samples. Since all samples are embedded onto a unit hypersphere, a spherical version of HMC [1] is employed, and Equation 3 describes how states are updated in spherical HMC.
>
> ## 3. Time cost compared to NPOS and VOS. [Q1]
>
> **A3**: Another excellent point! We have evaluated the computational efficiency of HamOS in Appendix F.1 and present the time cost to synthesize a mini-batch of OOD samples for VOS, NPOS, and HamOS in Table 15. Though the HMC sampling algorithm typically incurs more computational cost than simpler Gaussian sampling, our method still keeps the time cost close to VOS and NPOS and even surpasses NPOS on ImageNet-1K. The computational efficiency results from two factors:
>
> - **High quality sampling.** Since HamOS can generate diverse and representative outliers, it necessitates much fewer outliers to achieve superior performance than VOS and NPOS. In default configurations, HamOS only generates 20 OOD samples per class, whereas VOS samples 200 outliers and NPOS samples 400 outliers per class from Gaussian distribution. The low requirement of virtual outliers enables HamOS to perform efficiently, particularly in large-scale scenarios.
>
> - **High efficient implementation.** The Markov chains are generated parallelly across all classes using matrix operations. Additionally, the advanced similarity search technique also boosts the calculation of $k$-NN scores. This can be more straightforwardly verified by consulting the code in the supplementary material.
>
> Ultimately, HamOS can achieve superior performance against VOS and NPOS, as well as keeping the time cost feasible. We updated the time cost analysis of HamOS in the main context to ensure that readers have a clear understanding of HamOS's efficiency.
>
> ## 4. Stability of the experimental results. [Q2]
>
> **A4**: To demonstrate the stability of HamOS, we have reported the standard deviation beside the mean performance with smaller fonts in Table 1, 3. Full results are in Table 5. All experimental results in this work were acquired through multiple runs, confirming that HamOS consistently outperforms all the baselines. Additional empirical analysis on hard OOD benchmarks and the training-from-scratch scheme is also conducted to verify the stability of HamOS in Table 6, 7, 13, and 14. We clarify the standard deviation in the main context in the revised version.
>
> [1] Shiwei Lan, Bo Zhou, and Babak Shahbaba. Spherical hamiltonian monte carlo for constrained target distributions. In Proceedings of the 30th International Conference on Machine Learning, volume 32, pp. 629–637, 2013.

---

> ### Author Response · Authors · 2024-11-22
> **Kind reminder to Reviewer hv7F**
>
> Dear Reviewer hv7F,
>
> We sincerely appreciate your valuable comments! We understand that you may be too busy to check our rebuttal. We have clarified Figure 1 in Section 1 as well as fixed some minor typos. We also added computational comparion with VOS and NPOS (see Section 4.3) and described the standard deviation (see Section 4.1). The synthesis process is also detailed.
>
> We sincerely hope that the refinement can address your concerns. Please let us know if you still have any further concerns and we will be open to all possible discussions.
>
> Best regards,
>
> Authors of Submission 2198

---

> > ### Comment · Reviewer_hv7F · 2024-11-25
> > **Official Comment of Reviewer hv7F**
> >
> > Thank you for your reply. I will keep my original score.

---

> > > ### Author Response · Authors · 2024-11-25
> > > **Many Thanks!**
> > >
> > > Dear Reviewer hv7F,
> > >
> > > Thank you very much for your thoughtful review and for taking the time to read our response. We appreciate your decision to maintain the original score and your valuable advice during this procedure.
> > >
> > > Best regards,
> > >
> > > Authors of Submission 2198

---

### Author Response · Authors · 2024-11-20
**General Response to Reviewers and Revision Submitted**

We would like to thank the reviewers for their helpful and constructive comments. We are very glad to see that the reviewers find our proposed method **novel**, **innovative**, and **distinct** (5Vcs, by3L), with **clear** and **compelling** motivation (hv7F). We are delighted that they view the **technically innovation** of Markov chain modeling as **principled** and **theoretically sound** (hv7F, HEF3), and the application of HMC is **creative**, **unique**, and **novel** (5Vcs, by3L). Besides, the HamOS is **efficient** and **effective**, achieving **superior**, **SoTA** performance on **comprehensive** benchmarks (hv7F, 5Vcs). We are also delighted that our code is **cleanly integrated** and the visualizations are **intuitive** and **informative** (5Vcs).

We have addressed the reviewers' comments and concerns in individual responses to each reviewer. The reviews allowed us to improve our manuscript and the changes have been updated in the manuscript as summarized below:

For Reviewer hv7F:

- Clarified Figure 1 (see Section 1).
- Clarified the computational efficiency of HamOS (see Section 4.3 and Appendix F.1).
- Clarified the standard deviation of the results (see Section 4.1).
- Fixed some minor issues on grammar and typos.

For Reviewer 5Vcs:

- Clarified the advantages of HamOS compared to traditional score-based sampling (see Appendix G).
- Discussed the potential for cross-pollination (see Appendix H).
- Added the ablation study on the feature dimension (see Appendix F.2).
- Added the empirical analysis of the sensitivity of the initial midpoints (see Appendix F.1).

For Reviewer HEF3:

- Clarified the computational efficiency of HamOS (see Section 4.3 and Appendix F.1).
- Clarified the advantages of HamOS compared to other score-based sampling algorithms (see Appendix G).
- Clarified the effect of HamOS on ID-ACC (see Appendix G).
- Discussed on the quality of synthesized samples (see Appendix G).

For Reviewer by3L:

- Clarified the advantages of HamOS compared to other sampling algorithms (see Appendix G).
- Discussed the significance of HamOS to foundation models (see Appendix H).
- Discussed the application of HamOS on other data modalities (see Appendix G).
- Clarified the definition of "OOD-ness" (see Appendix G).
- Clarified the relation between inter-class dispersion and ID accuracy (see Section 4.2 and Appendix F.1).
- Revised "paradigm" to "framework" (see Section 1).

All changes have been highlighted in blue in the manuscript.

We appreciate your comments and time! We have tried our best to address your concerns and revised the paper following the suggestions. **Would you mind checking it and confirming if you have further questions?**

---

### Meta-Review · Area_Chair_9ghM · 2024-12-22

**Metareview:**

This paper presents HamOS, a framework for out-of-distribution (OOD) detection that uses Hamiltonian Monte Carlo (HMC) to synthesize outliers. The method models the synthesis process as Markov chains that traverse feature space guided by a potential energy function. Unlike existing approaches that rely on large pools of natural outliers or complex generative models, HamOS generates representative outliers using only in-distribution data.

Reviewers appreciated the technical soundness and the empirical validation across multiple benchmarks.
The reviews identified two primary concerns: The comparison to existing score-based sampling methods needed additional depth, particularly in explaining the advantages over simpler approaches. Additionally, reviewers would have liked experiments on non-vision task.
The authors adequately addressed most of the concerns in the rebuttal.
Most reviewers recommend acceptance, and I agree.

**Additional Comments On Reviewer Discussion:**

See above

---

### Decision · Program_Chairs · 2025-01-22

Accept (Poster)